# Non-Asymptotic Gap-Dependent Regret Bounds for Tabular MDPs

**Max Simchowitz**
UC Berkeley
msimchow@berkeley.edu

Kevin Jamieson
University of Washington
jamieson@cs.washington.edu

## Abstract

This paper establishes that optimistic algorithms attain gap-dependent and non-asymptotic logarithmic regret for episodic MDPs. In contrast to prior work, our bounds do not suffer a dependence on diameter-like quantities or ergodicity, and smoothly interpolate between the gap dependent logarithmic-regret, and the $\widetilde{\mathcal{O}}(\sqrt{HSAT})$-minimax rate. The key technique in our analysis is a novel "clipped" regret decomposition which applies to a broad family of recent optimistic algorithms for episodic MDPs.

## 1 Introduction

Reinforcement learning (RL) is a powerful paradigm for modeling a learning agent's interactions with an unknown environment, in an attempt to accumulate as much reward as possible. Because of its flexibility, RL can encode such a vast array of different problem settings - many of which are entirely intractable. Therefore, it is crucial to understand what conditions enable an RL agent to effectively learn about its environment, and to account for the success of RL methods in practice.

In this paper, we consider tabular Markov decision processes (MDPs), a canonical RL setting where the agent seeks to learn a *policy* mapping discrete states $x \in \mathcal{S}$ to one of finitely many actions $a \in \mathcal{A}$, in an attempt to maximize cumulative reward over an episode horizon $H$. We shall study the *regret* setting, where the learner plays a policy $\pi_k$ for a sequence of episodes $k = 1, \dots, K$, and suffers a regret proportional to the average sub-optimality of the policies $\pi_1, \dots, \pi_K$.

In recent years, the vast majority of literature has focused on obtaining *minimax* regret bounds that match the worst-case dependence on the number states $|\mathcal{S}|$, actions $|\mathcal{A}|$, and horizon length $H$; namely, a cumulative regret of $\sqrt{H|\mathcal{S}||\mathcal{A}|T}$, where $T = KH$ denotes the total number of rounds of the game [1]. While these bounds are succinct and easy to interpret, they paint an overly pessimistic account of the complexity of these problems, and do not elucidate the favorable structural properties of which a learning agent can hope to take advantage.

The earlier literature, on the other hand, establishes a considerable more favorable regret of the form $C \log T$, where $C$ is an instance-dependent constant given in terms of the *sub-optimality gaps* associated with each action at a given state, defined as

$$\texttt{gap}_\infty(x, a) = \mathbf{V}^{\pi^\star}(x) - \mathbf{Q}^{\pi^\star}(x, a), \tag{1}$$

where $\mathbf{V}^{\pi^\star}$ and $\mathbf{Q}^{\pi^\star}$ denote the value and $Q$ functions for an optimal policy $\pi^\star$, and the subscript-$\infty$ denotes these bounds hold for a non-episodic, infinite horizon setting. Depending on the constant $C$, the regret $C \log T$ can yield a major improvement over the $\sqrt{T}$ minimax scaling. Unfortunately, these analyses are asymptotic in nature, and only take effect after a large number of rounds, depending on other potentially-large, highly-conservative, or difficult-to-verify problem-dependent quantities such as hitting times or measures of uniform ergodicity [8, 13, 10].

To fully account for the empirical performance of RL algorithms, we seek regret bounds which take advantage of favorable problem instances, but apply in *finite time* and for practically realistic numbers of rounds $T$.

**Contributions:** As a first step in this direction, [14] introduced a novel algorithm called EULER, which enjoys reduced dependence on the episode horizon $H$ for favorable instances, while maintaining the same worst-case dependence for other parameters in their analysis as in [1].

In this paper, we take the next step by demonstrating that a common class of algorithms for solving MDPs, based on the *optimism* principle, attains gap-dependent, problem-specific bounds similar to those previously found only in the asymptotic regime. For concreteness, we specialize our analysis to a minor modification of the EULER algorithm we call StrongEuler; as we explain in Section 3, our analysis extends more broadly to other optimistic algorithms as well. We show that

- For any episodic MDP $\mathcal{M}$, StrongEuler enjoys a high probability regret bound of $C_{\mathcal{M}} \log(1/\delta)$ for all rounds $T \geq 1$, where the constant $C_{\mathcal{M}}$ depends on the sub-optimality gaps between actions at different states, as well as the horizon length, and contains an additive almost-gap-independent term that scales as $AS^2 \text{poly}(H)$ (Corollary 2.1).

Unlike previous gap-dependent regret bounds,

- The constant $C_{\mathcal{M}}$ does not suffer worst-case dependencies on other problem dependent quantities such as mixing times, hitting times or measures of ergodicity. However, the constant $C_{\mathcal{M}}$ *does* take advantage of *benign* problem instances (Definition 2.2).

- The regret bound of $C_{\mathcal{M}} \log(1/\delta)$ is valid for any total number of rounds $T \geq 1$. Selecting $\delta = 1/T$, this implies a *non-asymptotic* expected regret bound of $C_{\mathcal{M}} \log T$[1].

- The regret of StrongEuler interpolates between instance-dependent regret $C_{\mathcal{M}} \log T$ and minimax regret $\widetilde{\mathcal{O}}(\sqrt{H|\mathcal{S}||\mathcal{A}|T})$, the latter of which may be sharper for smaller $T$ (Theorem 2.4). Following [14], this dependence on $H$ may also be refined for benign instances.

Lastly, while the StrongEuler algorithm affords sharper regret bounds than past algorithms, our analysis techniques extend more generally to other optimism based algorithms:

- We introduce a novel "clipped" regret decomposition (Proposition 3.1) which applies to a broad family of optimistic algorithms, including the algorithms analyzed in [14, 6, 5, 9, 1].

- Following our analysis of StrongEuler, the clipped regret decomposition can establish analogous gap-dependent $\log T$-regret bounds for many of the algorithms mentioned above.

**What is $C_{\mathcal{M}}$?** In many settings, we show that $C_{\mathcal{M}}$ is dominated by an analogue to the sum over the reciprocals of the gaps defined in (1). This is known to be optimal for non-dynamic MDP settings like contextual bandits, and we prove a lower bound (Proposition 2.2) which shows that this is unimprovable for general MDPs as well. Furthermore, building on [14], we show this adapts to problems with additional structure, yielding, e.g., a horizon $H$-free bound for contextual bandits.

However, our gap-dependent bound also suffers from a certain dependence on the *smallest nonzero gap* $\text{gap}_{\min}$ (see Definition 2.1), which may dominate in some settings. We prove a lower bound (Theorem 2.3) which shows that optimistic algorithms in the recent literature - including StrongEuler - necessarily suffer a similar term in their regret. We believe this insight will motivate new algorithms for which this dependence can be removed, leading to new design principles and actionable insights for practitioners. Finally, our regret bound incurs an (almost) gap-independent burn-in term, which is standard for optimistic algorithms, and which we believe is an exciting direction of research to remove.

Altogether, we believe that the results in our paper serve as a preliminary but significant step to attaining sharp, instance-dependent, and *non-asymptotic* bounds for tabular MDPs, and hope that our analysis will guide the design of future algorithms that attain these bounds.

## 1.1 Related Work

Like the multi-armed bandit setting, regret bounds for MDP algorithms have been characterized both in *gap-independent* forms that rely solely on $S := |\mathcal{S}|, A := |\mathcal{A}|, H, T$, and in *gap-dependent* forms

which take into account the gaps (1), as well as other instance-specific properties of the rewards and transition probabilities.

**Finite Sample Bounds, Gap-Independent Bounds:** A number of notable recent works give undiscounted regret bounds for finite-horizon, tabular MDPs, nearly all of them relying on the principle of optimism which we describe in Section 3 [4, 1, 5, 9, 14]. Many of the more recent works [1, 14, 6] attain a regret of $\sqrt{HSAT}$, matching the known lower bound of $\sqrt{HSAT}$ established in [11, 8, 4]. As mentioned above, the EULER algorithm of [14] attains the minimax rates and simultaneously enjoys a reduced dependence on $H$ in benign problem instances, such as the contextual bandits setting where the transition probabilities do not depend on the current state or learners actions, or when the total cumulative rewards over any roll-out are bounded by 1 in magnitude.

**Diameter Dependent Bounds:** In the setting of infinite horizon MDPs with discounted regret, many previous works have established logarithmic regret bounds of the form $C(\mathcal{M}) \log T$, where $C(\mathcal{M})$ is a constant depending on the underlying MDP. Notably, [8] give an algorithm which attains a $\widetilde{\mathcal{O}}(\sqrt{D^2 S^2 AT})$ gap-independent regret, and an $\widetilde{\mathcal{O}}(\frac{D^2 S^2 A}{\mathrm{gap}_*} \log(T))$ gap-dependent regret bound, where $\mathrm{gap}_*$ is the difference between the mean infinite-horizon reward of $\pi_*$ and the next-best stationary policy, and where $D$ denotes the maximum expected traversal time between any two states $x, x'$, under the policy which attains the minimal traversal time between those two states. We note that if $\mathrm{gap}_\infty(x, a)$ denotes the sub-optimality of any action $a$ at state $x$ as in (1), then $\mathrm{gap}_* \leq \min_{x,a} \mathrm{gap}_\infty(x, a)$. The bounds in this work, on the other hand, depend on an average over inverse gaps, rather than a worst case. Moreover, the diameter $D$ can be quite large when there exist difficult-to-access states. We stress that the bound due to [8] is non-asymptotic, but the bound in terms of $\mathrm{gap}_*$ dependences other worst-case quantities measuring ergodicity.

**Asymptotic Bounds:** Prior to [8], and building on the bounds of [3], [13] presented bounds in terms of a diameter-related quantity $\bar{D} \geq D$, which captures the minimal hitting time between states when restricted to optimal policies. [13] prove that their algorithm enjoys a regret[2] of $\sum_{(s,a) \in \mathsf{CRIT}} \frac{\bar{D}^2}{\mathrm{gap}_\infty(x,a)} \log(T)$ asymptotically in $T$ where $\mathsf{CRIT}$ contains those sub-optimal state-action pairs $(x, a)$ such that $a$ can be made to the the unique, optimal action at $x$ by replacing $p(s'|s, a)$ with some other vector on the $\mathcal{S}$-simplex. Recently, [10] present per-instance lower bounds for both structured and unstructured MDPs, which apply to any algorithm which enjoys sub-linear regret on any problem instance, and an algorithm which matches these bounds asymptotically. This bound replaces $\bar{D}^2$ with $\bar{H}^2$, where $\bar{H}$ denotes the range of the bias functions, an analogue of $H$ for the non-episodic setting [2]. We further stress that whereas the logarithmic regret bounds of [8] hold for finite time with polynomial dependence on the problem parameters, the number of episodes needed for the bounds of [3, 13, 10] to hold may be exponentially large, and depend on additional, pessimistic problem-dependent quantities (e.g. a uniform hitting time in Proposition 29 in [12]).

**Novelty of this work:** The major contribution of our work is showing problem-dependent $\log(T)$ regret bounds which i) attain a refined dependence on the gaps, as in [13], ii) apply in finite time after a burn-in time only polynomial in $S$, $A$, $H$ and the gaps, iii) depend only on $H$ and not on the diameter $D$ (and thus, are not adversely affected by difficult to access states), and iv) smoothly interpolate between $\log T$ regret and the minimax $\sqrt{HSAT}$ rate attained by [1] et seq.

## 1.2 Problem Setting, Notation, and Organization

**Episodic MDP:** A *stationary*, episodic MDP is a tuple $\mathcal{M} := (\mathcal{S}, \mathcal{A}, H, r, p, p_0, R)$, where for each $x \in \mathcal{S}, a \in \mathcal{A}$ we have that $R(x, a) \in [0, 1]$ is a random reward with expectation $r(x, a)$, $p : \mathcal{S} \times \mathcal{A} \to \Delta^{\mathcal{S}}$ denotes transition probabilities, $p_0 \in \Delta^{\mathcal{S}}$ is an initial distribution over states, and $H$ is the horizon, or length of the episode. A policy $\pi$ is a sequence of mappings $\pi_h : \mathcal{S} \to \mathcal{A}$. For our given MDP $\mathcal{M}$, we let $\mathbb{E}^\pi$ and $\mathbb{P}^\pi$ denote the expectation and probability operator with respect to the law of sequence $(x_1, a_1), \ldots, (x_H, a_H)$, where $x_1 \sim p_0$, $a_h = \pi_h(x_h)$, $x_{h+1} \sim p(x_h, a_h)$. We define the *value* of $\pi$ as $\mathbf{V}_0^\pi := \mathbb{E}^\pi \left[ \sum_{h=1}^H r(x_h, a_h) \right]$ and for $h \in [H]$ and $x \in \mathcal{S}$, $\mathbf{V}_h^\pi(x) := \mathbb{E}^\pi \left[ \sum_{h' \geq h}^H r(x_{h'}, a_{a'}) \mid x_h = x \right]$, which we identify with a vector in $\mathbb{R}^{\mathcal{S}}$. We define the associated Q-function $\mathbf{Q}^\pi : \mathcal{S} \times \mathcal{A} \to \mathbb{R}$, $\mathbf{Q}_h^\pi(x, a) := r(x, a) + p(x, a)^\top \mathbf{V}_{h+1}^\pi$, so that

$\mathbf{Q}_h^{\pi}(x, \pi_h(x)) = \mathbf{V}_h^{\pi}(x)$. We denote the *set* of optimal policies $\pi^{\star} := \arg\max_{\pi} \mathbf{V}_0^{\pi}$, and let $\pi_h^{\star}(x) := \{a : \pi_h(x) = a, \pi \in \pi^{\star}\}$ denote the set of optimal actions. Lastly, given any optimal $\pi \in \pi^{\star}$, we introduce the shorthand $\mathbf{V}_h^{\star} = \mathbf{V}_h^{\pi}$ and $\mathbf{Q}_h^{\star} = \mathbf{Q}_h^{\pi}$, where we note that even when $\pi$ is not unique, $\mathbf{V}_h^{\star}$ and $\mathbf{Q}_h^{\star}$ do not depend on the choice of optimal policy.

**Episodic Regret:** We consider a game that proceeds in rounds $k = 1, \ldots, K$, where at each state an algorithm Alg selects a policy $\pi_k$, and observes a roll out $(x_1, a_1), \ldots, (x_H, a_H) \sim \mathbb{P}^{\pi_k}$. The goal is to minimize the cumulative simple regret, defined as $\mathrm{Regret}_K := \sum_{k=1}^{K} \mathbf{V}_0^{\star} - \mathbf{V}_0^{\pi_k}$.

**Notation and Organization:** For $n \in \mathbb{N}$, we define $[n] = \{1, \ldots, n\}$. For two expressions $f, g$ that are functions of any problem-dependent variables of $\mathcal{M}$, we say $f \lesssim g$ ($f \gtrsim g$, respectively) if there exists a universal constant $c > 0$ independent of $\mathcal{M}$ such that $f \leq cg$ ($f \geq cg$, respectively). $\lessgtr$ will denote an informal, approximate inequality. Section 2 presents our main results, and Section 3 sketches the proof and highlights the novelty of our techniques. All references to the appendix refer to the appendix of the supplement. All formal proofs, and many rigorous statement of results, are deferred to the appendix, whose organization and notation are described at length in Appendix A.

## 1.3   Optimistic Algorithms

Lastly, we introduce *optimistic algorithms* which select a policy which is optimal for an over-estimated, or *optimistic*, estimate of the true $Q$-function, $\mathbf{Q}^{\star}$.

**Definition 1.1** (Optimistic Algorithm). We say that an algorithm Alg satisifes *optimism* if, for each round $k \in [K]$ and stage $h \in [H]$, it constructs an *optimistic Q-function* $\overline{\mathbf{Q}}_{k,h}(x, a)$ and policy $\pi_k = (\pi_{k,h})$ satisfying $\forall x, a : \overline{\mathbf{Q}}_{k,H+1}(x, a) = 0, \overline{\mathbf{Q}}_{k,h}(x, a) \geq \mathbf{Q}_h^{\star}(x, a)$, and $\pi_{k,h}(x) \in \arg\max_a \overline{\mathbf{Q}}_{k,h}(x, a)$. The associated *optimistic value function* is $\overline{\mathbf{V}}_{k,h}(x) := \overline{\mathbf{Q}}_{k,h}(x, \pi_{k,h}(x))$.

We shall colloquially refer to an algorithm as *optimistic* if it satsifies optimism with high probability. Optimism has become the dominant approach for learning finite-horizon MDPs, and all recent low-regret algorithms are optimistic [5, 6, 1, 14, 9]. In *model-based* algorithms, the overestimates $\overline{\mathbf{Q}}_{k,h}$ are constructed recursively as $\overline{\mathbf{Q}}_{k,h}(x, a) = \widehat{r}_k(x, a) + \widehat{p}_k(x, a)^{\top} \overline{\mathbf{V}}_{k,h+1} + \mathbf{b}_{k,h}(x, a)$, where $\widehat{r}_k(x, a)$ and $\widehat{p}_k(x, a)$ are empirical estimates of the mean rewards and transition probabilities, and $\mathbf{b}_{k,h}(x, a) \geq 0$ is a confidence *bonus* designed to ensure that $\overline{\mathbf{Q}}_{k,h}(x, a) \geq \mathbf{Q}^{\star}(x, a)$. Letting $n_k(x, a)$ denote the total number of times a given state-action pair is visited, a simple bonus $\mathbf{b}_{k,h}(x, a) \approx \sqrt{\frac{H \log(SAHK/\delta)}{n_k(x,a)}}$ suffices to induce optimism, yielding the UCBVI-CH algorithm of [1]. This leads to an episodic regret bound of $\sqrt{H^2 SAT}$, a factor of $\sqrt{H}$ greater than the minimax rate. More refined bonuses based on the "Bernstein trick" achieve the optimal $H$-dependence [1], and the EULER algorithm of [14] adopts further refinements to replace worst-case $H$ dependence with more adaptive quantities. The StrongEuler algorithm considered in this work applies similarly adaptive bonuses, but our analysis extends to all aforementioned bonus configurations. We remark that there are also *model-free* optimistic algorithms based on Q-learning (see, e.g. [9]) that construct overestimates in a slightly different fashion. While our main technical contribution, the clipped regret decomposition (Proposition 3.1), applies to *all* optimistic algorithms, our subsequent analysis is tailored to model-based approaches, and may not extend straightforwardly to Q-learning methods.

## 2   Main Results

**Logarithmic Regret for Optimistic Algorithms:** We now state regret bounds that describe the performance of StrongEuler, an instance of the model-based, optimistic algorithms described above. StrongEuler is based on carefully selected bonuses from [14], and formally instantiated in Algorithm 1 in Appendix E. We emphasize that other optimistic algorithms enjoy similar regret bounds, but we restrict our analysis to StrongEuler to attain the sharpest $H$-dependence. The key quantities at play are the *suboptimality-gaps* between the Q-functions:

**Definition 2.1** (Suboptimality Gaps). For $h \in [H]$, define the stage-dependent suboptimality gap $\mathrm{gap}_h(x, a) := \mathbf{V}_h^{\star}(x) - \mathbf{Q}_h^{\star}(x, a)$, as well as the minimal stage-independent gap $\mathrm{gap}(x, a) := \min_h \mathrm{gap}_h(x, a)$, and the minimal gap $\mathrm{gap}_{\min} := \min_{x,a,h}\{\mathrm{gap}_h(x, a) : \mathrm{gap}_h(x, a) > 0\}$.

Note that any optimal $a^{\star} \in \pi_h^{\star}(x)$ satisfies the Bellman equation $\mathbf{Q}_h^{\star}(x, a^{\star}) = \max_a \mathbf{Q}_h^{\star}(x, a) = \mathbf{V}_h^{\star}(x)$, and thus $\mathrm{gap}_h(x, a^{\star}) = 0$ iff $a^{\star} \in \pi_h^{\star}(x)$. Following [14], we consider two illustrative benign problem settings which afford an improved dependence on the horizon $H$:

**Definition 2.2** (Benign Settings). We say that an MDP $\mathcal{M}$ is a *contextual bandit instance* if $p(x'|x,a)$ does not depend on $x$ or $a$. An MDP $\mathcal{M}$ has $\mathcal{G}$-*bounded rewards* if, for any policy $\pi$, $\sum_{h=1}^{H} R(x_h, a_h) \leq \mathcal{G}$ holds with probability 1 over trajectories $((x_h, a_h)) \sim \mathbb{P}^{\pi}$.

Lastly, we define $\mathcal{Z}_{\mathrm{opt}}$ as the set of pairs $(x,a)$ for which $a$ is optimal at $x$ for some stage $h \in [H]$: $\mathcal{Z}_{\mathrm{opt}} := \{(x,a) : \exists h \in [H] \text{ with } a \in \pi_h^{\star}(x)\}$ and its complement $\mathcal{Z}_{\mathrm{sub}} := \mathcal{S} \times \mathcal{A} - \mathcal{Z}_{\mathrm{opt}}$. Note that typically $|\mathcal{Z}_{\mathrm{opt}}| \lesssim H|\mathcal{S}|$ or even $|\mathcal{Z}_{\mathrm{opt}}| \lesssim |\mathcal{S}|$ (see Remark B.2 in the appendix). We now state our first result, which gives a gap-dependent regret bound that scales as $\log(1/\delta)$ with probability at least $1 - \delta$. The result is a consequence of a more general result stated as Theorem 2.4, itself a simplified version of more precise bounds stated in Appendix B.1.

**Corollary 2.1.** *Fix $\delta \in (0, 1/2)$, and let $A = |\mathcal{A}|$, $S = |\mathcal{S}|$, $M = (SAH)^2$. Then with probability at least $1 - \delta$,* StrongEuler *run with confidence parameter $\delta$ enjoys the following regret bound for all $K \geq 1$:*

$$
\mathrm{Regret}_K \lesssim \left( \sum_{(x,a) \in \mathcal{Z}_{\mathrm{sub}}} \frac{H^3}{\mathtt{gap}(x,a)} \log \frac{MT}{\delta} \right) + \frac{H^3 |\mathcal{Z}_{\mathrm{opt}}|}{\mathtt{gap}_{\mathtt{min}}} \log \frac{MT}{\delta}
$$
$$
+ H^4 SA(S \vee H) \log \frac{MH}{\mathtt{gap}_{\mathtt{min}}} \log \frac{MT}{\delta}. \tag{2}
$$

*Moreover, if $\mathcal{M}$ is either a contextual bandits instance, or has $\mathcal{G}$-bounded rewards for $\mathcal{G} \lesssim 1$, then the factors of $H^3$ on the first line can be sharped to $H$. In addition, if $\mathcal{M}$ is a contextual bandits instance, the factor of $H^3$ in the first term (summing over $(x,a) \in \mathcal{Z}_{\mathrm{sub}}$) can be sharped to 1.*

Setting $\delta = 1/T$ and noting that $\sum_{k=1}^{K} \mathbf{V}_0^{\star} - \mathbf{V}_0^{\pi_k} \leq KH = T$ with probability 1 (recall $R(x,a) \in [0,1]$), we see that the expected regret $\mathbb{E}[\sum_{k=1}^{K} \mathbf{V}_0^{\star} - \mathbf{V}_0^{\pi_k}]$ can be bounded by replacing $1/\delta$ with $T$ in right hand side of the inequality (2); this yields an expected regret that scales as $\log T$.

**Three regret terms:** The first term in Corollary 2.1 reflects the sum over sub-optimal state-action pairs, which a lower bound (Proposition 2.2) shows is unimprovable in general. In the infinite horizon setting, [10] gives an algorithm whose regret is asymptotically bounded by an analogue of this term. The third term characterizes the burn-in time suffered by nearly all model-based finite-time analyses and is the number of rounds necessary before standard concentration of measure arguments kick in. The second term is less familiar and is addressed in Section 2.2 below.

$H$ **dependence:** Comparing to known results from the infinite-horizon setting, one expects the optimal dependence of the first term on the horizon to be $H^2$. However, we cannot rule out that the optimal dependence is $H^3$ for the following three reasons: (i) the infinite-horizon analogues $D, \bar{D}, \bar{H}$ (Section 1.1) are not directly comparable to the horizon $H$; (ii) in the episodic setting, we have a potentially different value function $\mathbf{V}_h^{\star}$ for each $h \in [H]$, whereas the value functions of the infinite horizon setting are constant across time; (iii) the $H^3$ may be unavoidable for non-asymptotic (in $T$) bounds, even if $H^2$ is the optimal asymptotic dependence after sufficient burn-in (possibly depending on diameter-like quantities). Resolving the optimal $H$ dependence is left as future work. We also note that for contextual bandits, we incur *no $H$* dependence on the first term; and thus the first term coincides with the known asymptotically optimal (in $T$), instance-specific regret [7].

**Guarantees for other optimistic algorithms:** To make the exposition concrete, we only provide regret bounds for the StrongEuler algorithm. However, the "gap-clipping" trick (Proposition 3.1) and subsequent analysis template described in Section 3.1 can be applied to obtain similar bounds for other recent optimistic algorithms, as in [1, 5, 14, 6].[3]

## 2.1 Sub-optimality Gap Lower Bound

Our first lower bound shows that when the total number of rounds $T = KH$ is large, the first term of Corollary 2.1 is unavoidable in terms of regret. Specifically, for every possible choice of gaps, there exists an instance whose regret scales on the order of the first term in (2).

Following standard convention in the literature, the lower bound is stated for algorithms which have sublinear worst case regret. Namely, we say than an algorithm Alg is $\alpha$-*uniformly good* if, for any MDP instance $\mathcal{M}$, there exists a constant $C_{\mathcal{M}} > 0$ such that $\mathbb{E}^{\mathcal{M}}[\mathrm{Regret}_K] \leq C_{\mathcal{M}} K^{\alpha}$ for all $K$.[4]

**Proposition 2.2** (Regret Lower Bound). *Let $S \geq 2$, and $A \geq 2$, and let $\{\Delta_{x,a}\}_{x,a \in [S] \times [A]} \subset (0, H/8)$ denote a set of gaps. Then, for any $H \geq 1$, there exists an MDP $\mathcal{M}$ with states $\mathcal{S} = [S+2]$, actions $\mathcal{A} = [A]$, and $H$ stages, such that,*

$$\mathrm{gap}_1(x,a) = \Delta_{x,a}, \qquad \forall x \in [S], a \in \mathcal{A}$$
$$\mathrm{gap}_h(x,a) \geq 1/2, \qquad \forall x \in \{S+1, S+2\}, a \in \mathcal{A} - \{1\},$$

*and any $\alpha$-uniformly good algorithm satisfies*

$$\lim_{K \to \infty} \frac{\mathbb{E}^{\mathcal{M}}[\mathrm{Regret}_K]}{\log T} \gtrsim (1-\alpha) \sum_{x,a:\mathrm{gap}_1(x,a)>0} \frac{H^2}{\mathrm{gap}_1(x,a)}$$

The above proposition is proven in Appendix H, using a construction based on [4]. For simplicity, we stated an asymptotic lower bound. We remark that if the constant $C_{\mathcal{M}}$ is $\mathrm{poly}(|\mathcal{S}|, |\mathcal{A}|, H)$, then one can show that the above asymptotic bound holds as soon as $K \geq (|\mathcal{S}||\mathcal{A}|H/\mathrm{gap}_*)^{\mathcal{O}(1/(1-\alpha))}$, where $\mathrm{gap}_* := \{\min \mathrm{gap}_1(x,a) : \mathrm{gap}_1(x,a) > 0\}$. More refined non-asymptotic regret bounds can be obtained by following [7].

## 2.2 Why the dependence on $\mathrm{gap}_{\min}$?

ithout the second term, Corollary 2.1 would only suffer one factor of $1/\mathrm{gap}_{\min}$ due to the sum over state-actions pairs $(x,a) \in \mathcal{Z}_{\mathrm{sub}}$ (when the minimum is achieved by a single pair). However, as remarked above, $|\mathcal{Z}_{\mathrm{opt}}|$ typically scales like $|\mathcal{S}|$ and therefore the second term scales like $|\mathcal{S}|/\mathrm{gap}_{\min}$, with a dependence on $1/\mathrm{gap}_{\min}$ that is at least a factor of $|\mathcal{S}|$ more than we would expect. Here, we show that $|\mathcal{S}|/\mathrm{gap}_{\min}$ is unavoidable for the sorts of optimistic algorithms that we typically see in the literature; a rigorous proof is deferred to Appendix G.

**Theorem 2.3** (Informal Lower Bound). *Fix $\delta \in (0, 1/8)$. For universal constants $c_1, c_2, c_3, c_4$, if $\epsilon \in (0, c_1)$, and $S$ satisfies $c_2 \log(\epsilon^{-1}/\delta) \leq S \leq c_3 \epsilon^{-1}/\log(\epsilon^{-1}/\delta)$, there exists an MDP with $|\mathcal{S}| = S$, $|\mathcal{A}| = 2$ and horizon $H = 2$, such that exactly one state has a sub-optimality gap of $\mathrm{gap}_{\min} = \epsilon$ and all other states have a minimum sub-optimality gap $\mathrm{gap}_h(x,a) \geq 1/2$. For this MDP, $\sum_{h,x,a:\mathrm{gap}_h(x,a)>0} \frac{1}{\mathrm{gap}_h(x,a)} \lesssim S + \frac{1}{\mathrm{gap}_{\min}}$ but all existing optimistic algorithms for finite-horizon MDPs which are $\delta$-correct suffer a regret of at least $\frac{S}{\mathrm{gap}_{\min}} \log(1/\delta) \gtrsim \sum_{h,x,a:\mathrm{gap}_h(x,a)>0} \frac{\log(1/\delta)}{\mathrm{gap}_h(x,a)} + \frac{S \log(1/\delta)}{\mathrm{gap}_{\min}}$ with probability at least $1 - c_4 \delta$.*

The particular instance described in Appendix G that witnesses this lower bound is instructive because it demonstrates a case where optimism results in *over*-exploration.

## 2.3 Interpolating with Minimax Regret for Small $T$

We remark that while the logarithmic regret in Corollary 2.1 is non-asymptotic, the expression can be loose for a number of rounds $T$ that is small relative to the sum of the inverse gaps. Our more general result interpolates between the $\log T$ gap-dependent and $\sqrt{T}$ gap-independent regret regimes.

**Theorem 2.4** (Main Regret Bound for StrongEuler). *Fix $\delta \in (0, 1/2)$, and let $A = |\mathcal{A}|$, $S = |\mathcal{S}|$, $M = (SAH)^2$. Futher, define for all $\epsilon > 0$ the set $\mathcal{Z}_{\mathrm{sub}}(\epsilon) := \{(x,a) \in \mathcal{Z}_{\mathrm{sub}} : \mathrm{gap}(x,a) < \epsilon\}$. Then with probability at least $1 - \delta$, StrongEuler run with confidence parameter $\delta$ enjoys the following regret bound for all $K \geq 2$:*

$$\mathrm{Regret}_K \lesssim \min_{\epsilon > 0} \left\{ \sqrt{|\mathcal{Z}_{\mathrm{sub}}(\epsilon)| H \, T(\log T) \log \frac{MT}{\delta}} + \sum_{(x,a) \in \mathcal{Z}_{\mathrm{sub}} \setminus \mathcal{Z}_{\mathrm{sub}}(\epsilon)} \frac{H^3}{\mathrm{gap}(x,a)} \log \left( \frac{MT}{\delta} \right) \right\}$$

$$+ \min \left\{ \sqrt{|\mathcal{Z}_{\mathrm{opt}}| H \, T(\log T) \log \frac{MT}{\delta}}, \; |\mathcal{Z}_{\mathrm{opt}}| \frac{H^3}{\mathrm{gap}_{\min}} \log \left( \frac{MT}{\delta} \right) \right\}$$

$$+ H^4 SA(S \vee H) \min \log \frac{MT}{\delta} \left\{ \log \frac{MT}{\delta}, \log \frac{MH}{\mathrm{gap}_{\min}} \right\}$$

$$\lesssim \sqrt{HSAT \log(T) \log(\frac{MT}{\delta})} + H^4 SA(S \vee H) \log^2 \frac{TM}{\delta},$$

*where the second inequality follows from the first with $\max\{\max_\epsilon |\mathcal{Z}_{\mathrm{sub}}(\epsilon)|, |\mathcal{Z}_{\mathrm{opt}}|\} \leq SA$. Moreover, if $\mathcal{M}$ is an instance of contextual bandits, then the factors of $H$ under the square roots can be refined to a 1, and if $\mathcal{M}$ has $\lesssim 1$-bounded rewards, then these same factors of $H$ can be replaced by a $1/H$. In both settings, logarithmic terms can be refined as in Corollary 2.1.*

By the same argument as above, Theorem 2.4 with $\delta = 1/T$ implies an expected regret scaling like gap-dependent $\log T$ or worst-case $\sqrt{HSAT}$. In Appendix B.1, we state a more refined bound given in terms of the reward bound $\mathcal{G}$, and the maximal variance of any state-action pair (Theorem B.2).

## 3  Gap-Dependent bounds via 'clipping'

In this section, we (i) introduce the key properties of optimistic algorithms, (ii) explain existing approaches to the analysis of such algorithms, and (iii) introduce the "clipping trick", and sketch how this technique yields gap-dependent, non-asymptotic bounds.

**Definition 3.1** (Optimistic Surplus). Given an optimistic algorithm Alg, we define the (optimistic) *surplus* $\mathbf{E}_{k,h}(x,a) := \overline{\mathbf{Q}}_{k,h}(x,a) - r(x,a) - p(x,a)^{\top} \overline{\mathbf{V}}_{k,h+1}$. Alg is *strongly optimistic* if $\mathbf{E}_{k,h}(x,a) \geq 0$ for all $k \geq 1$, and $(x,a,h) \in \mathcal{S} \times \mathcal{A} \times [H]$, which implies that Alg is also optimistic.

While the nomenclature "suplus" is unique to our work, surplus-like terms arise in many prior regret analyses [5, 14]. The notion of *strong optimism* is novel to this work, and facilitates a sharper $H$-dependence in contextual bandit setting of Definition 2.2; intuitively, strong optimism means that the Q-function $\overline{\mathbf{Q}}_{k,h}$ at stage $h$ over-estimates $\mathbf{Q}_h^{\star}$ more than $\overline{\mathbf{Q}}_{k,h+1}$ does $\mathbf{Q}_{k,h+1}^{\star}$.

**The Regret Decomposition for Optimistic Algorithms:** Under optimism alone, we can see that for any $h$ and any $a^{\star} \in \pi^{\star}(x)$,

$$\overline{\mathbf{V}}_{k,h}(x) = \max_a \overline{\mathbf{Q}}_{k,h}(x,a) \geq \overline{\mathbf{Q}}_{k,h}(x,a^{\star}) \geq \mathbf{Q}_h^{\star}(x,a^{\star}) = \mathbf{V}_h^{\star}(x),$$

and therefore, we can bound the sub-optimality of $\pi_k$ as $\mathbf{V}_0^{\star} - \mathbf{V}_0^{\pi_k} \leq \overline{\mathbf{V}}_{k,0} - \mathbf{V}_0^{\pi_k}$.

We can decompose the regret further by introducing the following notation: we let $\boldsymbol{\omega}_{k,h}(x,a) := \mathbb{P}^{\pi_k}[(x_h, a_h) = (x,a)]$ denote the probability of visiting $x$ and playing $a$ at time $h$ in episode $k$. A standard regret decomposition (see e.g. Lemma E.15 [5]) then shows that for a trajectory $(x_h, a_h)_{h=1}^{H}$, $\overline{\mathbf{V}}_{k,0} - \mathbf{V}_0^{\pi_k} = \mathbb{E}^{\pi_k}[\sum_{h=1}^{H} \mathbf{E}_{k,h}(x_h, a_h)] = \sum_{h=1}^{H} \sum_{x,a} \boldsymbol{\omega}_{k,h}(x,a) \mathbf{E}_{k,h}(x,a)$, yielding a regret bound of

$$\sum_{k=1}^{K} \mathbf{V}_0^{\star} - \mathbf{V}_0^{\pi_k} \leq \sum_{k=1}^{K} \overline{\mathbf{V}}_{k,0} - \mathbf{V}_0^{\pi_k} \leq \sum_{k=1}^{K} \sum_{h=1}^{H} \sum_{x,a} \boldsymbol{\omega}_{k,h}(x,a) \mathbf{E}_{k,h}(x,a).$$

**Existing Analysis of MDPs:** We begin by sketching the flavor of minimax analyses. Introducing the notation $n_k(x,a) := \{\#\text{times } (x,a) \text{ is visited before episode } k\}$, existing analyses carefully manipulate the surpluses $\mathbf{E}_{k,h}(x,a)$ to show that $\sum_{h=1}^{H} \sum_{x,a} \boldsymbol{\omega}_{k,h}(x,a) \mathbf{E}_{k,h}(x,a) \lesssim \sum_{h=1}^{H} \sum_{x,a} \boldsymbol{\omega}_{k,h}(x,a) \frac{\overline{C}_{\mathcal{M}}}{\sqrt{n_k(x,a)}}$ + lower order terms, where typically $\overline{C}_{\mathcal{M}} = \text{poly}(H, \log(T/\delta))$. Finally, they replace $n_k(x,a)$ with an "idealized analogue", $\overline{n}_k(x,a) := \sum_{j=1}^{k} \sum_{h=1}^{H} \boldsymbol{\omega}_{j,h}(x,a) := \sum_{j=1}^{k} \boldsymbol{\omega}_j(x,a)$, where we introduce $\boldsymbol{\omega}_j(x,a) := \sum_{h=1}^{H} \boldsymbol{\omega}_{j,h}(x,a)$ denote the expected number of visits of $(x,a)$ at episode $j$. Letting $\{\mathcal{F}_k\}$ denote the filtration capturing all events up to the end episode $k$, we see that $\mathbb{E}[\overline{n}_k(x,a) - \overline{n}_{k-1}(x,a)|\mathcal{F}_{k-1}] = \boldsymbol{\omega}_k(x,a)$, and thus by standard concentration arguments (see Lemma B.7, or Lemma 6 in [6]), $\overline{n}_k(x,a)$ and $n_k(x,a)$ are within a constant factor of each other for all $k$ such that $\overline{n}_k(x,a)$ is sufficiently large. Hence, by replacing $n_k(x,a)$ with $\overline{n}_k(x,a)$, we have (up to lower order terms)

$$\sum_{k=1}^{K} \mathbf{V}_0^{\star} - \mathbf{V}_0^{\pi_k} \lesssim \sum_{x,a} \sum_{k=1}^{K} \boldsymbol{\omega}_k(x,a) \frac{\overline{C}_{\mathcal{M}}}{\sqrt{\overline{n}_k(x,a)}} + \text{ lower order terms.} \qquad (3)$$

A $\sqrt{SAK \, \text{poly}(H)}$ bound is typically concluded using a careful application of Cauchy-Schwartz, and an integration-type lemma (e.g., Lemma C.1). An analysis of this flavor is used in Appendix B.4.

On the other hand, one can *exactly* establish the identity $\mathbf{V}_0^{\star} - \mathbf{V}_0^{\pi_k} = \sum_{x,a} \sum_{h=1}^{H} \boldsymbol{\omega}_{k,h}(x,a) \text{gap}_h(x,a)$. Then one can achieve a gap dependent bound as soon as one can show that the algorithm ceases to select suboptimal actions $a$ at $(x,h)$ after sufficiently large $T$. Crucially, determining if action $a$ is (sub)optimal at $(x,h)$ requires precise knowledge about the value function at other states in the MDP at future stages $h' > h$. This difficulty is why previous gap-dependent analyses appeal to diameter or ergodicity assumptions, which ensure sufficient uniform exploration of the MDP to reason about the value function at subsequent stages.

## 3.1 The Clipping Trick

We now introduce the "clipping trick", a technique which merges both the minimax analysis in terms of the surpluses $\mathbf{E}_{k,h}(x,a)$, and the gap-dependent strategy, which attempts to control how many times a given suboptimal action is selected. Core to our analysis, define the *clipping* operator

$$\mathrm{clip}\left[x \mid \epsilon\right] = x\mathbb{I}\{x \geq \epsilon\},$$

for all $x, \epsilon > 0$. We can now state our first main technical result, which states that the sub-optimality $\mathbf{V}_0^\star - \mathbf{V}_0^{\pi_k}$ can be controlled by a sum over surpluses which have been *clipped* to zero whenever they are sufficiently small.

**Proposition 3.1.** *Let* $\mathrm{g\breve{a}p}_h(x,a) := \frac{\mathrm{gap}_{\min}}{2H} \vee \frac{\mathrm{gap}_h(x,a)}{4H}$. *Then, if* $\pi_k$ *is induced by an optimistic algorithm with surpluses* $\mathbf{E}_{k,h}(x,a)$,

$$\mathbf{V}_0^\star - \mathbf{V}_0^{\pi_k} \leq 2e \sum_{h=1}^H \sum_{x,a} \boldsymbol{\omega}_{k,h}(x,a)\,\mathrm{clip}\left[\mathbf{E}_{k,h}(x,a) \mid \mathrm{g\breve{a}p}_h(x,a)\right].$$

*If the algorithm is* strongly optimistic*, and $\mathcal{M}$ is a contextual bandits instance, we can replace* $\mathrm{g\breve{a}p}_h(x,a)$ *with* $\mathrm{g\breve{a}p}_h(x,a) := \frac{\mathrm{gap}_{\min}}{2H} \vee \frac{\mathrm{gap}_h(x,a)}{4}$.

The above proposition is a consequence of a more general bound, Theorem B.3, given in Appendix B. Unlike gap-dependent bounds that appeal to hitting-time arguments, we *do not* reason about when a suboptimal action $a$ will cease to be taken. Indeed, an algorithm may still choose a suboptimal action $a$ *even if* the surplus $\mathbf{E}_{k,h}(x,a)$ is small, because *future* surpluses may be large. Instead, we argue in two parts:

1. A sub-optimal action $a \notin \pi_h^\star(x)$ is taken only if $\overline{\mathbf{Q}}_{k,h}(x,a) \geq \mathbf{Q}_h^\star(x,a^\star)$ for some $a^\star \in \pi_h^\star(x)$, or equivalently in terms of the surplus, only if $\mathbf{E}_{k,h}(x,a)+p(x,a)^\top(\overline{\mathbf{V}}_{k,h+1}-\mathbf{V}_{k,h+1}^\star) > \mathrm{gap}_h(x,a)$. Thus if Alg selects a suboptimal action, then this is because either the *current* surplus $\mathbf{E}_{k,h}(x,a)$ is larger than $\Omega(\frac{\mathrm{gap}_h(x,a)}{H})$, or the expectation over *future* surpluses, captured by $p(x,a)^\top(\overline{\mathbf{V}}_{k,h+1}-\mathbf{V}_{k,h+1}^\star)$ is larger than $(1-\mathcal{O}\left(\frac{1}{H}\right))\mathrm{gap}_h(x,a)$. Intuitively, the first case occurs when $(x,a)$ has not been visited enough, and the second when the future state/action pairs have not experienced sufficient visitation. In the first case, we can clip the surplus at $\Omega(\frac{\mathrm{gap}_h(x,a)}{H})$; in the second, $\mathbf{E}_{k,h}(x,a) + p(x,a)^\top(\overline{\mathbf{V}}_{k,h+1} - \mathbf{V}_{k,h+1}^\star) \leq (1 + \mathcal{O}\left(\frac{1}{H}\right))p(x,a)^\top(\overline{\mathbf{V}}_{k,h+1} - \mathbf{V}_{k,h+1}^\star)$, and push the the contribution of $\mathbf{E}_{k,h}(x,a)$ into the contribution of future surpluses. This incurs a factor of at most $(1 + \mathcal{O}\left(\frac{1}{H}\right))^H \lesssim 1$, avoiding an exponential dependence on $H$.

2. Clipping surpluses for pairs $(x,a)$ for optimal $a \in \pi_h^\star(x)$ requires more care. We introduce "half-clipped" surpluses $\ddot{\mathbf{E}}_{k,h}(x,a) := \mathrm{clip}\left[\mathbf{E}_{k,h}(x,a) \mid \frac{\mathrm{gap}_{\min}}{2H}\right]$ where *all* actions are clipped at $\mathrm{gap}_{\min}/2H$, and recursively define value functions $\ddot{\mathbf{V}}_h^{\pi_k}(\cdot)$ corresponding to these clipped surpluses (see Definition D.1). We then show that, for $\ddot{\mathbf{V}}_0^{\pi_k} := \mathbb{E}_{x \sim p_0}\left[\ddot{\mathbf{V}}_1(x)\right]$, we have (Lemma D.2)

$$\mathbf{V}_0^\star - \mathbf{V}_0^{\pi_k} \leq 2(\ddot{\mathbf{V}}_0^{\pi_k} - \mathbf{V}_0^{\pi_k}).$$

This argument is based on carefully analyzing when $\pi_{k,h}$ first recommends a suboptimal action $\pi_{k,h}(x) \notin \pi^\star(x)$, and showing that when this occurs, $\mathbf{V}_0^\star - \mathbf{V}_0^{\pi_k}$ is roughly lower bounded by $\frac{\mathrm{gap}_{\min}}{H}$ times the probability of visiting a state $x$ where $\pi_{k,h}(x)$ plays suboptimally. We can then subtract off $\frac{\mathrm{gap}_{\min}}{2H}$ from all the surplus terms at the expense of at most halving the suboptimality, and using the fact $\mathbf{E}_{k,h} - \frac{\mathrm{gap}_{\min}}{2H} \leq \mathrm{clip}\left[\mathbf{E}_{k,h} \mid \frac{\mathrm{gap}_{\min}}{2H}\right]$ concludes the bound. This step is crucial, because it allows us to clip the surpluses even at pairs $(x,a)$ where $a \in \pi_h^\star(x)$ is the optimal action. We note that in the formal proof of Proposition 3.1, this half-clipping precedes the clipping of suboptimal actions described above.

Unfortunately, the first step involving the half-clipping is rather coarse, and leads to $S/\mathrm{gap}_{\min}$ term in the final regret bound. As argued in Theorem 2.3, this is unavoidable for existing optimistic algorithms, and suggests that Proposition 3.1 cannot be significantly improved in general.

## 3.2 Analysis of StrongEuler

Recall that StrongEuler is precisely described by Definition 1.1 up to our particular choice of confidence intervals defined (see Algorithm 1 in Appendix E). We now state a surplus bound (proved in Appendix F) that holds for these particular choice of confidence intervals, and which ensures that the strong optimism criterion of Definition 1.1 is satisfied:

**Proposition 3.2** (Surplus Bound for Strong Euler (Informal)). *Let $M = SAH$, and define the variances $\mathtt{Var}_{h,x,a}^{\star} := \mathrm{Var}[R(x,a)] + \mathrm{Var}_{x' \sim p(x,a)}[\mathbf{V}_{h+1}^{\star}(x')]$. Then, with probability at least $1 - \delta/2$, the following holds for all $(x,a) \in \mathcal{S} \times \mathcal{A}$, $h \in [H]$ and $k \geq 1$,*

$$0 \leq \mathbf{E}_{k,h}(x,a) \lesssim \underbrace{\sqrt{\frac{\mathtt{Var}_{h,x,a}^{\star} \log(M n_k(x,a)/\delta)}{n_k(x,a)}}}_{\mathbf{B}_{k,h}^{\mathrm{lead}}(x,a)} + \textit{lower order terms}.$$

We emphasize that Proposition 3.2, and its formal analogue Proposition B.4 in Appendix B.2, are the *only* part of the analysis that relies upon the particular form of the StrongEuler confidence intervals; to analyze other model-based optimistic algorithms, one would simply establish an analogue of this proposition, and continue the analysis in much the same fashion. While Q-learning [9] also satisfies optimism, it induces a more intricate surplus structure, which may require a different analysis.

Recalling the clipping from Proposition 3.1, we begin the gap-dependent bound with $\sum_{k=1}^{K} \mathbf{V}_0^{\star} - \mathbf{V}_0^{\pi_k} \lesssim \sum_{x,a,k,h} \boldsymbol{\omega}_{k,h}(x,a) \operatorname{clip}\left[\mathbf{E}_{k,h}(x,a) \,|\, \breve{\mathtt{gap}}_h(x,a)\right]$. Neglecting lower order terms, Proposition 3.2 ensures that this is approximately less than $\sum_{x,a,k,h} \boldsymbol{\omega}_{k,h}(x,a) \operatorname{clip}\left[\mathbf{B}_{k,h}^{\mathrm{lead}}(x,a) \,|\, \breve{\mathtt{gap}}_h(x,a)\right]$. Introduce the minimal (over $h$) clipping-gaps $\breve{\mathtt{gap}}(x,a) := \min_h \breve{\mathtt{gap}}(x,a) \geq \frac{\mathtt{gap}(x,a) \vee \mathtt{gap}_{\min}}{4H}$ and maximal variances $\mathtt{Var}_{x,a}^{\star} := \max_h \mathtt{Var}_{h,x,a}^{\star}$. We can then render $\mathbf{B}_{k,h}^{\mathrm{lead}}(x,a) \leq f(n_k(x,a))$, where $f(u) \lesssim \operatorname{clip}\left[\sqrt{\frac{1}{u}\mathtt{Var}_{x,a}^{\star} \log(Mu/\delta)} \,|\, \breve{\mathtt{gap}}(x,a)\right]$. Recalling the approximation $n_k(x,a) \approx \overline{n}_k(x,a)$ described above, we have, to first order,

$$\sum_{k=1}^{K} \mathbf{V}_0^{\star} - \mathbf{V}_0^{\pi_k} \lesssim \sum_{x,a,k,h} \boldsymbol{\omega}_{k,h}(x,a) \operatorname{clip}\left[\mathbf{B}_{k,h}^{\mathrm{lead}}(x,a) \,|\, \breve{\mathtt{gap}}_h(x,a)\right]$$

$$\lesssim \sum_{x,a,k} \boldsymbol{\omega}_k(x,a) f(n_k(x,a)) \lesssim \sum_{x,a,k} \boldsymbol{\omega}_k(x,a) f(\overline{n}_k(x,a)),$$

where we recall the expected visitations $\boldsymbol{\omega}_k(x,a) := \sum_{h=1}^{H} \boldsymbol{\omega}_{k,h}(x,a)$. Since $\overline{n}_k(x,a) := \sum_{j=1}^{k} \boldsymbol{\omega}_j(x,a)$, we can regard the above as an *integral* of the function $f(u)$ (see Lemma C.1), with respect to the *density* $\boldsymbol{\omega}_k(x,a)$. Evaluating this integral (Lemma B.9) yields (up to lower order terms)

$$\sum_{k=1}^{K} \mathbf{V}_0^{\star} - \mathbf{V}_0^{\pi_k} \lessapprox \sum_{x,a} \frac{H \mathtt{Var}_{x,a}^{\star} \log \frac{MT}{\delta}}{\min_h \breve{\mathtt{gap}}_h(x,a)} \lessapprox \sum_{x,a} \frac{H \mathtt{Var}_{x,a}^{\star} \log \frac{MT}{\delta}}{\mathtt{gap}(x,a) \vee \mathtt{gap}_{\min}}.$$

Finally, bounding $\mathtt{Var}_{x,a}^{\star} \leq H^2$ and splitting the bound into the states $\mathcal{Z}_{\mathtt{sub}} := \{(x,a) : \mathtt{gap}(x,a) > 0\}$ and $\mathcal{Z}_{\mathtt{opt}} := \{(x,a) : \mathtt{gap}(x,a) = 0\}$ recovers the first two terms in Corollary 2.1. In benign instances (Definition 2.2), we can bound $\mathtt{Var}_{h,x,a}^{\star} \lesssim 1$, improving the $H$-dependence. In contextual bandits, we save an addition $H$ factor via $\breve{\mathtt{gap}}_h(x,a) \gtrsim (\mathtt{gap}_{\min}/H) \vee \mathtt{gap}(x,a)$. The interpolation with the minimax rate in Theorem 2.4 is decribed in greater detail in Appendix B.4.

## Footnotes

[1]By this, we mean that for any *fixed* $T \geq 1$, one can attain $C_{\mathcal{M}} \log T$ regret. Extending the bound to anytime regret is left to future work

[2][13] actually presents a bound of the form $\frac{\bar{D}^2 SA}{\min_{(s,a) \in \mathsf{CRIT}} \mathrm{gap}_\infty(x,a)} \log(T)$ but it is straightforward to extract the claimed form from the proof.

[3] To achieve logarithmic regret, some of these algorithms require a minor modification to their confidence intervals; otherwise, the gap-dependent regret scales as $\log^2 T$. See Appendix E for details.

[4] We may assume as well that Alg is allowed to take the number of episodes $K$ as a parameter.

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
