[Supplementary Material]

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

:\mathtt{gap}_h(x,a)>0} \frac{1}{\mathtt{gap}_h(x,a)} \lesssim S + \frac{1}{\mathtt{gap}_{\mathtt{min}}}$ but all existing optimistic algorithms for finite-horizon MDPs which are $\delta$-correct suffer a regret of at least $\frac{S}{\mathtt{gap}_{\mathtt{min}}} \log(1/\delta) \gtrsim \sum_{h,x,a:\mathtt{gap}_h(x,a)>0} \frac{\log(1/\delta)}{\mathtt{gap}_h(x,a)} + \frac{S \log(1/\delta)}{\mathtt{gap}_{\mathtt{min}}}$ with probability at least $1 - c_4\delta$.*

The particular instance described in Appendix G that witnesses this lower bound is instructive because it demonstrates a case where optimism results in *over*-exploration.

## 2.3 Interpolating with Minimax Regret for Small $T$

We remark that while the logarithmic regret in Corollary 2.1 is non-asymptotic, the expression can be loose for a number of rounds $T$ that is small relative to the sum of the inverse gaps. Our more general result interpolates between the $\log T$ gap-dependent and $\sqrt{T}$ gap-independent regret regimes.

**Theorem 2.4** (Main Regret Bound for StrongEuler). *Fix $\delta \in (0, 1/2)$, and let $A = |\mathcal{A}|$, $S = |\mathcal{S}|$, $M = (SAH)^2$. Futher, define for all $\epsilon > 0$ the set $\mathcal{Z}_{\mathtt{sub}}(\epsilon) := \{(x,a) \in \mathcal{Z}_{\mathtt{sub}} : \mathtt{gap}(x,a) < \epsilon\}$. Then with probability at least $1 - \delta$, StrongEuler run with confidence parameter $\delta$ enjoys the following regret bound for all $K \geq 2$:*

$$\mathrm{Regret}_K \lesssim \min_{\epsilon>0} \left\{ \sqrt{|\mathcal{Z}_{\mathtt{sub}}(\epsilon)| H \, T (\log T) \log \tfrac{MT}{\delta}} \right\} + \sum_{(x,a)\in\mathcal{Z}_{\mathtt{sub}} \backslash \mathcal{Z}_{\mathtt{sub}}(\epsilon)} \frac{H^3}{\mathtt{gap}(x,a)} \log\left(\tfrac{MT}{\delta}\right) \right\}$$

$$+ \min\left\{ \sqrt{|\mathcal{Z}_{\mathtt{opt}}| \, H \, T (\log T) \log \tfrac{MT}{\delta}}, \; |\mathcal{Z}_{\mathtt{opt}}| \frac{H^3}{\mathtt{gap}_{\mathtt{min}}} \log\left(\tfrac{

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

}^\star_{x,a} := \max_h \mathtt{Var}^\star_{h,x,a}$. We can then render $\mathbf{B}^{\mathrm{lead}}_{k,h}(x,a) \leq f(n_k(x,a))$, where $f(u) \lesssim \mathrm{clip}\left[\sqrt{\frac{1}{u}\mathtt{Var}^\star_{x,a} \log(Mu/\delta)} \mid \mathtt{g\breve{a}p}(x,a)\right]$. Recalling the approximation $n_k(x,a) \approx \overline{n}_k(x,a)$ described above, we have, to first order,

$$\sum_{k=1}^K \mathbf{V}^\star_0 - \mathbf{V}^{\pi_k}_0 \lesssim \sum_{x,a,k,h} \boldsymbol{\omega}_{k,h}(x,a) \mathrm{clip}\left[\mathbf{B}^{\mathrm{lead}}_{k,h}(x,a) \mid \mathtt{g\breve{a}p}_h(x,a)\right]$$
$$\lesssim \sum_{x,a,k} \boldsymbol{\omega}_k(x,a) f(n_k(x,a)) \lesssim \sum_{x,a,k} \boldsymbol{\omega}_k(x,a) f(\overline{n}_k(x,a)),$$

where we recall the expected visitations $\boldsymbol{\omega}_k(x,a) := \sum_{h=1}^H \boldsymbol{\omega}_{k,h}(x,a)$. Since $\overline{n}_k(x,a) := \sum_{j=1}^k \boldsymbol{\omega}_j(x,a)$, we can regard the above as an *integral* of the function $f(u)$ (see Lemma C.1), with respect to the *density* $\boldsymbol{\omega}_k(x,a)$. Evaluating this integral (Lemma B.9) yields (up to lower order terms)

$$\sum_{k=1}^K \mathbf{V}^\star_0 - \mathbf{V}^{\pi_k}_0 \lessapprox \sum_{x,a} \frac{H\mathtt{Var}^\star_{x,a} \log \frac{MT}{\delta}}{\min_h \mathtt{g\breve{a}p}_h(x,a)} \lessapprox \sum_{x,a} \frac{H\mathtt{Var}^\star_{x,a} \log \frac{MT}{\delta}}{\mathtt{gap}(x,a) \vee \mathtt{gap}_{\min}}.$$

Finally, bounding $\mathtt{Var}^\star_{x,a} \leq H^2$ and splitting the bound into the states $\mathcal{Z}_{\mathtt{sub}} := \{(x,a) : \mathtt{gap}(x,a) > 0\}$ and $\mathcal{Z}_{\mathtt{opt}} := \{(x,a) : \mathtt{gap}(x,a) = 0\}$ recovers the first two terms in Corollary 2.1. In benign instances (Definition 2.2), we can bound $\mathtt{Var}^\star_{h,x,a} \lesssim 1$, improving the $H$-dependence. In contextual bandits, we save an addition $H$ factor via $\mathtt{g\breve{a}p}_h(x,a) \gtrsim (\mathtt{gap}_{\min}/H) \vee \mathtt{gap}(x,a)$. The interpolation with the minimax rate in Theorem 2.4 is decribed in greater detail in Appendix B.4.

## Footnotes

[1]By this, we mean that for any *fixed* $T \geq 1$, one can attain $C_{\mathcal{M}} \log T$ regret. Extending the bound to anytime regret is left to future work

[2][15] actually presents a bound of the form $\frac{\bar{D}^2 SA}{\min_{(s,a) \in \mathsf{CRIT}} \mathrm{gap}_\infty(x,a)} \log(T)$ but it is straightforward to extract the claimed form from the proof.

[3] To achieve logarithmic regret, some of these algorithms require a minor modification to their confidence intervals; otherwise, the gap-dependent regret scales as $\log^2 T$. See Appendix E for details.

[4] We may assume as well that Alg is allowed to take the number of episodes $K$ as a parameter.

[5]The condition can be relaxed somewhat to only needing to hold for a set $\mathcal{S}$ for which $p(x' \in \mathcal{S} \mid x, a)$ is close to 1; for simplicity, consider the unrelaxed notion as defined as above.

[6]Note that we induce $\overline{n}_k(x,a)$ to include a sum up to index $k$; this makes the following arguments more convenient, and will only accrue constant factors in the analysis

[7]The quantity $\mathrm{Var}[R(x,a)]$ below can also be replaced with an empirical variance, but we choose the true variance for simplicity.

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

# Contents

# A    Notation and Organization

**Organization:**  This section describes the organization of the appendix, and clarifying our notation. The remainder of the appendix is divided into three parts:

Part I presents more detailed statements of the regret upper bounds obtained by StrongEuler, and their complete proofs. Section B.2 introduces Corollary B.1 and Theorem B.2, refining Corollary 2.1 and Theorem 2.4, from the main text. The section continues to prove both results. In addition, we introduce Theorem B.3, which refines the clipped regret decomposition Proposition 3.1. The proofs in this section rely on numerous technical lemmas, whose proofs are defered to Section C. Finally, this section states Proposition B.4, which ensures that StrongEuler is optimistic and provides a precise bound on the surpluses $\mathbf{E}_{k,h}(x, a)$, described informally in Proposition 3.2.

In Part II, we present the StrongEuler algorithm and its guarantees. Section E describes how StrongEuler instantiates the *model-based* examples of optimistic algorithms described in Section 1.3; the algorithm and choice of confidence bonuses are specified in pseudocode. In Section F, we prove the surplus bound Proposition B.4, and verify that StrongEuler is strongly optimistic

Lastly, Part III contains the proofs of our lower bounds. Section G proves the $\Omega(S/\mathtt{gap}_{\min})$ lower bound described in Theorem 2.3, and rigorously describes the class of algorithms to which it applies. Finally, Section H proves the information theoretic lower bound, Proposition 2.2.

**Notational Rationale:**  Unfortunately, the regret analysis of tabular MDPs requires significant notational overhead. Here we take a moment to highlight some notational conventions that we shall use throughout. The superscript $(\cdot)^\star$ denotes "optimal" quantities, i.e. the optimal policy $\pi^\star$, the optimal value $\mathbf{V}^\star$, and variances of the optimal policy $\mathtt{Var}^\star_{h,x,a}$. The accents $\overline{(\cdot)}$ will be used to denote upper bounds on quantities, e.q. an optimistic Q-function $\overline{\mathbf{Q}}_{k,h}$ is an upper bound on $\mathbf{Q}^\star_{k,h}$, and $\overline{\mathtt{Var}}$ is an upper bound on the variance, and so on. $\underline{(\cdot)}$ will denote lower bounds on quantities. For example, StrongEuler will maintain lower bounds on the values $\underline{\mathbf{V}}_{k,h} \leq \mathbf{V}^\star$. The accent $\check{(\cdot)}$ will pertain to clipped quantities; e.g. $\check{\mathtt{gap}}$ is the gap-value at which surpluses are clipped. Many quantities, like $\mathtt{gap}_h(x, a)$ (gaps) and $\mathtt{Var}^\star_{h,x,a}$ (variances) depend on the triples $(x, a, h)$. The quantities $\mathtt{gap}(x, a)$ and $\mathtt{Var}^\star_{x,a}$ with $h$ suppresed to denote worse-case bounds on these term over $h \in [H]$; e.g. $\mathtt{gap}(x, a) := \min_{h \in [H]} \mathtt{gap}_h(x, a)$ and $\mathtt{Var}^\star_{x,a} := \max_{h \in [H]} \mathtt{Var}^\star_{h,x,a}$.

**Policies, Value Functions, Q-functions**

$\pi = (\pi_h)_{h=1}^H$ denotes a policy with $\pi_h : \mathcal{S} \to \mathcal{A}$

$\mathbf{V}_0^\pi$ denotes the value of $\pi$

$\mathbf{V}_h^\pi(x)$ denotes the value of $\pi$ at $h \in [H]$ and $x \in \mathcal{S}$

$\mathbf{Q}_h^\pi(x, a)$ denotes Q-function of $\pi$

$\mathbf{V}_0^\star, \mathbf{V}_h^\star(x), \mathbf{Q}^\star(x, a)$ denote optimal value, value function, Q-function

$\pi_h^\star(x)$ denotes the set of optimal actions at $h \in [H]$, $x \in \mathcal{S}$.

$\overline{\mathbf{V}}_{k,h}(x) / \overline{\mathbf{Q}}_{k,h}(x, a)$ denotes optimistic value/Q function

$\pi_{k,h}(x) = \arg\max_a \overline{\mathbf{Q}}_{k,h}(x, a)$ denotes optimistic policy

---

**Problem Dependent Quantities**

$\mathrm{gap}_h(x, a) := \mathbf{V}_h^\star(x) - \mathbf{Q}_h^\star(x, a)$.

$\mathrm{gap}(x, a) := \min_h \mathrm{gap}_h(x, a)$

$\mathrm{gap}_{\min} := \min_{x,a}\{\mathrm{gap}_h(x, a) : \mathrm{gap}_h(x, a) > 0\}$

$\boldsymbol{\alpha}_{x,a,h} \in [0, 1]$ denotes transition suboptimality (Definition B.3)

$\mathrm{Var}_{h,x,a}^\star := \mathrm{Var}[R(x, a)] + \mathrm{Var}_{x' \sim p(x,a)}[\mathbf{V}_{h+1}^\star(x)]$

$\mathrm{Var}_{x,a}^\star := \max_h \mathrm{Var}_{x,a}^\star$

$\overline{\mathrm{Var}} := \max_{x,a,h} \mathrm{Var}_{h,x,a}^\star$.

$\mathcal{G} \le H$: upper bound on $\sum_{h=1}^H R(x, \pi_h(x))$ (Definition 2.2)

$\overline{H}_T := \min\{\overline{\mathrm{Var}}, \mathcal{G}^2/H\}$

---

**Quantities for Analysis**

$\mathbf{L}(u) := \sqrt{2 \log(10 M^2 \max\{u, 1\}^2/\delta)}$

$\boldsymbol{\omega}_{k,h}(x, a) := \mathbb{P}^{\pi_k}[(x_h, a_h) = (x, a)]$ denotes the surplus

$\boldsymbol{\omega}_k(x, a) := \sum_{h=1}^H \boldsymbol{\omega}_{k,h}(x, a)$ denotes the surplus

$n_k(x, a)$ denotes the number of times $(x, a)$ is observed up to time $k - 1$

$\overline{n}_k(x, a) := \sum_{t=1}^k \boldsymbol{\omega}_t(x, a)$.

$\boldsymbol{\tau}(x, a)$ denotes time after which $\overline{n}_k(x, a)$ is sufficiently large

$\mathcal{A}^{\mathrm{conc}}$ (good concentration event)

$\mathcal{E}^{\mathrm{samp}}$ (good sampling event, Lemma B.7)

$H_{\mathrm{sample}} \lesssim H \log \frac{M}{\delta}$ number of sampes for $\mathcal{E}^{\mathrm{samp}}$ to apply

$\mathrm{g\breve{a}p}_h(x, a) := \frac{\mathrm{gap}_{\min}}{2H} \vee \frac{\mathrm{gap}_h(x,a)}{4(H\boldsymbol{\alpha}_{x,a,h} \vee 1)}$ (clipped gap)

$\mathrm{g\breve{a}p}_{\min} := \min_{x,a,h} \mathrm{g\breve{a}p}_h(x, a)$ (clipped gap)

$\mathrm{Var}_{h,x,a}^\pi := \mathrm{Var}[R(x, a)] + \mathrm{Var}_{x' \sim p(x,a)}[\mathbf{V}_{h+1}^\pi(x')]$

$\mathrm{Var}_{h,x,a}^{(k)} = \min\left\{\mathrm{Var}_{h,x,a}^\star, \mathrm{Var}_{h,x,a}^{\pi_k}\right\}$

$\mathbf{E}_{k,h}(x, a) := \overline{\mathbf{Q}}_{k,h}(x, a) - r(x, a) - p(x, a)^\top \overline{\mathbf{V}}_{k,h+1}$ denotes the surplus

$\mathbf{E}_{k,h}(x, a) \lesssim \mathbf{B}_{k,h}^{\mathrm{lead}}(x, a) + \mathbb{E}^{\pi_k}\left[\sum_{t=h}^H \mathbf{B}_k^{\mathrm{fut}}(x_t, a_t) \mid (x_h, a_h) = (x, a)\right]$ (surplus bound, Proposition B.4)

$\mathbf{B}_{k,h}^{\mathrm{lead}}(x, a) := H \wedge \sqrt{\frac{\mathrm{Var}_{h,x,a}^{(k)} \log\left(\frac{M n_k(x,a)}{\delta}\right)}{n_k(x, a)}}$ (lead bound on surplus)

$\mathbf{B}_k^{\mathrm{fut}}(x, a) = H^3 \wedge H^3 \left(\sqrt{\frac{S \log\left(\frac{M n_k(x,a)}{\delta}\right)}{n_k(x, a)}} + \frac{S \log\left(\frac{M n_k(x,a)}{\delta}\right)}{n_k(x, a)}\right)^2$ (bound on future surpluses)

# Part I

# Precise Results and Analysis

## B  Precise Statement and Rigorous Proof Sketch of Main Regret Bounds

In this section, we present a precise statements and formal proofs of the upper bounds, Corollary 2.1 and Theorem 2.4, from the main text. These bounds both makes the improvements in the benign instances of Definition 2.2, and takes advantage of other possibly-favorable instance-specific quantities. The remainder of the section is organized as follows. In Section B.1, we introduce the relevant problem-dependent quantitites in terms of which we state our more refined bounds. We then state Corollary B.1, a more precise analogue of the $\log$-regret bounds in Corollary 2.1, followed by Theorem B.2, which refines the regret bound Theorem 2.4 in interpolating between the $\log T$ and $\sqrt{T}$ regimes.

Next, in Section B.2, we set up the preliminaries for the proof of our upper bound, including (a) Theorem B.3, the granular clipping bound strengthening Proposition 3.1, (b) Proposition B.4, which upper bounds the surpluses $\mathbf{E}_{k,h}(x, a)$ for StrongEuler, and (c) Lemma B.6 which combines the two into a useful form.

Then, in Section B.2, we present a rigorous proof of Corollary B.1 based on integration tools developed in Section C. Finally, we modify the arguments slightly to obtain the interpolation in Theorem B.2. The proof of Proposition 3.1 is given in Section F, Theorem B.3 is given in Section D, and the remainder of technical results in the present section are established in Section C.

We emphasize that the tools in this section provide a general recipe for establishing similar regret bounds for the existing model-based optimistic algorithms in the literature. We have attempted to present our tools in a modular fashion in hope that they can be borrowed to automate the proofs of similar guarantees in related settings.

### B.1  More Precise Statement of Regret Bound Theorem 2.4

We shall begin by stating a more precise version of Theorem 2.4, Following [16], we begin by defnining the variances of the value optimal functions::

**Definition B.1** (Variance Terms). We define the variance of a triple $(x, a, h)$ as

$$\mathrm{Var}^{\star}_{h,x,a} := \mathrm{Var}[R(x,a)] + \mathrm{Var}_{x' \sim p(x,a)}[\mathbf{V}^{\star}_{h+1}(x')],$$

and the statewise maximal variances as $\mathrm{Var}^{\star}_{x,a} := \max_h \mathrm{Var}^{\star}_{h,x,a}$, and the maximal variance as $\overline{\mathrm{Var}} := \max_{x,a,h} \mathrm{Var}^{\star}_{h,x,a}$.

**Remark B.1** (Typical Bounds on the variance ). While $\overline{\mathrm{Var}} \leq H^2$ for general MDPs (see e.g. [1]), we have $\overline{\mathrm{Var}}$ is smaller for the benign instances in Definition 2.2. We briefly summarize this discussion from [16]: If $\mathcal{M}$ has $\mathcal{G}$ bounded rewards, then $\mathbf{V}^{\star}_{h+1}(x) \leq \mathcal{G}$ for any $x$, and thus $\mathrm{Var}^{\star}_{h,x,a} \leq 1 + \mathcal{G}^2$, which is $\lesssim 1$ if $\mathcal{G} \lesssim 1$. For contextual bandits, $p = p(x, a)$ does not depend on $x, a$, and $\mathbf{V}^{\star}_{h+1}(x) = (\max_a R(x, a)) + (\mathbb{E}_{x' \sim p} \mathbf{V}^{\star}_{h+2}(x'))$, where the second term does not dependent Hence, $\mathrm{Var}_{x' \sim p}[\mathbf{V}^{\star}_{h+1}(x')] \leq \mathrm{Var}[(\max_a R(x, a))] \leq 1$, and thus $\mathrm{Var}^{\star}_{h,x,a} \leq 2$.

We can then define an associated "effective horizon", which replaces $H$ with a possibly smaller problem dependent quantity:

**Definition B.2** (Effective Horizon). Suppose that $\mathcal{M}$ has $\mathcal{G}$-bounded rewards, as in Definition 2.2) We define the *effective horizon* as

$$\overline{H}_T := \min \left\{ \overline{\mathrm{Var}}, \frac{\mathcal{G}^2}{H} \log T \right\} .$$

Since any horizon-$H$ MDP has $H$-bounded rewards, $\overline{H}_T$ always satisfies $\overline{H}_T \leq \min \left\{ H^2, H \log T \right\}$.

We note that the bound $\overline{\mathrm{Var}} \lesssim 1$ for contextual bandits implies $\overline{H}_T \lesssim 1$, whereas if $\mathcal{M}$ has $\mathcal{G}$-bounded rewards with $\mathcal{G} \lesssim 1$, $\overline{H}_T \lesssim 1 \wedge \frac{1}{H} \log T$.

Lastly, we shall introduce one more condition we call *transition suboptimality*, which is a notion of distributional closeness that enables the improved clipping and sharper regret bounds for the special case of contextual bandits (Definition 2.2):

**Definition B.3** (Transition Sub-optimality). *Given $\boldsymbol{\alpha} \in [0, 1]$, we say that a tuple $(x, a, h)$ is $\boldsymbol{\alpha}$-transition suboptimal if there exists an $a^\star \in \pi_h^\star(x)$ such that*

$$p(x'|x, a) - p(x'|x, a^\star) \leq \boldsymbol{\alpha} p(x'|x, a) \quad \forall x' \in \mathcal{S}.$$

Intuitively, the condition states that the transition distributions $p(x, a)$ and $p(x, a^\star)$ are close in a pointwise, multiplicative sense. This is motivated by the contextual bandit setting of Definition 2.2, where each $(x, a, h)$ is exactly 0-transition suboptimal. For arbitrary MDPs, the bound $p(x'|x, a^\star) \geq 0$ implies that every triple $(x, a, h)$ is at most 1-transition suboptimal.[5]

With these definitions in place, we can state the more precise analogue of Corollary 2.1 as follows:

**Corollary B.1** (Logarithmic Regret Bound for StrongEuler). *Fix $\delta \in (0, 1/2)$, and let $A = |\mathcal{A}|$, $S = |\mathcal{S}|$, $M = (SAH)^2$. Then with probability at least $1 - \delta$, StrongEuler run with confidence parameter $\delta$ enjoys the following regret bound for all $K \geq 2$:*

$$\mathrm{Regret}_K \lesssim \sum_{(x,a) \in \mathcal{Z}_{\mathrm{sub}}} \frac{\mathrm{Var}_{x,a}^\star (1 \vee \boldsymbol{\alpha} H)}{\mathrm{gap}(x,a)} \log\left(\frac{M}{\delta}\left(\frac{H}{\mathrm{gap}(x,a)} \wedge T\right)\right) + |\mathcal{Z}_{\mathrm{opt}}| \frac{\overline{\mathrm{Var}} H}{\mathrm{gap}_{\mathrm{min}}(x,a)} \log\left(\frac{M}{\delta}\left(\frac{H}{\mathrm{gap}(x,a)} \wedge T\right)\right)$$

$$+ H^4 SA(S \vee H) \log \frac{TM}{\delta} \min\left\{\log \frac{HM}{\mathrm{gap}_{\mathrm{min}}}, \log \frac{TM}{\delta}\right\}$$

*In particular, if $\mathcal{M}$ is an instance of contextual bandits, then $\overline{\mathrm{Var}}$ can be replaced by 1, $\overline{H}_T$ can be replaced by 1 and $\max\{\boldsymbol{\alpha} H, 1\} = 1$. If $\mathcal{M}$ has $\mathcal{G} \lesssim 1$ bounded rewards, then $\overline{\mathrm{Var}}$ can be replaced by 1 in the above bound,*

Moreover, our more precise analogue of Theorem 2.4, which interpolates between the $\log T$ and $\sqrt{T}$ regimes, is as follows:

**Theorem B.2** (Main Regret Bound for StrongEuler). *Fix $\delta \in (0, 1/2)$, and let $A = |\mathcal{A}|$, $S = |\mathcal{S}|$, $M = (SAH)^2$. Let $\overline{H}_T$ be as in Definition B.2, and suppose that each tuple $(x, a, h)$ is $\boldsymbol{\alpha}$-transition suboptimal. Futher, define $\mathcal{Z}_{\mathrm{sub}}(\epsilon) := \{(x, a) \in \mathcal{Z}_{\mathrm{sub}} : \mathrm{gap}(x, a) < \epsilon\}$. Then with probability at least $1 - \delta$, StrongEuler run with confidence parameter $\delta$ enjoys the following regret bound for all $K \geq 2$:*

$$\mathrm{Regret}_K \lesssim \min_{\epsilon > 0}\left\{\sqrt{\overline{H}_T |\mathcal{Z}_{\mathrm{sub}}(\epsilon)| T \log \frac{MT}{\delta}} + \sum_{(x,a) \in \mathcal{Z}_{\mathrm{sub}} \setminus \mathcal{Z}_{\mathrm{sub}}(\epsilon)} \frac{\max\{\boldsymbol{\alpha} H, 1\} \mathrm{Var}_{x,a}^\star}{\mathrm{gap}(x,a)} \left(\frac{M}{\delta}\left(\frac{H}{\mathrm{gap}(x,a)} \wedge T\right)\right)\right\}$$

$$+ \min\left\{\sqrt{\overline{H}_T |\mathcal{Z}_{\mathrm{opt}}| T \log \frac{MT}{\delta}}, |\mathcal{Z}_{\mathrm{opt}}| \frac{H\overline{\mathrm{Var}}}{\mathrm{gap}_{\mathrm{min}}} \log\left(\frac{M}{\delta}\left(\frac{H}{\mathrm{gap}(x,a)} \wedge T\right)\right)\right\}$$

$$+ H^4 SA(S \vee H) \log \frac{MT}{\delta} \min\left\{\log \frac{MH}{\mathrm{gap}_{\mathrm{min}}}, \log \frac{MT}{\delta}\right\}$$

$$\lesssim \sqrt{HSAT \log(T) \log\left(\frac{MT}{\delta}\right)} + H^4 SA(S \vee H) \log^2 \frac{TM}{\delta},$$

*where the second inequality follows from the first with $\max\{\max_\epsilon |\mathcal{Z}_{\mathrm{sub}}(\epsilon)|, |\mathcal{Z}_{\mathrm{opt}}|\} \leq SA$. In particular, if $\mathcal{M}$ is an instance of contextual bandits, then $\overline{\mathrm{Var}}$ can be replaced by 1, $\overline{H}_T$ can be replaced by 1 and $\max\{\boldsymbol{\alpha} H, 1\} = 1$. If $\mathcal{M}$ has $\mathcal{G} \lesssim 1$ bounded rewards, then $\overline{\mathrm{Var}}$ can be replaced by 1 in the above bound, and $\overline{H}_T$ replaced by $\min\{1, \frac{\log T}{H}\}$.*

We observe that Theorem 2.4, and Corollary 2.1 are direct consequences of the above theorem.

**Remark B.2** (Bounds on $|\mathcal{Z}_{\text{opt}}|$). Note that $|\mathcal{Z}_{\text{opt}}| \leq \sum_{x,h} |\pi_h^\star(x)|$; in particular if for each $(x,h)$ there is exactly one optimal action, then $|\mathcal{Z}_{\text{opt}}| \leq H|\mathcal{S}|$. If in addition the same action is optimal at $x$ for each $h \in [H]$, then $|\mathcal{Z}_{\text{opt}}| = |\mathcal{S}|$. For many environments $|\mathcal{Z}_{\text{opt}}| \lesssim |\mathcal{S}|$; for instance, a race car doing many laps around a track may have $h$-dependent optimal actions in the first and last laps, but for the steady-state laps the optimal action will depend just on the current state.

**Remark B.3** (Coupling Variances and Gaps). For state action pairs $(x,a) \in \mathcal{Z}_{\text{sub}}$, Corollary B.1 Theorem B.2 suffer for the term $\frac{(1 \vee \alpha H) \text{Var}_{x,a}^\star}{\text{gap}(x,a)}$, where $\text{Var}_{x,a}^\star := \max_h \text{Var}_{h,x,a}^\star$ is the maximal variance over stages, and $\text{gap}(x,a) = \min_h \text{gap}_h(x,a)$ is the minimal gap. This quantity can be refined to defend on roughly $\max_h \frac{\text{Var}_{h,x,a}^\star}{\text{gap}_h(x,a)}$, coupling the variance and gap terms. To do so, one needs to bin the gaps into intervals of $[2^{j-1}H, 2^j]$ or integers $j \in \mathbb{N}$, and apply numerous careful manipulations. In the interest of brevity, we defer the details to a later work.

## B.2 Rigorous proof of upper bounds: Preliminaries

We now turn to a rigorous proof of the regret bounds for StrongEuler: Corollary B.1 and Theorem B.2 (and consequently Theorem 2.4 and Corollary 2.1).

We first state our generalized surplus clipping bound in terms of the transition-suboptimality condition, which generalizes Proposition 3.1:

**Theorem B.3.** *Suppose that each tuple $(x,a,h)$ is $\alpha_{x,a,h}$ transition-suboptimal, and set $\text{gǎp}_h(x,a) := \frac{\text{gap}_{\min}}{2H} \vee \frac{\text{gap}_h(x,a)}{4(H\alpha_{x,a,h} \vee 1)}$. Then, if $\pi_k$ is induced by a strongly optimistic algorithm with surpluses $\mathbf{E}_{k,h}(x,a)$,*

$$\mathbf{V}_0^\star - \mathbf{V}_0^{\pi_k} \leq 2e \sum_{h=1}^H \sum_{x,a} \boldsymbol{\omega}_{k,h}(x,a) \, \text{clip}\left[\mathbf{E}_{k,h}(x,a) \mid \text{gǎp}_h(x,a)\right].$$

*If the algorithm is optimistic but not strongly optimistic, then the above holds by replacing $\alpha_{x,a,h}$ with 1 in the definition of $\text{gǎp}_h(x,a)$.*

The proof of the above theorem is given in Section D. We remark that the above theorem specializes to Proposition 3.1 by noting that each tuple $(x,a,h)$ is 0-transition suboptimal for contextual bandits. For simplicitiy, we shall assume in the proof of Theorem B.2 that each state is $\alpha$-suboptimal for a common $\alpha$; the bound can be straightforwardly refined to allow $\alpha$ to vary across $(x,a,h)$.

Next, in order to ensure optimal $H$-dependence when interpolating with the $\mathcal{O}\left(\sqrt{T}\right)$ regret bounds, we introduce policy-dependent variance quantities:

**Definition B.4.** Define the variances

$$\text{Var}_{h,x,a}^\pi := \text{Var}[R(x,a)] + \text{Var}_{x' \sim p(x,a)}[\mathbf{V}_{h+1}^\pi(x')].$$

where we recall that $\text{Var}_{h,x,a}^\star := \text{Var}_{h,x,a}^{\pi_*}$. Further, define $\text{Var}_{h,x,a}^{(k)} = \min\left\{\text{Var}_{h,x,a}^\star, \text{Var}_{h,x,a}^{\pi_k}\right\}$.

We are now ready to state the formal version of Proposition 3.2, which upper bounds the surpluses of StrongEuler, and verifies that the algorithm satisfies strong optimism:

**Proposition B.4** (Surplus Bound for StrongEuler). *There exists a universal constant $\mathbf{c} \geq 1$ and event $\mathcal{A}^{\text{conc}}$, with $\mathbb{P}[\mathcal{A}^{\text{conc}}] \geq 1 - \delta/2$, such that on $\mathcal{A}^{\text{conc}}$, for all $x \in \mathcal{S}$, $a \in \mathcal{A}$, $h \in [H]$ and $k \geq 1$,*

$$0 \leq \frac{1}{\mathbf{c}} \mathbf{E}_{k,h}(x,a) \leq \mathbf{B}_{k,h}^{\text{lead}}(x,a) + \mathbb{E}^{\pi_k}\left[\sum_{t=h}^H \mathbf{B}_k^{\text{fut}}(x_t, a_t) \mid (x_h, a_h) = (x,a)\right].$$

*where have defined the terms*

$$\mathbf{B}^{\text{lead}}_{k,h}(x,a) := H \wedge \sqrt{\frac{\text{Var}^{(k)}_{h,x,a} \log\left(\frac{Mn_k(x,a)}{\delta}\right)}{n_k(x,a)}}, \quad and$$

$$\mathbf{B}^{\text{fut}}_k(x,a) = H^3 \wedge H^3 \left(\sqrt{\frac{S \log\left(\frac{Mn_k(x,a)}{\delta}\right)}{n_k(x,a)}} + \frac{S \log\left(\frac{Mn_k(x,a)}{\delta}\right)}{n_k(x,a)}\right)^2.$$

The above proposition is proven in Appendix F. Here $\mathbf{B}^{\text{lead}}$ denotes a "lead term" in the analysis, which contributes to the dominate factors in our regret bounds. $\mathbf{B}^{\text{fut}}$ notates "future" bound terms under a rollout of $\pi_k$ starting at a given triple $(x, a, h)$; these terms are responsible for the lower $\widetilde{\mathcal{O}}\left(SAH^4(S \vee H)\right)$-term in the regret.

**Remark B.4** (Remarks on Proposition B.4). First, the dominant term in the upper bound on $\mathbf{E}_{k,h}$ is $\mathbf{B}^{\text{lead}}_{k,h}(x,a)$, which decays as $\widetilde{\mathcal{O}}\left(n_k(x,a)^{-1/2}\right)$. The terms $\mathbf{B}^{\text{fut}}_k(x,a)$ decay more rapidly $\widetilde{\mathcal{O}}\left(n_k(x,a)^{-1}\right)$, and will thus be responsible for the (nearly gap-free) portion of the regret. Second, in order to analyze similar optimistic algorithms in the same vein (e.g. [1, 4, 5]), one would instead prove the appropriate analogue to Proposition B.4 and follow the remaining steps of the present proof. Little would change, except one would be forced to replace $\text{Var}^\star_{h,x,a}$ with a more pessimistic, less problem-dependent quantity. Lastly, note that the lead term $\mathbf{B}^{\text{lead}}_{k,h}(x,a)$ depends on the *minimum* of the variance of the optimal value function, $\text{Var}^\star_{h,x,a}$ and of the variance of the value function for $\pi_k$, $\text{Var}^{\pi_k}_{h,x,a}$. As in the aforementioned works, this dependence on $\text{Var}^{\pi_k}_{h,x,a}$ is crucial for obtaining the correct minimax $\widetilde{O}(\sqrt{HSAT})$ regret.

Next, let us combine Proposition B.4 with our main clipping theorem, Theorem B.3. Since $\mathbf{E}_{k,h}(x,a) \lesssim \mathbf{B}^{\text{lead}}_{k,h}(x,a) + \mathbb{E}^{\pi_k}\left[\sum(\dots) \mid (\dots)\right]$, combining the two results into a convenient form requires that we reason about how to distribute clipping operations across sums of terms. To this end, we invoke the following technical lemma:

**Lemma B.5** (Distributing the clipping operator). *Let $m \geq 2$, $a_1, \dots, a_m \geq 0$, and $\epsilon \geq 0$. $\text{clip}\left[\sum_{i=1}^m a_i \mid \epsilon\right] \leq 2\sum_{i=1}^m \text{clip}\left[a_i \mid \frac{\epsilon}{2m}\right]$.*

*Proof.* Let us assume without loss of generality $0 \leq a_1 \leq \dots \leq a_m$, and that $\sum_{i=1}^m a_i \geq \epsilon$. Defining the index $i^* := \min\{i : a_i \geq \frac{\epsilon}{2m}\}$, we observe that $a_{i^*} \geq \frac{\epsilon}{2m}$, and since $(a_i)$ are non-decreasing by assumption, $\sum_{i=i^*}^m a_i = \sum_{i=i^*}^m \text{clip}\left[a_i \mid \frac{\epsilon}{2m}\right] \leq \sum_{i=1}^m \text{clip}\left[a_i \mid \frac{\epsilon}{2m}\right]$. It therefore suffices to show that $\sum_{i=1}^m a_i \leq 2\sum_{i=i^*}^m a_i$. To this end, we see that, since $a_i \leq \frac{\epsilon}{2m}$ for $i < i^*$, $\sum_{i=1}^{i^*-1} a_i \leq \sum_{i=1}^{i^*-1} \frac{\epsilon}{2m} \leq \frac{(i^*-1)\epsilon}{2m} \leq \epsilon/2$. On the other hand, since $\sum_{i=1}^m a_i \geq \epsilon$, we must have that $\sum_{i=i^*}^m a_i \geq \frac{\epsilon}{2}$, and thus $\sum_{i=1}^m a_i \leq 2\sum_{i=i^*}^m a_i$, as needed. $\square$

Applying the above lemma careful, we arrive at the following useful regret decomposition:

**Lemma B.6** (Clipped Regret Decomposition Lead and Future Bounds). *Let $\text{găp}_{\min} = \min_{x,a,h} \text{găp}_h(x,a)$. Then on the event $\mathcal{A}^{\text{conc}}$ the regret of StrongEuler is bounded by*

$$\text{Regret}_K \leq 4e \sum_{k=1}^K \sum_{h=1}^H \sum_{x,a} \boldsymbol{\omega}_{k,h}(x,a) \, \text{clip}\left[\mathbf{cB}^{\text{lead}}_{k,h}(x,a) \mid \frac{\text{găp}_h(x,a)}{4}\right].$$

$$+ 8He \sum_{k=1}^K \sum_{h=1}^H \sum_{x,a} \boldsymbol{\omega}_{k,h}(x,a) \, \text{clip}\left[\mathbf{cB}^{\text{fut}}_k(x',a') \mid \frac{\text{găp}_{\min}}{8SAH}\right],$$

*where $\mathbf{c}$ is a universal constant.*

### B.3 Proof of Corollary B.1: A proof via integration

Note that Lemma B.6 bounds $\text{Regret}_K$ by a sum of *local* bounnds terms $\mathbf{B}^{\text{lead}}_{k,h}(x,a)$ and $\mathbf{B}^{\text{fut}}_k$, which depend only on the number of samples $n_k(x,a)$ obtained from state action pair $(x,a)$. More precisely, we can represent the bound terms by defining the functions

$$g_{x,a}^{\text{lead}}(u) := \sqrt{\frac{\text{Var}_{x,a}^{\star}\log\left(\frac{Mu}{\delta}\right)}{u}}, \quad \text{and} \quad g^{\text{fut}}(u) = H^3\left(\sqrt{\frac{S\log\left(\frac{Mu}{\delta}\right)}{u}} + \frac{S\log\left(\frac{Mu}{\delta}\right)}{u}\right)^2.$$

Further, define $\epsilon_{x,a}^{\text{lead}} := \min_h \frac{\text{găp}_h(x,a)}{4}$, $\epsilon^{\text{fut}} := \frac{\text{găp}_{\min}}{8SAH}$, and lastly set

$$f_{x,a}^{\text{lead}}(u) := H \wedge \text{clip}\left[\mathbf{c}g_{x,a}^{\text{lead}}(u)|\epsilon_{x,a}^{\text{lead}}\right], \quad f^{\text{fut}}(u) := H^3 \wedge \text{clip}\left[\mathbf{c}g^{\text{fut}}(u)|\epsilon^{\text{fut}}\right].$$

Then, recalling the definitions of $\mathbf{B}^{\text{lead}}, \mathbf{B}^{\text{fut}}$, and the fact that $\mathbf{B}_{k,h}^{\text{lead}}(x,a) \leq H$ and $\mathbf{B}_k^{\text{fut}}(x,a) \leq H^3$, we can write

$$\text{Regret}_K \lesssim \sum_{k=1}^{K}\sum_{h=1}^{H}\sum_{x,a}\boldsymbol{\omega}_{k,h}(x,a)f_{x,a}^{\text{lead}}(n_k(x,a)) + H\sum_{k=1}^{K}\sum_{h=1}^{H}\sum_{x,a}\boldsymbol{\omega}_{k,h}(x,a)f^{\text{fut}}(n_k(x,a)). \tag{4}$$

As described in Section 3, the crucial step now is to relate the empirical conunts $n_k(x,a)$ to the visitation probabililties. Precisely, let us aggregate

$$\boldsymbol{\omega}_k(x,a) := \sum_{h=1}^{H}\boldsymbol{\omega}_{k,h}(x,a), \quad \overline{n}_k(x,a) := \sum_{j=1}^{k}\boldsymbol{\omega}_j(x,a).$$

Note that if $\{\mathcal{F}_k\}$ denotes the filtration corresponding to the episodes $k$, then, $\mathbb{E}[n_k(x,a) \mid \mathcal{F}_{k-1}] = n_{k-1}(x,a) + \boldsymbol{\omega}_{k-1}(x,a)$. In other words, $\overline{n}_{k-1}(x,a)$ is precise the sum of the increments $\mathbb{E}[n_j(x,a) - n_{j-1}(x,a) \mid \mathcal{F}_{j-1}]$ for $j = 1, \ldots, k-1$.[6] Hence, by a now-standard martingale concentration argument, we find that $n_k(x,a)$ will be be lower bounded by $\overline{n}_k(x,a)$, provided that the latter quantity is sufficiently large. More precisely:

**Lemma B.7** (Sampling Event). *Define the event*

$$\mathcal{E}^{\text{samp}}(H_{\text{sample}}) := \left\{\forall(x,a), \forall k \geq \boldsymbol{\tau}_{H_{\text{sample}}}(x,a), \quad n_k(x,a) \geq \frac{1}{4}\overline{n}_k(x,a)\right\},$$

$$\text{where} \quad \boldsymbol{\tau}_n(x,a) := \inf\{k : \overline{n}_k(x,a) \geq n\}$$

*Then, for some $H_{\text{sample}} \lesssim H\log\frac{M}{\delta}$, $\mathcal{E}^{\text{samp}}(H_{\text{sample}})$ holds with probability at least $1 - \delta/2$.*

Lemma B.7 is proved in Appendix C.2 as a consequence of Lemma 6 in [6]. Together, the events $\mathcal{A}^{\text{conc}}$ and $\mathcal{E}^{\text{samp}}$ account for $1 - \delta$ probability with which our regret bounds hold. For short, we will let $\mathcal{E}^{\text{samp}}$ denote $\mathcal{E}^{\text{samp}}(H_{\text{sample}})$ when clear from context, and $\boldsymbol{\tau} := \boldsymbol{\tau}_{H_{\text{sample}}}$.

After neglecting the first $\boldsymbol{\tau}(x,a)$ samples in the sum (5), we can approximately bound

$$\text{Regret}_K \lessapprox \sum_{x,a}\sum_{k\geq\boldsymbol{\tau}(x,a)}^{K}\sum_{h=1}^{H}\boldsymbol{\omega}_{k,h}(x,a)f_{x,a}^{\text{lead}}(\overline{n}_k(x,a)/4)$$

$$+ \sum_{x,a}H\sum_{k=\boldsymbol{\tau}(x,a)}^{K}\sum_{h=1}^{H}\boldsymbol{\omega}_{k,h}(x,a)f^{\text{fut}}(\overline{n}_k(x,a)/4),$$

where $\lessapprox$ denotes an informal inequality. Now, $\boldsymbol{\omega}_{k,h}(x,a)$ and $\overline{n}_k(x,a)/4$ are directly related via $\overline{n}_k(x,a)/4 := \sum_{j=1}^{k}\sum_{h=1}^{H}\boldsymbol{\omega}_{j,h}(x,a)$. Hence, we can view the above regret bounds as discrete integrals of the functions $f_{x,a}^{\text{lead}}$ and $f^{\text{fut}}(\overline{n}_k(x,a)/4)$. This argument is made precise by the following lemma, which comprises the workhorse of out argument:

**Lemma B.8** (Integral Conversion). *Suppose that the event $\mathcal{E}^{\text{samp}}(H_{\text{sample}})$ holds. Then, for any collection of functions $f_{x,a}(\cdot)$ non-increasing functions from $\mathbb{N} \to \mathbb{R}$ bounded aboved by $f_{\max}$ and any $\epsilon_{x,a,h} \geq 0$, we have that*

$$\sum_{k=1}^{K}\sum_{h=1}^{H}\sum_{x,a}\boldsymbol{\omega}_{k,h}(x,a)f_{x,a}(n_k(x,a)) \leq 2ASH_{\text{sample}}f_{\max} + \sum_{x,a}\mathbb{I}(\overline{n}_K(x,a) \geq H)\int_{H}^{\overline{n}_K(x,a)}f(u/4)du$$

Since $H_{\text{sample}} \lesssim H \log(M/\delta)$, and the functions $f_{x,a}^{\text{fut}}$, $f_{x,a}^{\text{lead}}$ are bounded by $H^3$, we see that, on $\mathcal{E}^{\text{samp}} \cap \mathcal{A}^{\text{conc}}$, it holds that

$$\text{Regret}_K \lesssim SAH^5 \log(M/\delta) + \sum_{x,a} \mathbb{I}(\overline{n}_K(x,a) \geq H) \int_H^{\overline{n}_K(x,a)} f_{x,a}^{\text{lead}}(u/4) du$$

$$+ \sum_{x,a} \mathbb{I}(\overline{n}_K(x,a) \geq H) \int_H^T f^{\text{fut}}(u/4) du, . \tag{5}$$

where for the term on the second line, we have bounded $\overline{n}_K(x,a) \leq T$ and used that $f^{\text{fut}}(\cdot) \geq 0$.

All that remains is to evaluate the above integrals. This is directly adressed by the following technical lemma, proved in Section C.5:

**Lemma B.9** (General Integration Computations). *Let $f(u) \leq \min\{f_{\max}, \text{clip}\,[g(u)|\epsilon]\}$ where $\epsilon \in [0,H]$ and $g(u)$ is a non-increasing function is specified in each of two cases that follow. Further, let $M \geq 1$, and $\delta \in (0, 1/2)$ be problem dependent constants. Finally, let $\lesssim$ denote inequality up to a problem independent constant. Then, the following integral computations hold:*

(a) *Suppose that $C > 0$ is a problem depedendent constant satisfying $\log C \lesssim \log(2M)$, and that $g(u) \lesssim \sqrt{\frac{C \log(Mu/\delta)}{u}}$. Then,*

$$\int_H^N f(u/4) du \lesssim \min\left\{\sqrt{CN \log \frac{MN}{\delta}}, \frac{C}{\epsilon} \log\left(\frac{M}{\delta} \cdot \min\{T, \frac{H}{\epsilon}\}\right)\right\}.$$

(b) *Suppose that $C, C' > 0$ are a problem depedendent constant satisfying $\log(CC') \lesssim \log 2M$, and that $g(u) \lesssim C\left(\sqrt{\frac{C' \log(Mu/\delta)}{u}} + \frac{C' \log(Mu/\delta)}{u}\right)^2$. Then,*

$$\int_H^N f(u/4) du \lesssim (1 + C') f_{\max} \log(\frac{M}{\delta})$$

$$+ CC' \log\left(\frac{MN}{\delta}\right) \min\left\{\log \frac{MN}{\delta}, \log\left(\frac{MH}{\epsilon}\right)\right\}$$

*Note that the special case $g(u) \lesssim \frac{C}{\log(Mu/\delta)} u$ can be obtained by setting $C' = 1$ in the above inequality.*

*Lastly, the above computations hold if $f(u/4)$ is replaced by $f(u/c)$ for any universal constant $c > 0$. Moreover, the above computations hold if $f(u) \lesssim \min\{f_{\max}, g(u)\}$ by taking $\epsilon = 0$ and setting $\frac{1}{\epsilon} = \infty$.*

**Remark B.5** (Integration without anytime bounds). If instead we consider functions $g(u)$ satisfying the looser bounds (a) $g(u) \lesssim \sqrt{\frac{C \log(MT/\delta)}{u}}$ and (b) $g(u) \lesssim C\left(\sqrt{\frac{C' \log(MT/\delta)}{u}} + \frac{C' \log(MT/\delta)}{u}\right)^2$ for $T \geq N$, then we can recover the bounds

$$\int_H^N f(u/4) du \lesssim \begin{cases} \min\left\{\sqrt{CN \log \frac{MT}{\delta}}, \frac{C}{\epsilon} \log \frac{MT}{\delta}\right\} & \text{case (a)} \\ (1 + C') f_{\max} \log \frac{MT}{\delta} + CC' \log\left(\frac{MT}{\delta}\right) \min\{\log \frac{MT}{\delta}, \log \frac{\log(MT/\delta)}{\epsilon}\} & \text{case (b)} \end{cases}$$

These sorts of bounds arise when the confidence intervals are derived via union bounds over all time $T$, rather than via anytime estimates. In particular, we see that using a naive union bounded over all time $T$ incurs a dependence on $\log T \cdot (\log \log T)$, and thus does not imply a strictly $\mathcal{O}(\log T)$ regret.

Let us conclude by applying the above lemma to the terms at hand. First, applying the Part (a) with $f = f_{x,a}^{\text{lead}}$, $g = g_{x,a}^{\text{lead}}$, $C = \text{Var}_{x,a}^\star$, and $H \geq \epsilon = \epsilon_{x,a}^{\text{lead}} := \min_h \frac{\breve{\text{gap}}_h(x,a)}{4} \gtrsim \frac{\text{gap}(x,a)}{(1 \vee \alpha H)}$ for

$(x,a) \in \mathcal{Z}_{\text{sub}}$, and $H \geq \epsilon x, a \gtrsim \frac{\text{gap}_{\min}}{H}$ for $(x,a) \in \mathcal{Z}_{\text{opt}}$, we have that

$$\sum_{x,a} \mathbb{I}(\overline{n}_K(x,a) \geq H) \int_H^{\overline{n}_K(x,a)} f_{x,a}^{\text{lead}}(u/4) du$$

$$\lesssim \sum_{x,a} \frac{\text{Var}_{x,a}^\star}{\min_h \text{gǎp}_h(x,a)} \log\left(\frac{MH}{\delta \min_h \text{gǎp}_h(x,a)}\right)$$

$$\lesssim \sum_{(x,a)\in\mathcal{Z}_{\text{sub}}} \frac{\text{Var}_{x,a}^\star(1 \vee \boldsymbol{\alpha} H)}{\text{gap}(x,a)} \log\frac{MH}{\delta\text{gap}(x,a)} + \sum_{(x,a)\in\mathcal{Z}_{\text{opt}}} \frac{\text{Var}_{x,a}^\star H}{\text{gap}_{\min}(x,a)} \log\frac{MH}{\delta\text{gap}_{\min}}$$

$$\leq \sum_{(x,a)\in\mathcal{Z}_{\text{sub}}} \frac{\text{Var}_{x,a}^\star(1 \vee \boldsymbol{\alpha} H)}{\text{gap}(x,a)} \log\frac{MH}{\delta\text{gap}(x,a)} + |\mathcal{Z}_{\text{opt}}| \frac{\overline{\text{Var}}\, H}{\text{gap}_{\min}(x,a)} \log\frac{MH}{\delta\text{gap}_{\min}}.$$

Similarly, applying the Part (b) with $f = f^{\text{fut}}$, $g = g^{\text{fut}}$, $C' = S$ and $C = H^3$, and $\epsilon = \epsilon^{\text{fut}} := \frac{\text{gǎp}_{\min}}{8SAH} \gtrsim \frac{\text{gap}_{\min}}{H^2}$ (and also satisfying $\epsilon^{\text{fut}} \leq H$), we can bound

$$\lesssim S^2 A H^3 \log\left(\frac{MT}{\delta}\right) \min\left\{\log\frac{MT}{\delta}, \log\left(\frac{MH}{\text{gap}_{\min}}\right)\right\},$$

Plugging the above two displays into (5) concludes the proof of Corollary B.1.

## B.4 Proof of Theorem B.2

We conclude the section by proving the regret bound of Theorem B.2, which interpolates between the $\sqrt{T}$ and $\log T$ regimes. Let us recall the subset $\mathcal{Z}_{\text{sub}}(\epsilon) := \{(x,a) : \text{gap}(x,a) < \epsilon\}$, as well as $\overline{H}_T := \min\left\{\overline{\text{Var}}, \frac{\mathcal{G}^2}{H} \log T\right\}$ .. Retracing the proof of Corollary B.1, it suffices to establish only two points:

$$\sum_{(x,a)\in\mathcal{Z}_{\text{sub}}(\epsilon)} \sum_{k=1}^K \sum_{h=1}^H \boldsymbol{\omega}_{k,h}(x,a) \,\text{clip}\left[\mathbf{cB}_{k,h}^{\text{lead}}(x,a) \,\middle|\, \frac{\text{gǎp}_h(x,a)}{4}\right]$$

$$\lesssim \sqrt{\overline{H}_T\, |\mathcal{Z}_{\text{sub}}(\epsilon)|\, T \log\frac{MT}{\delta}} + SAH^2 \log(M\delta/T)$$

$$\sum_{(x,a)\in\mathcal{Z}_{\text{opt}}} \sum_{k=1}^K \sum_{h=1}^H \boldsymbol{\omega}_{k,h}(x,a) \,\text{clip}\left[\mathbf{cB}_{k,h}^{\text{lead}}(x,a) \,\middle|\, \frac{\text{gǎp}_h(x,a)}{4}\right]$$

$$\lesssim \sqrt{\overline{H}_T\, |\mathcal{Z}_{\text{opt}}|\, T \log\frac{MT}{\delta}} + SAH^2 \log(M\delta/T)$$

For both of these inequalities, we will discard the clipping, and thus the two bounds will be syntatically the same. Hence, let us simply prove the following bound:

$$\sum_{k=1}^K \sum_{h=1}^H \sum_{(x,a)\in\mathcal{Z}_{\text{opt}}} \mathbf{B}_{k,h}^{\text{lead}}(x,a) \lesssim \sqrt{\overline{H}_T\, |\mathcal{Z}_{\text{opt}}|\, T \log\frac{MT}{\delta}}.$$

Since $\overline{H}_T := \min\left\{\overline{\text{Var}}, \frac{\mathcal{G}^2}{H} \log T\right\}$, it suffices to prove the above bound first with $\overline{H}_T$ replaced by $\overline{\text{Var}}$, and then replaced by $\frac{\mathcal{G}^2}{H} \log T$.

**Bound with $\overline{\text{Var}}$:** To obtain a bound involving $\overline{\text{Var}}$, we use the fact that $\mathbf{B}_{k,h}^{\text{lead}}(x,a) \lesssim g^{\text{lead}}(n_k(x,a))$, for the function $g^{\text{lead}}(u) = \sqrt{\overline{\text{Var}}\, \log(Mu/\delta)/u}$. Hence, following the integration arguments in the proof of Corollary B.1, clipped at $\epsilon = 0$, we can bound

$$\sum_{k=1}^K \sum_{h=1}^H \boldsymbol{\omega}_{k,h}(x,a) \mathbf{B}_{k,h}^{\text{lead}}(x,a) \leq H \log(M/\delta) + \sqrt{\overline{\text{Var}}\, \overline{n}_K(x,a) \log(\overline{n}_K(x,a)M/\delta)}$$

$$\leq H^2 \log(M/\delta) + \sqrt{\overline{\text{Var}}\, \overline{n}_K(x,a) \log(TM/\delta)}.$$

Hence, by Cauchy Schwartz, and the bound $\sum_{(x,a)\in\mathcal{Z}_{\text{opt}}}\overline{n}_K(x,a) \le \sum_{(x,a)}\overline{n}_K(x,a) = T$,

$$\sum_{(x,a)\in\mathcal{Z}_{\text{opt}}}\sum_{k=1}^{K}\sum_{h=1}^{H}\boldsymbol{\omega}_{k,h}(x,a)\mathbf{B}_{k,h}^{\text{lead}}(x,a) \lesssim SAH\log(M/\delta) + \sum_{x,a\in\mathcal{Z}_{\text{opt}}}\sqrt{\overline{\text{Var}}\,\overline{n}_K(x,a)\log(TM/\delta)}$$

$$\le SAH\log(M/\delta) + \sqrt{\overline{\text{Var}}\,|\mathcal{Z}_{\text{opt}}|\sum_{x,a\in\mathcal{Z}_{\text{opt}}}\overline{n}_K(x,a)\log(TM/\delta)}$$

$$\le SAH\log(M/\delta) + \sqrt{\overline{\text{Var}}\,|\mathcal{Z}_{\text{opt}}|T\log(TM/\delta)},$$

as needed.

**Bound with $\overline{H}_T$:**  This bound requires a little more subtely. Define the function $f(u) = (1/\max\{u,1\})$. Then, using the definition of $\mathbf{B}_{k,h}^{\text{lead}}(x,a)$ from Proposition B.4, we have

$$\sum_{(x,a)\in\mathcal{Z}_{\text{opt}}}\sum_{h=1}^{H}\mathbf{B}_{k,h}^{\text{lead}}(x,a) \lesssim \sum_{k=1}^{K}\sum_{h=1}^{H}\sum_{(x,a)\in\mathcal{Z}_{\text{opt}}}\boldsymbol{\omega}_{k,h}(x,a)(H \wedge \sqrt{\log(Mn_k(x,a)/\delta)f(n_k(x,a))\text{Var}_{h,x,a}^{\star}})$$

$$\le \sum_{k=1}^{K}\sum_{h=1}^{H}\sum_{(x,a)\in\mathcal{Z}_{\text{opt}}}\boldsymbol{\omega}_{k,h}(x,a)(H \wedge \sqrt{\log(MT/\delta)f(n_k(x,a))\text{Var}_{h,x,a}^{\star}}).$$

Applying the recipe we used for Corollary B.1 will not quite carry over in this setting. Instead, we apply an argument based on Cauchy-Schwartz, defered to Section C.4:

**Lemma B.10** (Cauchy-Schwartz Integration Lemma for $\mathcal{G}$-bounds). *Let $\{V_{x,a,k,h}\}$ be a sequence of numbers, and let $f(u)$ be a nonnegative, non-decreasing function, $f_{\max} > 0$, $\mathbf{L}$ a problem dependent parameter, and let $\mathcal{Z}_0 \subset \mathcal{S} \times \mathcal{A}$. Then, on $\mathcal{E}^{\text{samp}}$,*

$$\sum_{(x,a)\in\mathcal{Z}_0}\sum_{k=1}^{K}\sum_{h=1}^{H}\boldsymbol{\omega}_{k,h}(x,a)f_{\max} \wedge \sqrt{\mathbf{L}f(n_k(x,a))V_{x,a,k,h}}$$

$$\le |\mathcal{Z}_0|H_{\text{sample}}f_{\max} + \sqrt{\mathbf{L}\sum_{k=1}^{K}\mathbb{E}^{\pi_k}\left[\sum_{h=1}^{H}V_{x,a,k,h}\right] \cdot |\mathcal{Z}|(Hf(H) + \int_1^T f(u)du)}.$$

We apply the above lemma with $V_{x,a,k,h} = \text{Var}_{x,a,h}^{\pi_k}$, $f_{\max} = H$ and $f(u) = 1/\max\{u,1\}$, and $\mathbf{L} = \log(MT/\delta)$. It is easy to see that the term $|\mathcal{Z}_0|H_{\text{sample}}f_{\max} \lesssim SAH^2\log(M/\delta)$ will already absorbed into terms already present in the final bound. On the other hand, by a now-standard law of total variance argument,

$$\sum_{k=1}^{K}\mathbb{E}^{\pi_k}\left[\sum_{h=1}^{H}\text{Var}_{x_h,a_h,h}^{\pi_k}\right] \le K \max_{\pi}\mathbb{E}^{\pi}\left[\sum_{h=1}^{H}\text{Var}_{x_h,a_h,h}^{\pi}\right] \le T\mathcal{G}^2/H, \qquad (6)$$

where the last inequality is from the proof of [16], Proposition 6. On the other hand, we can bound $(Hf(H) + \int_1^T f(u)du) \le 1 + \log T$. This finally yields

$$\sum_{k=1}^{K}\mathbb{E}^{\pi_k}\left[\sum_{h=1}^{H}\text{Var}_{x_h,a_h,h}^{\pi_k}\right] \lesssim SAH^2\log(M/\delta) + \sqrt{\log(MT/\delta)T \cdot \mathcal{G}^2/H\log(T)},$$

as needed.

# C   Proof of Technical Lemmas

## C.1   Proof of clipping with future bounds, Lemma B.6

Since strong optimistm holds on $\mathcal{A}^{\text{conc}}$, Theorem B.3 yields

$$\mathbf{V}_0^{\star} - \mathbf{V}_0^{\pi_k} \le 2e\sum_{h=1}^{H}\sum_{x,a}\boldsymbol{\omega}_{k,h}(x,a)\,\text{clip}\left[\mathbf{E}_{k,h}(x,a) \,|\, \breve{\text{g}}\text{ăp}_h(x,a)\right].$$

Applying Lemma B.5 with $m = 2$, $a_1 = \mathbf{cB}_{k,h}^{\text{lead}}(x,a)$, and

$$a_2 = \mathbb{E}^{\pi_k}\left[\sum_{t=h}^{H}\mathbf{cB}_k^{\text{fut}}(x,a) \mid (x_h, a_h) = (x,a)\right],$$

$$\mathbf{V}_0^{\star} - \mathbf{V}_0^{\pi_k} \leq 4e\sum_{h=1}^{H}\sum_{x,a}\boldsymbol{\omega}_{k,h}(x,a)\,\text{clip}\left[\mathbf{cB}_{k,h}^{\text{lead}}(x,a)\mid\frac{\text{gǎp}_h(x,a)}{4}\right].$$

$$4e\sum_{h=1}^{H}\sum_{x,a}\boldsymbol{\omega}_{k,h}(x,a)\,\text{clip}\left[\mathbb{E}^{\pi_k}\left[\sum_{t=h}^{H}\mathbf{cB}_k^{\text{fut}}(x_t,a_t)\mid(x_h,a_h)=(x,a)\right]\mid\frac{\text{gǎp}_h(x,a)}{4}\right].$$

The term on the right hand side of the first line of the above display is exactly as needed. Let us turn our attention to the term on the second line. We have that

$$\mathbb{E}^{\pi_k}\left[\sum_{t=h}^{H}\mathbf{cB}_k^{\text{fut}}(x_t,a_t)\mid(x_h,a_h)=(x,a)\right] = \sum_{x',a'}\sum_{t=h}^{H}\mathbf{cB}_k^{\text{fut}}(x',a')\mathbb{P}[(x_t,a_t)=(x',a')\mid(x_h,a_h)=(x,a)].$$

Hence, applying Lemma B.5 with the terms $a_i$-terms corresponding to $\mathbf{B}_k^{\text{fut}}(x',a')\mathbb{P}[(x_t,a_t)=(x',a')\mid(x_h,a_h)=(x,a)]$ and the number of such terms $m$ bounded by $S\dot{A}H$, we have

$$\text{clip}\left[\mathbb{E}^{\pi_k}\left[\sum_{t=h}^{H}\mathbf{B}_k^{\text{fut}}(x_t,a_t)\mid(x_h,a_h)=(x,a)\right]\mid\frac{\text{gǎp}_h(x,a)}{4}\right]$$

$$\leq 2\sum_{x',a'}\sum_{t=h}^{H}\text{clip}\left[\mathbf{cB}_k^{\text{fut}}(x',a')\mathbb{P}[(x_t,a_t)=(x',a')\mid(x_h,a_h)=(x,a)]\mid\frac{\text{gǎp}_h(x,a)}{8SAH}\right].$$

Since $\text{clip}\left[\alpha x\mid\epsilon\right]\leq\alpha\,\text{clip}\left[x\mid\epsilon\right]$ for $\alpha\leq 1$, and since the probabilities $\mathbb{P}[(x_t,a_t)=(x',a')\mid(x_h,a_h)=(x,a)]$ are bounded by 1, we can bound the above by

$$2\sum_{x',a'}\sum_{t=h}^{H}\mathbb{P}[(x_t,a_t)=(x',a')\mid(x_h,a_h)=(x,a)]\,\text{clip}\left[\mathbf{cB}_k^{\text{fut}}(x',a')\mid\frac{\text{gǎp}_h(x,a)}{8SAH}\right].$$

Hence,

$$4e\sum_{h=1}^{H}\sum_{x,a}\boldsymbol{\omega}_{k,h}(x,a)\,\text{clip}\left[\mathbb{E}^{\pi_k}\left[\sum_{t=h}^{H}\mathbf{cB}_k^{\text{fut}}(x_t,a_t)\mid(x_h,a_h)=(x,a)\right]\mid\frac{\text{gǎp}_h(x,a)}{4}\right]$$

$$\leq 8e\sum_{h=1}^{H}\sum_{x,a}\sum_{x',a'}\boldsymbol{\omega}_{k,h}(x,a)\sum_{t=h}^{H}\mathbb{P}[(x_t,a_t)=(x',a')\mid(x_h,a_h)=(x,a)]\,\text{clip}\left[\mathbf{cB}_k^{\text{fut}}(x',a')\mid\frac{\text{gǎp}_h(x,a)}{8SAH}\right]$$

$$\leq 8e\sum_{h=1}^{H}\sum_{x,a}\sum_{x',a'}\boldsymbol{\omega}_{k,h}(x,a)\sum_{t=h}^{H}\mathbb{P}[(x_t,a_t)=(x',a')\mid(x_h,a_h)=(x,a)]\,\text{clip}\left[\mathbf{cB}_k^{\text{fut}}(x',a')\mid\frac{\text{gǎp}_{\min}}{8SAH}\right]$$

$$= 8e\sum_{h=1}^{H}\sum_{t=h}^{H}\sum_{x',a'}\boldsymbol{\omega}_{k,h}(x',a')\,\text{clip}\left[\mathbf{cB}_k^{\text{fut}}(x',a')\mid\frac{\text{gǎp}_{\min}}{8SAH}\right]$$

$$\leq 8He\sum_{h=1}^{H}\sum_{x,a}\boldsymbol{\omega}_{k,h}(x,a)\,\text{clip}\left[\mathbf{cB}_k^{\text{fut}}(x',a')\mid\frac{\text{gǎp}_{\min}}{8SAH}\right].$$

Altogether,

$$\mathbf{V}_0^{\star} - \mathbf{V}_0^{\pi_k} \leq 4e\sum_{h=1}^{H}\sum_{x,a}\boldsymbol{\omega}_{k,h}(x,a)\,\text{clip}\left[\mathbf{cB}_{k,h}^{\text{lead}}(x,a)\mid\frac{\text{gǎp}_h(x,a)}{4}\right].$$

$$+ \quad 8He\sum_{h=1}^{H}\sum_{x,a}\boldsymbol{\omega}_{k,h}(x,a)\,\text{clip}\left[\mathbf{cB}_k^{\text{fut}}(x',a')\mid\frac{\text{gǎp}_{\min}}{8SAH}\right].$$

Summing over $k = 1, \ldots, K$ proves the inequality.

## C.2 Proof of sampling lemma (Lemma B.7)

Recall $(\mathcal{E}^{\text{samp}})' := \{\forall k, s, a : n_k(x, a) \geq \frac{1}{2}\overline{n}_{k-1}(x, a) - H \log \frac{2HSA}{\delta}\}$; Lemma 6 in [6] in shows that this event occurs with probability at least $1 - \delta/2$. We show that $(\mathcal{E}^{\text{samp}})' \subseteq \mathcal{E}^{\text{samp}}(H_{\text{sample}})$, for $H_{\text{sample}} = 4H \log \frac{2HSA}{\delta} \lesssim H \log \frac{M}{\delta}$.

Noting that $\overline{n}_k \leq \overline{n}_{k-1} + H$, $(\mathcal{E}^{\text{samp}})'$ implies that $n_k \geq \frac{1}{2}\overline{n}_k(x, a) - H \log \frac{2HSA}{\delta} - H = \frac{1}{2}\overline{n}_k(x, a) - H \log \frac{2eHSA}{\delta}$. Hence, for any $k \geq \boldsymbol{\tau}(x, a)$, we have $\overline{n}_k \geq 4H \log \frac{2HSA}{e\delta}$ and thus $n_k(x, a) \geq \frac{\overline{n}_k}{4} + \frac{\overline{n}_k}{4} - H \log \frac{2eHSA}{\delta} \geq \frac{\overline{n}_k}{4}$. Bounding $\log \frac{2HSA}{e\delta} \leq \widetilde{\mathbf{L}}(1)$ concludes the proof.

## C.3 Proof of integral conversion, Lemma B.8

Recall that $\boldsymbol{\tau}(x, a)$ denote $\inf\{k : \overline{n}_k(x, a) \geq H_{\text{sample}}\}$. Then,

$$\sum_{k=1}^{K} \sum_{x,a} \boldsymbol{\omega}_k(x, a) f_{x,a}(n_k(x, a)) = \sum_{x,a} \sum_{k=1}^{\boldsymbol{\tau}(x,a)-1} \boldsymbol{\omega}_k(x, a) f_{x,a}(n_k(x, a)) + \sum_{x,a} \sum_{k=\boldsymbol{\tau}(x,a)}^{} f_{x,a}(n_k)$$

$$\leq \sum_{x,a} \sum_{k=1}^{\boldsymbol{\tau}(x,a)-1} \boldsymbol{\omega}_k(x, a) f_{\max} + \sum_{x,a} \sum_{k=\boldsymbol{\tau}(x,a)}^{} f_{x,a}(n_k)$$

$$\leq SAH_{\text{sample}} f_{\max} + \sum_{x,a} \sum_{k=\boldsymbol{\tau}(x,a)}^{} f_{x,a}(n_k(x, a))$$

$$\leq SAH_{\text{sample}} f_{\max} + \sum_{x,a} \sum_{k=\boldsymbol{\tau}(x,a)}^{} f_{x,a}(\overline{n}_k(x, a)/4),$$

since $\sum_{k=1}^{\boldsymbol{\tau}(x,a)-1} \boldsymbol{\omega}_k(x, a) = \overline{n}_{\boldsymbol{\tau}(x,a)-1}(x, a) \leq H_{\text{sample}}$ madn $f(\cdot) \leq f_{\max}$. We now appeal to the following integration lemma, which we prove momentarily.

**Lemma C.1** (Integration over $\boldsymbol{\omega}_k(x, a)$). *Let $f : [H, \infty) \to \mathbb{R}_{>0}$ be a non-increasing function. Then,*

$$\sum_{k=\boldsymbol{\tau}(x,a)}^{K} \boldsymbol{\omega}_k(x, a) f(\overline{n}_k(x, a)) \leq Hf(H) + \int_{H}^{\overline{n}_K(x,a)} f(u)du. \tag{7}$$

To conclude the proof of Lemma B.8, we apply the above for each $(x, a)$ with the functions $f(u) \leftarrow f_{x,a}(u/4)$, and note $Hf_{x,a}(H/4) \leq Hf_{\max} \leq H_{\text{sample}} f_{\max}$. $\qquad\square$

*Proof of Lemma C.1.* The proof generalizes Lemma E.5 in [5]. For ease of notation, define $k_0 = \tau(x, a)$. We can define the step function $g : [k_0, K] \to \mathbb{R}$ via $g(t) = \sum_{k=k_0}^{K-1} \boldsymbol{\omega}_{k+1}(x, a)\mathbb{I}(t \in [k, k+1)]$. Then, letting $G(t) := \overline{n}_{k_0}(x, a) + \int_0^t g(u)du$, we see that $G'(t) = g(t)$ almost everywhere, $G$ is non-decreasing, and $G(k) = \overline{n}_k(x, a)$ for all $k \in [k_0, K]$. We can therefore express

$$\sum_{k>\tau(x,a)}^{K} \boldsymbol{\omega}_k(x, a) f(\overline{n}_k(x, a)) = \sum_{k=k_0+1}^{K} \boldsymbol{\omega}_k(x, a) f(\overline{n}_k(x, a)) = \sum_{k=k_0+1}^{K} \left(\int_{k-1}^{k} g(t)dt\right) f(G(k))\cdot$$

$$\overset{(i)}{\leq} \sum_{k=k_0+1}^{K} \left(\int_{k-1}^{k} g(t)f(G(t))dt\right) = \int_{k=k_0}^{K} g(t)f(G(t))dt$$

$$\overset{(ii)}{=} \int_{G(k_0)}^{G(K)} f(u)du \overset{(iii)}{=} \int_{\overline{n}_{k_0}(x,a)}^{\overline{n}_K(x,a)} f(u)du,$$

where $(i)$ uses the fact that $f \circ G$ is non-increasing, $(ii)$ is the Fundamental Theorem of Calculus, with $G'(t) = g(t)$, and $(iii)$ is $G(k) = \overline{n}_k(x, a)$ for $k \in [k_0, K]$. Hence, we have the bound

$$\sum_{k \geq k_0}^{K} \boldsymbol{\omega}_k(x, a) f(\overline{n}_k(x, a)) \leq \boldsymbol{\omega}_{k_0}(x, a) f(\overline{n}_{k_0}(x, a)) + \int_{\overline{n}_{k_0}(x,a)}^{\overline{n}_K(x,a)} f(u) du$$

$$\overset{(i)}{\leq} H f(\overline{n}_{k_0}(x, a)) + \int_{\overline{n}_{k_0}(x,a)}^{\overline{n}_K(x,a)} f(u) du \overset{(ii)}{\leq} H f(H) + \int_{H}^{\overline{n}_K(x,a)} f(u) du,$$

where $(i)$ uses $\boldsymbol{\omega}_{k_0} \leq H$, and that $f(u) \geq 0$, and $(ii)$ uses the fact that $f$ is nonincreasing, and $\overline{n}_{k_0}(x, a) \geq H_{\text{sample}} \geq H$. $\qquad \square$

## C.4 Proof of interal conversion for $\mathcal{G}$-bounds, Lemma B.10

Let $\boldsymbol{\tau}(x, a) = \boldsymbol{\tau}_{H_{\text{sample}}}(x, a)$. Then, as in the proof of Lemma B.8,

$$\sum_{(x,a) \in \mathcal{Z}} \sum_{k=1}^{K} \sum_{h=1}^{H} \boldsymbol{\omega}_{k,h}(x, a)(f_{\max} \wedge \sqrt{\widetilde{\mathbf{L}} f(n_k(x, a)) V_{x,a,k,h}})$$

$$\lesssim \sum_{(x,a) \in \mathcal{Z}} \sum_{k=\boldsymbol{\tau}(x,a)}^{K} \sum_{h=1}^{H} \boldsymbol{\omega}_{k,h}(x, a)\sqrt{\widetilde{\mathbf{L}} f(n_k(x, a)) V_{x,a,k,h}} + |\mathcal{Z}_{\text{opt}}| H_{\text{sample}} f_{\max}.$$

By Cauchy-Schwartz

$$\sum_{(x,a) \in \mathcal{Z}} \sum_{k=\boldsymbol{\tau}(x,a)}^{K} \sum_{h=1}^{H} \boldsymbol{\omega}_{k,h}(x, a) \qquad\qquad\qquad\qquad\qquad\qquad\qquad\qquad \lesssim$$

$$\sqrt{\sum_{(x,a) \in \mathcal{Z}} \sum_{k=\boldsymbol{\tau}(x,a)}^{K} \sum_{h=1}^{H} \boldsymbol{\omega}_{k,h}(x, a) V_{x,a,k,h}} \qquad \times \mathbf{L}^{1/2} \sqrt{\sum_{(x,a) \in \mathcal{Z}} \sum_{k=\boldsymbol{\tau}(x,a)}^{K} \sum_{h=1}^{H} \boldsymbol{\omega}_{k,h}(x, a) f(n_k(x, a))}.$$

The first term in the above product can be bounded as

$$\sum_{(x,a) \in \mathcal{Z}} \sum_{k=\boldsymbol{\tau}(x,a)}^{K} \sum_{h=1}^{H} \boldsymbol{\omega}_{k,h}(x, a) V_{x,a,k,h} \leq \sum_{k=1}^{K} \sum_{h=1}^{H} \sum_{x,a} \boldsymbol{\omega}_{k,h}(x, a) V_{x,a,k,h} = \sum_{k=1}^{K} \mathbb{E}^{\pi_k}[\sum_{h=1}^{H} V_{x,a,k,h}].$$

Using Lemma C.1, the second term can be bounded as

$$\sqrt{\sum_{(x,a) \in \mathcal{Z}} \sum_{k=\boldsymbol{\tau}(x,a)}^{K} \sum_{h=1}^{H} \boldsymbol{\omega}_{k,h}(x, a) f(n_k(x, a))} \leq |\mathcal{Z}|(H f(H) + \int_{1}^{T} f(u) du).$$

## C.5 General Integral computations (Lemma B.9)

For convenience, let us restate the lemma we are about to prove.

**Lemma B.9** (General Integration Computations). *Let $f(u) \leq \min\{f_{\max}, \text{clip}\,[g(u)|\epsilon]\}$ where $\epsilon \in [0, H]$ and $g(u)$ is a non-increasing function is specified in each of two cases that follow. Further, let $M \geq 1$, and $\delta \in (0, 1/2)$ be problem dependent constants. Finally, let $\lesssim$ denote inequality up to a problem independent constant. Then, the following integral computations hold:*

(a) *Suppose that $C > 0$ is a problem depedendent constant satisfying $\log C \lesssim \log(2M)$, and that $g(u) \lesssim \sqrt{\frac{C \log(Mu/\delta)}{u}}$. Then,*

$$\int_{H}^{N} f(u/4) du \lesssim \min\left\{\sqrt{CN \log \frac{MN}{\delta}}, \frac{C}{\epsilon} \log\left(\frac{M}{\delta} \cdot \min\{T, \frac{H}{\epsilon}\}\right)\right\}.$$

*(b) Suppose that $C, C' > 0$ are a problem depedendent constant satisfying $\log(CC') \lesssim \log 2M$, and that $g(u) \lesssim C\left(\sqrt{\frac{C' \log(Mu/\delta)}{u}} + \frac{C' \log(Mu/\delta)}{u}\right)^2$. Then,*

$$\int_H^N f(u/4)du \lesssim (1 + C') f_{\max} \log(\tfrac{M}{\delta})$$
$$+ CC' \log\left(\tfrac{MN}{\delta}\right) \min\left\{\log \tfrac{MN}{\delta}, \log\left(\tfrac{MH}{\epsilon}\right)\right\}$$

*Note that the special case $g(u) \lesssim \frac{C}{\log(Mu/\delta)}u$ can be obtained by setting $C' = 1$ in the above inequality.*

*Lastly, the above computations hold if $f(u/4)$ is replaced by $f(u/c)$ for any universal constant $c > 0$. Moreover, the above computations hold if $f(u) \lesssim \min\{f_{\max}, g(u)\}$ by taking $\epsilon = 0$ and setting $\frac{1}{\epsilon} = \infty$.*

*Proof.* By inflating $C$ by a problem-independent constant if necessary, we may assume without loss of generality that $g(u) = \sqrt{C \log(Mu/\delta)/u}$ in part (a) and $g(u) = C(\sqrt{C' \log(Mu/\delta)/u} + \sqrt{C' \log(Mu/\delta)/u})^2$, with equality rather than approximate inequality $\lesssim$.

Next, define

$$n_{\text{end}} := \begin{cases} \max\{u : g(u/4) \geq \epsilon\} & \epsilon > 0 \\ N & \epsilon = 0. \end{cases}$$

Throughout, we shall assume the case $\epsilon > 0$, as the $\epsilon = 0$ can be derived by just taking $n_{\text{end}} = N$. Note then that $f(u/4) = \text{clip}\,[g(u/4)|\epsilon] = 0$ for all $u > n_{\text{end}}$. Hence, it suffices to upper bound

$$\mathbb{I}(n_{\text{end}} \geq H) \int_H^{N \wedge n_{\text{end}}} f_{\max} \wedge g(u/4)du.$$

Lastly, let us define $\widetilde{\mathbf{L}}(u) := \log(Mu/\delta)$ for $u \geq H$. We shall rquire the following inversion lemma, which is standard in the multi-arm bandits literature.

**Lemma C.2** (Inversion Lemma). *There exists a universal constant $c > 0$ such that for all $b \geq 0$, $\widetilde{\mathbf{L}}(u)/u \leq b$ as long as $u \geq \widetilde{\mathbf{L}}(1 + b^{-1})/cb$. Moreover, for $u \lesssim \widetilde{\mathbf{L}}(b^{-1})/cb$, it holds that $\widetilde{\mathbf{L}}(u) \lesssim \widetilde{\mathbf{L}}(1 + b^{-1})$.*

*Proof.* Let $u = \widetilde{\mathbf{L}}(1/b)/cb$ for a constant $c$ to be chosen shortly. Then,

$$\widetilde{\mathbf{L}}(u)/u = cb\frac{\widetilde{\mathbf{L}}(\frac{1}{cb}\widetilde{\mathbf{L}}(b^{-1}))}{\widetilde{\mathbf{L}}(1 + b^{-1})} = cb\frac{\log\frac{1}{c} + \log(M/b\delta) + \log(\log\frac{M}{b\delta})}{\log(M/b\delta)}$$
$$\leq \frac{cb\log\frac{1}{c}}{\log 2} + 2cb,$$

where we use $\log\log(x) \leq x$ and $\widetilde{\mathbf{L}}(1 + b^{-1}) \geq \widetilde{\mathbf{L}}(1) \geq \log 2$. It is easy to see that this quantity is less than $b$ for a constant $c$ sufficiently small that does not depend on $M, \delta, b$. The second statement follows from an analogous computation. $\square$

**Proof of Part (a):** Suppose $g(u) = \sqrt{\frac{C\widetilde{\mathbf{L}}(u)}{u}}$. It is straightforward to bound

$$\mathbb{I}(n_{\text{end}} \geq H) \int_H^{N \wedge n_{\text{end}}} f_{\max} \wedge g(u/4)du \lesssim \mathbb{I}(n_{\text{end}} \geq H)\sqrt{C\widetilde{\mathbf{L}}(N \wedge n_{\text{end}})} \int_1^{N \wedge n_{\text{end}}} \frac{du}{\sqrt{u}}$$
$$\lesssim \sqrt{C\widetilde{\mathbf{L}}(N \wedge n_{\text{end}})} \cdot \sqrt{(N \wedge n_{\text{end}})}$$
$$\lesssim \min\left\{\sqrt{CN\widetilde{\mathbf{L}}(N)}, \sqrt{Cn_{\text{end}}\widetilde{\mathbf{L}}(n_{\text{end}})}\right\}$$
$$= \min\left\{\sqrt{CN\log\frac{MN}{\delta}}, \sqrt{Cn_{\text{end}}\widetilde{\mathbf{L}}(n_{\text{end}})}\right\}.$$

To conclude, let us find $n_{\text{end}}(x, a)$. By our inversion Lemma C.2, we can see that

$$n_{\text{end}} \lesssim \frac{C}{\epsilon^2} \widetilde{\mathbf{L}}(1 + \frac{C}{\epsilon^2}) \lesssim \frac{C}{\epsilon^2} \widetilde{\mathbf{L}}(1 + \frac{C}{\epsilon})$$

$$\widetilde{\mathbf{L}}(n_{\text{end}}) \lesssim \widetilde{\mathbf{L}}(1 + \frac{C}{\epsilon}).$$

Therefore,

$$\sqrt{C n_{\text{end}} \widetilde{\mathbf{L}}(n_{\text{end}})} \lesssim \frac{C}{\epsilon} \widetilde{\mathbf{L}}\left(1 + \frac{C}{\epsilon}\right).$$

Moreover, for if $\log C \lesssim \log M$ and $\epsilon \le H$, we can bound $\widetilde{\mathbf{L}}\left(1 + \frac{C}{\epsilon}\right) \lesssim \log \frac{MH}{\epsilon}$. Hence, we haveshow

$$\int_H^N f(u/4) du \lesssim \min\left\{ \sqrt{CN \log \frac{MN}{\delta}}, \frac{C}{\epsilon} \log\left(\frac{M}{\delta} \cdot \frac{H}{\epsilon}\right) \right\}.$$

To conclude, it remains to show that we can replace $\frac{H}{\epsilon}$ with $N$. For this, we use a simpler argument:

$$\int_H^N f(u/4) du \lesssim \int_H^N \text{clip}\left[ \frac{\sqrt{C\widetilde{\mathbf{L}}(u)}}{u} | \epsilon \right]$$

$$\le \int_H^N \text{clip}\left[ \frac{\sqrt{C\widetilde{\mathbf{L}}(N)}}{u} | \epsilon \right]$$

$$= \sqrt{\widetilde{\mathbf{L}}(T)} \int_H^N \text{clip}\left[ \frac{1}{\sqrt{u}} | \epsilon' \right], \quad \text{where } \epsilon' = \sqrt{\widetilde{\mathbf{L}}(T)}.$$

Using similar arguments to above, we can bound $\int_H^N \text{clip}\left[ \frac{1}{\sqrt{u}} | \epsilon' \right] \lesssim \frac{1}{\epsilon'}$, yielding the bound $\int_H^N f(u/4) du \lesssim \frac{\widetilde{\mathbf{L}}^{1/2}(T)}{\epsilon'} = \frac{\widetilde{\mathbf{L}}(T)}{\epsilon}$.

**Proof of Part (b): A first step**    This proof will require slightly more care than part (b). We shall first require the following lemma:

**Claim C.3.** *In the setting of Lemma B.9, if $g(u) = \frac{C \log \frac{Mu}{\delta}}{u} = \frac{C\widetilde{\mathbf{L}}(u)}{u}$, then*

$$\int_H^N f(u/4) du \lesssim f_{\max} \log M + C \log(MT/\delta) \min\left\{ \log(\frac{MT}{\delta}), \log\left(\frac{MH}{\epsilon}\right) \right\}$$

*Proof of Claim C.3.* Define $n_0 = 2 + \log(M/\delta)$. Then, we have

$$\mathbb{I}(n_{\text{end}} \ge H) \int_H^{N \wedge n_{\text{end}}} f_{\max} \wedge g(u/4) du \le f_{\max} n_0 + \mathbb{I}(N \wedge n_{\text{end}} \ge n_0) \cdot \int_{n_0}^{N \wedge n_{\text{end}}} g(u/4) du.$$

$$\le f_{\max} n_0 + \int_{n_0}^{n_0 + N \wedge n_{\text{end}}} g(u/4) du$$

$$\lesssim f_{\max} \log(M/\delta) + \int_{n_0}^{n_0 + N \wedge n_{\text{end}}} g(u/4) du.$$

Now take $g(u) = C\widetilde{\mathbf{L}}(u)/u$. Since $\widetilde{\mathbf{L}}(u) \lesssim \log(M/\delta) + \log(u)$ for $u \ge n_0 \ge 2$, it it is straightforward to bound

$$\int_{n_0}^{n_0 + N \wedge n_{\text{end}}} g(u/4) du \lesssim C \log(M/\delta) \log(1 + \frac{N \wedge n_{\text{end}}}{n_0}) + C \log^2(1 + \frac{N \wedge n_{\text{end}}}{n_0})$$

$$\lesssim C \log(MT/\delta) \log(1 + \frac{N \wedge n_{\text{end}}}{n_0}),$$

where in the final inequality, we use $N \leq T$, $M/\delta \geq 2$, and $n_0 \geq 1$. By the same token, we can crudely bound the above by $C \lesssim \log^2(MT/\delta)$.

Let us now develop a more refined bound by taking advantage of $n_{\text{end}}$. By our inversion lemma, we have

$$n_{\text{end}} \lesssim \frac{C}{\epsilon} \widetilde{\mathbf{L}} \left( 1 + \frac{C}{\epsilon} \right) = \frac{C}{\epsilon} \left( \log 1 + \frac{C}{\epsilon}) + \log \frac{M}{\delta} \right).$$

Since $n_0 = 1 + \log \frac{M}{\delta}$,

$$\frac{n_{\text{end}}}{n_0} \lesssim \frac{C}{\epsilon} \log(1 + \frac{C}{\epsilon}) + \frac{C}{\epsilon} \frac{\log \frac{M}{\delta}}{\log \frac{M}{\delta}} \lesssim \frac{C}{\epsilon} \log \left( 1 + \frac{C}{\epsilon} \right).$$

Hence, with some algebra we can bound

$$\log(1 + \frac{n_{\text{end}}}{n_0}) \lesssim \log(1 + \frac{C}{\epsilon})$$

This leads to the more refined bound $\int_{n_0}^{n_0 + N \wedge n_{\text{end}}} g(u/4) du \lesssim C \log(1 + \frac{C}{\epsilon}) \log(\widetilde{M}T)$. Again, since $\log C \lesssim \log M$ and $\epsilon \leq H$, we bound again bound $\log(1 + \frac{C}{\epsilon}) \lesssim \log \frac{MH}{\epsilon}$. $\qquad\square$

**Concluding the proof of Part (b)**   Define

$$n_0 := \{\inf u : \sqrt{\frac{C' \widetilde{\mathbf{L}}(u)}{u}} \leq 1\}.$$

Then, we have

$$\int_H^N f(u/4) du \leq f_{\max} n_0 + \int_{n_0}^N f_{\max} \wedge \text{clip}\left[g(u/4)|\epsilon\right] du.$$

Note that for $u \geq n_0$, $g(u/4) \lesssim h(u/4)$, where $h(u) \leq \frac{CC'}{u} \widetilde{\mathbf{L}}(u)$. Hence, applying the bound from Lemma C.3 with $C \leftarrow CC'$, we have

$$\int_H^N f(u/4) du \lesssim f_{\max} \log \widetilde{M} + CC' \log(\widetilde{M}T) \min \left\{ \log(\frac{MT}{\delta}), \log(\frac{MH}{\epsilon}) \right\}$$

On the otherhand, by our inversion lemma and using $C' \leq \widetilde{M}^{\mathcal{O}(1)}$, we can bound

$$n_0 \leq C' \widetilde{\mathbf{L}}(C') \lesssim C' \log(\frac{M}{\delta}).$$

Combining these two pieces yields the bound. $\qquad\square$

# D  Proof of 'clipping' bound: Proposition 3.1 / Theorem B.3

In this section, we prove Theorem B.3 (of which Proposition 3.1 in the body is a direct consequence), which allows us to clip the surpluses when they are below a certain value. The center of our analysis is the following lemma, which tells us that if $\mathsf{gap}_h(x,a) > 0$ for a pair $(x,a,h)$, then either the surplus $\mathbf{E}_{k,h}(x,a)$ is large, or expected difference in value functions at the next stage, $p(x,a)^\top (\overline{\mathbf{V}}_{k,h+1} - \mathbf{V}_{h+1}^{\pi_k})$, is large:

**Lemma D.1** (Fundamental Gap Bound). *Then suppose that* Alg *is strongly optimistic, and consider a pair $(x,a,h)$ with $\mathsf{gap}_h(x,a) > 0$ which is is $\boldsymbol{\alpha}$-transition optimal. Then*

$$\mathsf{gap}_h(x,a) \le \mathbf{E}_{k,h}(x,a) + \boldsymbol{\alpha} \cdot p(x,a)^\top (\overline{\mathbf{V}}_{k,h+1} - \mathbf{V}_{h+1}^{\pi_k}).$$

*If* Alg *is possibly not strongly optimistic, then the above holds still holds $\boldsymbol{\alpha} = 1$.*

Lemma D.1 is established in Section D.2. Notice that as $\boldsymbol{\alpha}$ gets close to zero, the above bound implies that when $\mathbf{E}_{k,h}(x,a)$ is much smaller than the $\mathsf{gap}_h(x,a)$, the difference in value functions at the next stage, $p(x,a)^\top (\overline{\mathbf{V}}_{k,h+1} - \mathbf{V}_{h+1}^{\pi_k})$, must become even larger to compensate. The extreme case is $\boldsymbol{\alpha} = 0$, e.g. in contextual bandits, where the gap always lower bounds the surplus.

Continuing with the proof of Theorem B.3, we begin with the "half-clipping" which clips the surpluses at at most $\mathsf{gap}_{\min}$:

**Definition D.1** (Half Clipped Value Function). We define the half-clipped surplus $\ddot{\mathbf{E}}_{k,h}(x,a) := \mathrm{clip}\left[\mathbf{E}_{k,h}(x,a) \mid \epsilon_{\mathrm{clip}}\right]$, where $\epsilon_{\mathrm{clip}} := \mathsf{gap}_{\min}/(2H)$. We set $\ddot{\mathbf{V}}_{k,H+1}^{\pi_k}(x) = 0$ for all $x \in \mathcal{S}$, and recursively define

$$\ddot{\mathbf{Q}}_h^{\pi_k}(x,a) = r(x,a) + \ddot{\mathbf{E}}_{k,h}(x,a) + p(x,a)^\top \ddot{\mathbf{V}}_{k,h+1}^{\pi_k}, \quad \ddot{\mathbf{V}}_{k,h}^{\pi_k}(x) := \ddot{\mathbf{Q}}_h^{\pi_k}(x, \pi_{k,h}(x)),$$

denote the value and Q-functions of under $\pi_k$ associated with MDP whose transitions are transitions $p(\cdot,\cdot)$ and non-stationary rewards $r(x,a) + \ddot{\mathbf{E}}_{k,h}(x,a)$ at stage $h$.

After the half-clipping has been introduced, it is no longer the case that $\pi_k$ is optimal for this half clipped MDP. As a result, it is not certain that the half-clipped Q-function for $\pi_k$ is *optimistic* in the sense that $\ddot{\mathbf{Q}}_{k,h}^{\pi_k}(x,a) \ge \mathbf{Q}_h^\star(x,a)$. We shall instead show that if $\ddot{\mathbf{V}}^{\pi_{k,h}}$ is approximately optimistic, in the sense that its excess relative to $\mathbf{V}^{\pi_k}$, $\ddot{\mathbf{V}}_0^{\pi_{k,h}} - \mathbf{V}_0^{\pi_k}$ is at least a constant factor of the regret $\mathbf{V}_0^\star - \mathbf{V}_0^{\pi_k}$:

**Lemma D.2** (Lower Bound on Half-Clipped Surplus). *For $\epsilon_{\mathrm{clip}} = \mathsf{gap}_{\min}/2H$, it holds that*

$$\ddot{\mathbf{V}}_0^{\pi_k} - \mathbf{V}_0^{\pi_k} = \mathbb{E}^{\pi_k}\left[\sum_{h=1}^H \ddot{\mathbf{E}}_{k,h}(x_h, a_h)\right] \ge \frac{1}{2}(\mathbf{V}_0^\star - \mathbf{V}_0^{\pi_k}),$$

The above bound is established in Section D.1. Hence, to establish the bound of Theorem B.3, it suffices to bound the gap $\ddot{\mathbf{V}}_0^{\pi_{k,h}} - \mathbf{V}_0^{\pi_k}$. For a given $h$, and an $x : \pi_{k,h}(x) \notin \pi_h^\star(x)$, let us consider the difference

$$\ddot{\mathbf{V}}_h^{\pi_k}(x) - \mathbf{V}_h^{\pi_k}(x) = \ddot{\mathbf{E}}_{k,h}(x, \pi_{k,h}(x)) + p(x, \pi_{k,h}(x))^\top \left(\ddot{\mathbf{V}}_{h+1}^{\pi_k} - \mathbf{V}_{h+1}^{\pi_k}\right).$$

We now introduce the following lemma, proven Section D.3, which allows us to further clip the bonus for suboptimal actions $a \notin \pi_h^\star(x)$, i.e. , actions with $\mathsf{gap}_h(x,a) > 0$:

**Lemma D.3** (Gap Clipping). *Suppose either* Alg *is strongly optimistic and each tuple is $\boldsymbol{\alpha}_{x,a,h}$-transition suboptimal. Then the fully-clipped surpluses*

$$\check{\mathbf{E}}_{k,h}(x,a) := \mathrm{clip}\left[\mathbf{E}_{k,h}(x,a) \mid \epsilon_{\mathrm{clip}} \vee \frac{\mathsf{gap}_h(x,a))}{4(\boldsymbol{\alpha}_{x,a,h} H \vee 1)}\right]$$

*satisfy the bound*

$$\ddot{\mathbf{V}}_h^{\pi_k}(x) - \mathbf{V}_h^{\pi_k}(x) \le \check{\mathbf{E}}_{k,h}(x, \pi_{k,h}(x)) + \left(1 + \frac{1}{H}\right) p(x, \pi_{k,h}(x))^\top \left(\ddot{\mathbf{V}}_{h+1}^{\pi_k} - \mathbf{V}_{h+1}^{\pi_k}\right)$$

*If* Alg *is just optimistic, then the above bound holds with $\boldsymbol{\alpha}_{x,a,h} = 1$.*

Unfolding the above lemma, and noting that even when Alg is not strongly optimistic, the clipping ensures that $\check{\mathbf{E}}_{k,h}(x,a) \geq 0$, so that we can bound

$$\ddot{\mathbf{V}}_{k,0}^{\pi_k} - \mathbf{V}_0^{\pi_k} = \mathbb{E}^{\pi_k}[\ddot{\mathbf{V}}_1^{\pi_k}(x_1) - \mathbf{V}_1^{\pi_k}(x_1)]$$

$$\leq \mathbb{E}^{\pi_k}[\check{\mathbf{E}}_{k,h}(x_1, a_1) + \left(1 + \frac{1}{H}\right) p(x, \pi_{k,h}(x))^\top \left(\ddot{\mathbf{V}}_2^{\pi_k} - \mathbf{V}_2^{\pi_k}\right)]$$

$$= \mathbb{E}^{\pi_k}[\check{\mathbf{E}}_{k,h}(x_1, a_1) + \left(1 + \frac{1}{H}\right) \left(\ddot{\mathbf{V}}_2^{\pi_{k,h}}(x_2) - \mathbf{V}_2^{\pi_k}(x_2)\right)]$$

$$\leq \mathbb{E}^{\pi_k}\left[\sum_{h=1}^{H} \left(\prod_{h'=2}^{h}\left(1 + \frac{1}{H}\right)\right) \check{\mathbf{E}}_{k,h}(x_h, a_h)\right] \leq \left(1 + \frac{1}{H}\right)^H \mathbb{E}^{\pi_k}\left[\sum_{h=1}^{H} \check{\mathbf{E}}_{k,h}(x_h, a_h)\right]$$

$$\leq e\mathbb{E}^{\pi_k}\left[\sum_{h=1}^{H} \check{\mathbf{E}}_{k,h}(x_h, a_h)\right] = e \sum_{x,a} \sum_{h=1}^{H} \boldsymbol{\omega}_{k,h}(x,a)\check{\mathbf{E}}_{k,h}(x,a),$$

where we recall $\boldsymbol{\omega}_{k,h}(x,a) = \mathbb{P}^{\pi_k}[(x_h, a_h) = (x,a)]$. Combining with our earlier bound $\mathbf{V}_0^\star - \mathbf{V}_0^{\pi_k} \leq 2(\ddot{\mathbf{V}}_{k,0}^{\pi_k}(x) - \mathbf{V}_0^{\pi_k}(x))$ from Lemma D.2, we find that $\mathbf{V}_0^\star - \mathbf{V}_0^{\pi_k} \leq 2e \sum_{x,a} \sum_{h=1}^{H} \boldsymbol{\omega}_{k,h}(x,a)\check{\mathbf{E}}_{k,h}(x,a)$, thereby demonstrating Theorem B.3.

### D.1 Proof of Lemma D.2

We can with a crude comparison between the clipped and optimistic value functions.

**Lemma D.4.** *We have that* $\ddot{\mathbf{E}}_{k,h}(x, \pi_{k,h}(x)) \geq \mathbf{E}_{k,h}(x, \pi_{k,h}(x)) - \epsilon_{\mathrm{clip}}$, *which implies*

$$\ddot{\mathbf{V}}_{k,h}^{\pi_k}(x) + (H - h + 1)\epsilon_{\mathrm{clip}} \geq \overline{\mathbf{V}}_{k,h}(x) \geq \mathbf{V}_h^{\pi_k}(x). \tag{8}$$

*Proof.* The bound $\ddot{\mathbf{E}}_{k,h}(x, \pi_{k,h}(x)) \geq \mathbf{E}_{k,h}(x, \pi_{k,h}(x)) - \epsilon_{\mathrm{clip}}$ follows directly from

$$\ddot{\mathbf{E}}_{k,h}(x,a) = \mathbf{E}_{k,h}(x,a)\mathbb{I}(\mathbf{E}_{k,h}(x,a) \geq \epsilon_{\mathrm{clip}}) \geq \mathbf{E}_{k,h}(x,a) - \epsilon_{\mathrm{clip}}.$$

Hence,

$$\ddot{\mathbf{V}}_{k,h}^{\pi_k}(x) - \mathbf{V}_h^{\pi_k}(x) \overset{(i.a)}{=} \mathbb{E}^{\pi_k}\left[\sum_{t=h}^{H} \ddot{\mathbf{E}}_{k,t}(x_t, \pi_{k,h}(x_t)) \mid x_h = x\right]$$

$$\geq \mathbb{E}^{\pi_k}\left[\sum_{t=h}^{H} \mathbf{E}_{k,t}(x_t, \pi_{k,h}(x_t))) - \epsilon_{\mathrm{clip}} \mid x_h = x\right]$$

$$= \mathbb{E}^{\pi_k}\left[\sum_{t=h}^{H} \mathbf{E}_{k,t}(x_t, \pi_{k,h}(x_t)))\right] - (H - h + 1)\epsilon_{\mathrm{clip}}$$

$$\overset{(i.b)}{=} \overline{\mathbf{V}}_{k,h}(x) - \mathbf{V}_h^{\pi_k}(x) - (H - h + 1)\epsilon_{\mathrm{clip}},$$

where $(i.a)$ and $(i.b)$ follow by recursively unfolding the identities $\ddot{\mathbf{V}}_{k,h}^{\pi_k}(x) - \mathbf{V}_h^{\pi_k}(x) = \ddot{\mathbf{E}}_{k,h}(x,a)$ $+p(x,a)^\top(\ddot{\mathbf{V}}_{k,h+1}^{\pi_k}(x) - \mathbf{V}_h^{\pi_k}(x))$ and $\overline{\mathbf{V}}_{k,h}(x) - \mathbf{V}_h^{\pi_k}(x) = \mathbf{E}_{k,h}(x,a) + p(x,a)^\top(\overline{\mathbf{V}}_{k,h+1}(x) - \mathbf{V}_h^{\pi_k}(x))$. $\square$

We now turn to proving Lemma D.2.

*Proof.* The strategy is as follows. We shall introduce the events over $\mathbb{P}^{\pi_k}$, $\mathcal{E}_h := \{\pi_{k,h}(x_h) \notin \pi_h^\star(x_h)\}$, which is the event that the policy $\pi_{k,h}$ does not prescribe an optimal action $x_h$. We further define the events

$$\mathcal{A}_h = \mathcal{E}_h \cap \bigcap_{h'<h} \mathcal{E}_{h'}^c,$$

which is the event that the policy $\pi_k$ agrees with an optimal action on $x_1, \dots, x_{h-1}$, and disagrees on $x_h$. Below, our goal will be to establish the following two formulae for the suboptimality gap $\mathbf{V}_0^\star - \mathbf{V}_0^{\pi_k}$ and $\ddot{\mathbf{V}}_0^{\pi_k} - \mathbf{V}_0^{\pi_k}$:

$$\ddot{\mathbf{V}}_0^{\pi_k} - \mathbf{V}_0^{\pi_k} \geq \sum_{h=1}^{H} \mathbb{E}^{\pi_k}[\mathbb{I}(\mathcal{A}_h)\{\mathrm{gap}(x_h, \pi_{k,h}(x_h)) - H\epsilon_{\mathrm{clip}} + \mathbf{Q}_h^\star(x_h, \pi_{k,h}(x_h)) - \mathbf{V}_h^{\pi_k}(x_h)\}]$$
(9)

and

$$\mathbf{V}_0^\star - \mathbf{V}_0^{\pi_k} = \sum_{h=1}^{H} \mathbb{E}^{\pi_k}[\mathbb{I}(\mathcal{A}_h)\ \{\mathrm{gap}(x_h, \pi_{k,h}(x_h)) + \mathbf{Q}_h^\star(x_h, \pi_{k,h}(x_h)) - \mathbf{V}_h^{\pi_k}(x_h)\}] \qquad (10)$$

Note that on $\mathcal{A}_h$, $\mathcal{E}_h = \{\pi_{k,h}(x_h) \notin \pi_h^\star(x_h)\}$ also occurs, and therefore $\mathrm{gap}(x_h, \pi_{k,h}(x_h)) \geq \mathrm{gap}_{\min}$. In particular, displays (9) and (10) both imply

$$\ddot{\mathbf{V}}_0^{\pi_{k,h}} - \mathbf{V}_0^{\pi_k} \overset{(i)}{\geq} \sum_{h=1}^{H} \mathbb{E}^{\pi_k}[\mathbb{I}(\mathcal{A}_h)\left\{\frac{1}{2}\mathrm{gap}(x_h, \pi_{k,h}(x_h)) + \mathbf{Q}_h^\star(x_h, \pi_{k,h}(x_1)) - \mathbf{V}_h^{\pi_k}(x)\right\}]$$

$$\overset{(ii)}{\geq} \frac{1}{2}\sum_{h=1}^{H} \mathbb{E}^{\pi_k}[\mathbb{I}(\mathcal{A}_h)\{\mathrm{gap}(x_h, \pi_{k,h}(x_j)) + \mathbf{Q}_h^\star(x_h, \pi_{k,h}(x_h)) - \mathbf{V}_h^{\pi_k}(x)\}]$$

$$\overset{(iii)}{\geq} \frac{1}{2}(\mathbf{V}_0^\star - \mathbf{V}_0^{\pi_k}),$$

where $(i)$ uses $\epsilon_{\mathrm{clip}} = \frac{\mathrm{gap}_{\min}}{2H}$ and display (9), $(ii)$ uses that $\mathbf{Q}_h^\star(x_h, \pi_{k,h}(x_h)) - \mathbf{V}_h^{\pi_k}(x) \geq 0$, and $(iii)$ uses display (10).

Let us start with proving (9). First, consider a stage $h$, state $x$, and suppose that $\pi_{k,h}(x) \notin \pi_h^\star(x)$. Observe that by Lemma D.4, optimism, and the definition of $\mathrm{gap}_h(x, a)$, we have that for any $a^\star \in \pi_h^\star(x)$,

$$H\epsilon_{\mathrm{clip}} + \ddot{\mathbf{V}}_{k,h}^{\pi_k}(x) \geq \overline{\mathbf{V}}_{k,h}(x) = \overline{\mathbf{Q}}_{k,h}(x, \pi_{k,h}(x)) \geq \overline{\mathbf{Q}}_{k,h}(x, a^\star)$$

$$\geq \mathbf{Q}_h^\star(x, a^\star) = \mathrm{gap}(x, \pi_{k,h}(x)) + \mathbf{Q}_h^\star(x, \pi_{k,h}(x)).$$

Subtracting, we find that for $\pi_{k,h}(x) \notin \pi_h^\star(x)$,

$$\ddot{\mathbf{V}}_{k,h}^{\pi_k}(x) - \mathbf{V}_h^{\pi_k}(x) \geq \mathrm{gap}(x, \pi_{k,h}(x)) - H\epsilon_{\mathrm{clip}} + \mathbf{Q}_h^\star(x, \pi_{k,h}(x)) - \mathbf{V}_h^{\pi_k}(x). \qquad (11)$$

Now, on the other hand, if $\pi_{k,h}(x) \in \pi_h^\star(x)$, then,

$$\ddot{\mathbf{V}}_{k,h}^{\pi_k}(x) - \mathbf{V}_h^{\pi_k}(x) = \ddot{\mathbf{E}}_{k,h}(x, \pi_{k,h}(x)) + r(x, \pi_{k,h}(x)) + p(x, \pi_{k,h}(x))^\top \ddot{\mathbf{V}}_{k,h+1}^{\pi_{k,h}}$$

$$- r(x, \pi_{k,h}(x)) - p(x, \pi_{k,h}(x))^\top \mathbf{V}_h^{\pi_k}$$

$$= \ddot{\mathbf{E}}_{k,h}(x, \pi_{k,h}(x)) + p(x, \pi_{k,h}(x))^\top (\ddot{\mathbf{V}}_{k,h+1}^{\pi_{k,h}} - \mathbf{V}_{h+1}^{\pi_k}) \qquad (12)$$

$$\overset{(i)}{=} \ddot{\mathbf{E}}_{k,h}(x, \pi_{k,h}(x)) + p(x, \pi_{k,h}(x))^\top \partial \ddot{\mathbf{V}}_{h+1}$$

$$\overset{(ii)}{\geq} p(x, \pi_{k,h}(x))^\top \partial \ddot{\mathbf{V}}_{h+1}, \qquad (13)$$

where in $(i)$ we have defined the increment $\partial \ddot{\mathbf{V}}_h := \ddot{\mathbf{V}}_{k,h}^{\pi_{k,h}} - \mathbf{V}_h^{\pi_k}$ with $\partial \ddot{\mathbf{V}}_{H+1} = 0$, and $(ii)$ holds since $\ddot{\mathbf{E}}_{k,h}(x, \pi_{k,h}(x)) = \mathbf{E}_{k,h}(x, \pi_{k,h}(x))\mathbb{I}(\mathbf{E}_{k,h}(x, \pi_{k,h}(x)) \geq \epsilon_{\mathrm{clip}}) \geq 0$.

Now, recalling that $\mathcal{E}_h$ denotes the event that $\pi_{k,h}(x) \notin \pi_h^\star(x)$, we have

$$\mathbb{E}^{\pi_k}[\partial \ddot{\mathbf{V}}_1] \geq \mathbb{E}^{\pi_k}[\mathbb{I}(\mathcal{E}_1)\{\mathrm{gap}(x_1, \pi_{k,1}(x_1)) - H\epsilon_{\mathrm{clip}} + \mathbf{Q}_h^\star(x_1, \pi_{k,1}(x_1)) - \mathbf{V}_h^{\pi_k}(x)\}]$$
(by Eq. (11))

$$+ \mathbb{E}^{\pi_k}\left[\mathbb{I}(\mathcal{E}_1^c)p(x_1, \pi_{k,1}(x_1))^\top \partial \ddot{\mathbf{V}}_2\right].$$
(by Eq. (13))

We continue with

$$\mathbb{E}^{\pi_k}\left[\mathbb{I}(\mathcal{E}_1^c)p(x_1,\pi_{k,1}(x_1))^\top\partial\ddot{\mathbf{V}}_2\right]$$
$$\geq \mathbb{E}^{\pi_k}\left[\mathbb{I}(\mathcal{E}_1^c)\mathbb{I}(\mathcal{E}_2)\left\{\mathtt{gap}(x_2,\pi_{k,2}(x_2))-H\epsilon_{\mathrm{clip}}+\mathbf{Q}_2^\star(x_2,\pi_{k,2}(x_2))-\mathbf{V}_2^{\pi_k}(x)\right\}\right]$$
$$+\mathbb{E}^{\pi_k}\left[\mathbb{I}(\mathcal{E}_1^c)\mathbb{I}(\mathcal{E}_2^c)p(x_2,\pi_{k,h}(x_2))^\top\partial\ddot{\mathbf{V}}_3\right].$$

Recalling the event $\mathcal{A}_h = \mathcal{E}_h \cap \bigcap_{h'<h}\mathcal{E}_{h'}^c$, we can continue the above induction to find that,

$$\mathbb{E}^{\pi_k}[\partial\ddot{\mathbf{V}}_1] \geq \sum_{h=1}^{H}\mathbb{E}^{\pi_k}\left[\mathbb{I}(\mathcal{A}_h)\left\{\mathtt{gap}(x_h,\pi_{k,h}(x_h))-H\epsilon_{\mathrm{clip}}+\mathbf{Q}_h^\star(x_h,\pi_{k,h}(x_h))-\mathbf{V}_h^{\pi_k}(x)\right\}\right]$$

$$+\underbrace{\mathbb{E}^{\pi_k}[\mathbb{I}(\bigcap_{h=1}^{H}\mathcal{E}_h^c)p(x_h,\pi_{k,h}(x_h))^\top\partial\ddot{\mathbf{V}}_{H+1}]}_{=0},$$

as needed. Now let's prove (10). We can always write

$$\mathbf{V}_h^\star(x)-\mathbf{V}_h^{\pi_k}(x)=\mathtt{gap}_h(x,a)+\mathbf{Q}_h^\star(x,\pi_{k,h}(x))-\mathbf{V}_h^{\pi_k}(x),$$

where $\mathtt{gap}_h(x,a)=0$ when $\pi_h^\star(x)\in\pi_{k,h}(x)$, that is, on $\mathcal{E}^c$. Hence, the same line of reasoning used to prove Eq. (9) (omitting the subtracted $\epsilon_{\mathrm{clip}}H$), verifies Eq. (10).  $\square$

## D.2   Proof of Lemma D.1

*Proof.* For simplicity, set $a=\pi_{k,h}(x)$, and let $a^\star\in\pi_h^\star(x)$ be an action which witnesses the $\alpha$ transition-suboptimality condition. We then have

$$\overline{\mathbf{V}}_{k,h}(x)\overset{(i)}{=}\overline{\mathbf{Q}}_{k,h}(x,a)\overset{(ii)}{\geq}\overline{\mathbf{Q}}_{k,h}(x,a^\star)$$
$$=\mathbf{Q}_h^\star(x,a^\star)+\left(\overline{\mathbf{Q}}_{k,h}(x,a^\star)-\mathbf{Q}_h^\star(x,a^\star)\right)$$
$$\overset{(iii)}{=}\mathtt{gap}_h(x,a)+\mathbf{Q}_h^\star(x,a)+\left(\overline{\mathbf{Q}}_{k,h}(x,a^\star)-\mathbf{Q}_h^\star(x,a^\star)\right),$$

where $(i)$ is by definition of $\overline{\mathbf{V}}_{k,h}(x)$, $(ii)$ is since $a=\pi_{k,h}(x)=\arg\max_{a'}\overline{\mathbf{Q}}_{k,h}(x,a')$, and $(iii)$ is the definition of $\mathtt{gap}_h(x,a)$. Rearranging, we have

$$\mathtt{gap}_h\leq\overline{\mathbf{V}}_{k,h}(x)-\mathbf{Q}_h^\star(x,a)-\left(\overline{\mathbf{Q}}_{k,h}(x,a^\star)-\mathbf{Q}^\star(x,a^\star)\right) \tag{14}$$

If Alg is not necessarily strongly optimistic then we bound $\overline{\mathbf{Q}}_{k,h}(x,a^\star)-\mathbf{Q}^\star(x,a^\star)\geq 0$ and $\mathbf{Q}_h^\star(x,a)\geq\mathbf{V}_h^{\pi_k}(x)$, yielding

$$\mathtt{gap}_h(x,a)\leq\overline{\mathbf{V}}_{k,h}(x)-\mathbf{V}_h^{\pi_k}(x)$$
$$=\overline{\mathbf{Q}}_{k,h}(x,a)-\mathbf{V}_h^{\pi_k}(x)$$
$$=\mathbf{E}_{k,h}(x,a)+r(x,a)+p(x,a)^\top\overline{\mathbf{V}}_{k,h+1}-\mathbf{V}_h^{\pi_k}(x)$$
$$=\mathbf{E}_{k,h}(x,a)+p(x,a)^\top(\overline{\mathbf{V}}_{k,h+1}-\mathbf{V}_{h+1}^{\pi_k})$$

which corresponds to the desired bound for $\alpha=1$.

When Alg is strongly optimistic, we handle (14) more carefully. Specifically, we compute

$$\overline{\mathbf{V}}_{k,h}(x)-\mathbf{Q}_h^\star(x,a)=\mathbf{E}_{k,h}(x,a)+r(x,a)+p(x,a)^\top\overline{\mathbf{V}}_{k,h+1}-\left(r(x,a)+p(x,a)^\top\mathbf{V}_{h+1}^\star\right)$$
$$=\mathbf{E}_{k,h}(x,a)+p(x,a)^\top(\overline{\mathbf{V}}_{k,h+1}-\mathbf{V}_{h+1}^\star).$$

Moreover, recalling that $a^*\in\pi_h^\star(x)$, we have

$$\overline{\mathbf{Q}}_{k,h}(x,a^*)-\mathbf{Q}^\star(x,a^*)=r(x,a^*)+\mathbf{E}_{k,h}(x,a^*)+p(x,a^*)^\top\overline{\mathbf{V}}_{k,h+1}-r(x,a^*)-p(x,a^*)^\top\mathbf{V}_{h+1}^\star$$
$$=\mathbf{E}_{k,h}(x,a^*)+p(x,a^*)^\top(\overline{\mathbf{V}}_{k,h+1}-\mathbf{V}_{h+1}^\star).$$

where the last inequality uses strong optimism of Alg. Hence,

$$
\begin{aligned}
\text{gap}_h(x,a) &\leq \overline{\mathbf{V}}_{k,h}(x) - \mathbf{Q}_h^\star(x,a) - \left(\overline{\mathbf{Q}}_{k,h}(x,a^\star) - \mathbf{Q}^\star(x,a^\star)\right) \\
&= \mathbf{E}_{k,h}(x,a) + p(x,a)^\top(\overline{\mathbf{V}}_{k,h+1} - \mathbf{V}_{h+1}^\star) - \left(\mathbf{E}_{k,h}(x,a^*) + p(x,a^*)^\top(\overline{\mathbf{V}}_{k,h+1} - \mathbf{V}_{h+1}^\star)\right) \\
&= \mathbf{E}_{k,h}(x,a) - \mathbf{E}_{k,h}(x,a^*) + (p(x,a) - p(x,a^*))^\top(\overline{\mathbf{V}}_{k,h} - \mathbf{V}_{h+1}^\star) \\
&\leq \mathbf{E}_{k,h}(x,a) + (p(x,a) - p(x,a^*))^\top(\overline{\mathbf{V}}_{k,h} - \mathbf{V}_{h+1}^\star) \qquad\qquad \text{(Strong Optimism)} \\
&\leq \mathbf{E}_{k,h}(x,a) + \boldsymbol{\alpha} p(x,a)^\top(\overline{\mathbf{V}}_{k,h} - \mathbf{V}_{h+1}^\star),
\end{aligned}
$$

where the lastar line uses the component-wise inequalityes $p(x,a) - p(x,a^\star) \leq \boldsymbol{\alpha} p(x,a)$ due to the fact that $a^\star$ witnesses the $\boldsymbol{\alpha}$ transition-suboptimality, and $\overline{\mathbf{V}}_{k,h} - \mathbf{V}_{h+1}^\star \geq 0$ due to optimism. $\qquad\square$

### D.3 Proof of Lemma D.3

*Proof.* For ease, we suppress the dependence of $\boldsymbol{\alpha}$ on $(x,a,h)$. By our fundamental gap bound (Lemma D.1) and then Lemma D.4, we have that

$$
\begin{aligned}
\text{gap}_h(x,a) &\leq \mathbf{E}_{k,h}(x,a) + \boldsymbol{\alpha} \cdot p(x,a)^\top(\overline{\mathbf{V}}_{k,h+1} - \mathbf{V}_{h+1}^{\pi_k}) \\
&\leq \ddot{\mathbf{E}}_{k,h}(x,a) + \boldsymbol{\alpha} \cdot p(x,a)^\top(\ddot{\mathbf{V}}_{k,h+1}^{\pi_k} - \mathbf{V}_{h+1}^{\pi_k}) + (H-h+1)\boldsymbol{\alpha}\epsilon_{\text{clip}} \\
&\leq \ddot{\mathbf{E}}_{k,h}(x,a) + \boldsymbol{\alpha} \cdot p(x,a)^\top(\ddot{\mathbf{V}}_{k,h+1}^{\pi_k} - \mathbf{V}_{h+1}^{\pi_k}) + \text{gap}_h(x,a)/2,
\end{aligned}
$$

where the inequality bounds $\boldsymbol{\alpha}(H-h+1)\epsilon_{\text{clip}} \leq \boldsymbol{\alpha}\text{gap}_{\min}/2 \leq \boldsymbol{\alpha} \cdot \text{gap}_h(x,a)/2 \leq \text{gap}_h(x,a)/2$. This yields

$$
\tfrac{1}{2}\text{gap}_h(x,a) \leq \ddot{\mathbf{E}}_{k,h}(x,a) + \boldsymbol{\alpha} \cdot p(x,a)^\top(\ddot{\mathbf{V}}_{k,h+1}^{\pi_k} - \mathbf{V}_{h+1}^{\pi_k}).
$$

Now, fix a constant $c \in (0,1]$ to be chosen later. Either we have that $\ddot{\mathbf{E}}_{k,h}(x,a) \geq \tfrac{c}{2}\text{gap}_h(x,a)$, or otherwise,

$$
\boldsymbol{\alpha} \cdot p(x,a)^\top(\ddot{\mathbf{V}}_{k,h+1}^{\pi_k} - \mathbf{V}_{h+1}^{\pi_k}) \geq (1-c)\frac{1}{2}\text{gap}_h(x,a) \geq \frac{1-c}{c}\ddot{\mathbf{E}}_{k,h}(x,a),
$$

which can be rearranged into

$$
\ddot{\mathbf{E}}_{k,h}(x,a) \leq \frac{c\boldsymbol{\alpha}}{1-c}p(x,a)^\top(\ddot{\mathbf{V}}_{k,h+1}^{\pi_k} - \mathbf{V}_{h+1}^{\pi_k}).
$$

Hence, we have

$$
\begin{aligned}
\ddot{\mathbf{E}}_{k,h}(x,a) &\leq \ddot{\mathbf{E}}_{k,h}(x,a)\mathbb{I}\left\{\ddot{\mathbf{E}}_{k,h}(x,a) \geq \tfrac{c}{2}\text{gap}_h(x,a)\right\} \\
&\quad + \frac{c\boldsymbol{\alpha}}{1-c}p(x,a)^\top(\ddot{\mathbf{V}}_{k,h+1}^{\pi_k} - \mathbf{V}_{h+1}^{\pi_k})\mathbb{I}\left\{\ddot{\mathbf{E}}_{k,h}(x,a) < \tfrac{c}{2}\text{gap}_h(x,a)\right\} \\
&\leq \ddot{\mathbf{E}}_{k,h}(x,a)\mathbb{I}\left\{\ddot{\mathbf{E}}_{k,h}(x,a) \geq \tfrac{c}{2}\text{gap}_h(x,a)\right\} + \frac{c\boldsymbol{\alpha}}{1-c}p(x,a)^\top(\ddot{\mathbf{V}}_{k,h+1}^{\pi_k} - \mathbf{V}_{h+1}^{\pi_k}),
\end{aligned}
$$

and thus,

$$
\begin{aligned}
\ddot{\mathbf{E}}_{k,h}(x,a) + p(x,a)^\top(\ddot{\mathbf{V}}_{k,h+1}^{\pi_k} - \mathbf{V}_{h+1}^{\pi_k}) &\leq \ddot{\mathbf{E}}_{k,h}(x,a)\mathbb{I}\left\{\ddot{\mathbf{E}}_{k,h}(x,a) \geq \tfrac{c}{2}\text{gap}_h(x,a)\right\} \\
&\quad + (1 + \frac{c\boldsymbol{\alpha}}{1-c})p(x,a)^\top(\ddot{\mathbf{V}}_{k,h+1}^{\pi_k} - \mathbf{V}_{h+1}^{\pi_k}).
\end{aligned}
$$

In particular, choosing $c = \tfrac{1}{2}\min\{1, (\boldsymbol{\alpha}H)^{-1}\}$, we have $(1 + \frac{c\boldsymbol{\alpha}}{1-c}) \leq 1 + \frac{1}{H}$, and

$$
\frac{1}{2} = \frac{(1 \wedge (\boldsymbol{\alpha}H)^{-1})}{4} = \frac{1}{4(\boldsymbol{\alpha}H \vee 1)},
$$

so that $\ddot{\mathbf{E}}_{k,h}(x,a)\mathbb{I}\left\{\ddot{\mathbf{E}}_{k,h}(x,a) \geq \tfrac{c}{2}\text{gap}_h(x,a)\right\} = \check{\mathbf{E}}_{k,h}(x,a)$. This concludes the proof. $\qquad\square$

# Part II

# StrongEuler **and its surpluses**

## E  The StrongEuler **Algorithm**

Before continuing, let us define a logarithmic factor we shall use throughout:

$$\mathbf{L}(u) := \sqrt{2\log(10M^2 \max\{u,1\}/\delta)}, \tag{15}$$

where we recall that $M = SAH \geq 2$. This section formally presents StrongEuler, which makes two subtle modification of the EULER algorithm of [16].

First, similar to [5, 6], StrongEuler refines the log factors in the bonuses to depend on the number of samples $n_k(x,a)$ via $\mathbf{L}(n_k(x,a)) \propto \log(Mn_k(x,a)/\delta)$, rather than the overall time $T = KH$ via $\mathbf{L}(n_k(x,a)) \propto \log(MT/\delta)$, which is necessary to ensure the optimal $\log T$ regret. Following [5, 6], our confidence bounds can be slightly refined using law-of-iterated logarithm bounds, but for simplicity we do not pursue this direction here.

Second, StrongEuler satisfies *strong optimism*. We remind the reader that strong optimism is not necessary to achieve gap dependent bounds, but can achieve sharper bounds for settings with simple transition dynamics like contextual bandits. The EULER algorithm, or its predecessors (e.g. [1]), would also achieve-gap dependent bounds due to our analysis. Moreover, running these algorithms with the refined $\log(Mn_k(x,a)/\delta)$ log factors would also yield $\log T$- asymptotic regret, whereas implementing $\log(MT/\delta)$ confidence intervals may yield asymptotic regret that scales as $\log^2 T$ (see Remark B.5).

The EULER algorithm proceeds by standard optimistic value iteration, with carefully chosen exploration bonuses, and keeps track of various variance-related quantities:

---

**Algorithm 1:** StrongEuler

---

1 **Input:**
2 **Initialized:** For each $a \in \mathcal{A}$ $x, x' \in \mathcal{S}$, $n_1(x,a) = 0$, $n_1(x' \mid x, a) = 0$, $\mathsf{rsum}_1 = 0$, $\mathsf{rsumsq}_1 = 0$,
   $\widehat{p}_1(x,a) = 0$, $\widehat{\mathrm{Var}}_1[R(x,a)] = 0$
3 **for** $k = 1, 2, \ldots$ **do**
4 $\quad \overline{\mathbf{V}}_{k,H+1} \leftarrow 0$
5 $\quad$ **for** $h = H, H-1, \ldots, 1$ **do**
6 $\quad\quad$ **for** $x \in \mathcal{S}$ **do**
7 $\quad\quad\quad$ **for** $a \in \mathcal{A}$ **do**
8 $\quad\quad\quad\quad$ Call ConstructBonuses.
9 $\quad\quad\quad\quad \overline{\mathbf{Q}}_{k,h}(x,a) \leftarrow \min\{H - h + 1, \widehat{r}(x,a) + \widehat{p}_{k,h}(x,a)^\top \overline{\mathbf{V}}_{k,h+1} +$
10 $\quad\quad\quad\quad\quad\quad\quad\quad\quad \mathbf{b}_{k,h}^{\mathrm{prob}}(x,a) + \mathbf{b}_k^{\mathrm{rw}}(x,a) + \mathbf{b}_{k,h}^{\mathrm{str}}(x,a)\}$
11 $\quad\quad\quad$ **end**
12 $\quad\quad\quad \pi_{k,h}(x) := \arg\max_a \overline{\mathbf{Q}}_{k,h}(x,a), \widehat{a} \leftarrow \pi_{k,h}(x)$
13 $\quad\quad\quad \overline{\mathbf{V}}_{k,h}(x) := \overline{\mathbf{Q}}_{k,h}(x,\widehat{a})$
14 $\quad\quad\quad \underline{\mathbf{V}}_{k,h}(x) =$
   $\quad\quad\quad\quad \max\{0, \widehat{r}(x,\widehat{a}) - \mathbf{b}_{k,h}^{\mathrm{rw}}(x,\widehat{a}) + \widehat{p}_{k,h}(x,\widehat{a})^\top \underline{\mathbf{V}}_{k,h+1} - \mathbf{b}_{k,h}^{\mathrm{prob}}(x,\widehat{a}) - \mathbf{b}_{k,h}^{\mathrm{str}}(x,\widehat{a})\}.$
15 $\quad\quad$ **end**
16 $\quad$ **end**
17 $\quad$ **Call** RolloutAndUpdate$(k)$.
18 **end**

---

The RolloutAndUpdate function (Algorithm 2 below) executes one trajectory according to the policy $\pi_k$, and records all count- and variance- data regarding the relevant rewards and transition probabilities. Finally, the bonuses are are defined in Algorithm 3.

---
**Algorithm 2:** RolloutAndUpdate($k$)
---
1 **Input:** Global current episode $k$, global counts and empirical probabilities. Initialize $k+1$-th episode counts: $n_{k+1}(\cdot,\cdot) \leftarrow n_k(\cdot,\cdot)$, $n_{k+1}(\cdot \mid \cdot,\cdot) \leftarrow n_k(\cdot \mid \cdot,\cdot)$, $\mathsf{rsum}_{k+1}(\cdot,\cdot) \leftarrow \mathsf{rsum}_k(\cdot,\cdot)$, $\mathsf{rsumsq}_{k+1}(\cdot,\cdot) \leftarrow \mathsf{rsumsq}_k(\cdot,\cdot)$.
2 **for** $h = 1,\ldots,H$ **do**
3      Observe state $x_h$, play $a_h = \pi_{k,h}(x_h)$, recieve reward $R$ and view next state $x_{h+1}$.
4      $n_k(x_h,a_h) \mathrel{+}= 1$, $n_k(x_{h'}|x_h,a_h) \mathrel{+}= 1$, $\mathsf{rsum}(x,a) \mathrel{+}= R$, $\mathsf{rsum}(x,a) \mathrel{+}= R^2$
5 **end**
6 **for** $a \in \mathcal{A}, x \in \mathcal{S}$ **do**
7      **for** $x' \in \mathcal{S}$ **do**
8          $\widehat{p}_{k+1}(x'|x,a) = \frac{n_k(x_{h'}|x_h,a_h)}{n_k(x_h,a_h)}$
9      **end**
10      $\overline{r}_{k+1}(x,a) = \frac{\mathsf{rsum}_{k+1}}{n_k(x_h,a_h)}$, $\widehat{\mathrm{Var}}_{k+1}[R(x,a)] = \frac{\mathsf{rsumsq}_{k+1}}{n_k(x_h,a_h)} - \overline{r}_{k+1}(x,a)^2$.
11 **end**
12 ,
---

---
**Algorithm 3:** ConstructBonuses
---
1 **Bonuses:**

$$\mathbf{b}_k^{\mathrm{rw}}(x,a) := 1 \wedge \left( \sqrt{\frac{2\widehat{\mathrm{Var}}_k[R(x,a)]\mathbf{L}(n_k(x,a))}{n_k(x,a)}} + \frac{8\mathbf{L}(n_k(x,a))}{3(n_k(x,a)-1)} \right) \tag{16}$$

$$\mathbf{b}_{k,h}^{\mathrm{prob}}(x,a) := H \wedge \left( \sqrt{\frac{2\mathrm{Var}_{\widehat{p}_k(x,a)}[\overline{\mathbf{V}}_{k,h+1}]\mathbf{L}(n_k(x,a))}{n_k(x,a)}} + \frac{8H\mathbf{L}(n_k(x,a))}{3(n_k(x,a)-1)} \right.$$
$$\left. + \sqrt{\frac{2\mathbf{L}(n_k(x,a))\|\underline{\mathbf{V}}_{k,h+1} - \overline{\mathbf{V}}_{k,h+1}\|_{2,\widehat{p}_k(x,a)}^2}{n_k(x,a)}} \right). \tag{17}$$

$$\mathbf{b}_{k,h}^{\mathrm{str}}(x,a) := \|\overline{\mathbf{V}}_{k,h+1} - \underline{\mathbf{V}}_{k,h+1}\|_{2,\widehat{p}_k(x,a)} \sqrt{\frac{S\mathbf{L}(n_k(x,a))}{n_k(x,a)}} + \frac{8}{3}\frac{SH\mathbf{L}(n_k(x,a))}{n_k(x,a)} \tag{18}$$
---

## F    Analysis of StrongEuler: Proof of Proposition B.4

Proposition B.4 requires demonstrating a lower bound on the surplus, $0 \le \mathbf{E}_{k,h}(x,a)$, thereby establishing strong optimism, as well as an upper bound on the surplus, which we shall use to analyze the same complexity. We address strong optimism first in the next subsection, and then the upper bound in the following subsection. Throughout, we will assume that a good event $\mathcal{A}^{\mathrm{conc}}$ holds. To keep the proofs modular, the event $\mathcal{A}^{\mathrm{conc}}$ will only appear as an assumption in the supporting lemmas used in Sections F.1 and F.2. Then, in Section F.3, we formally define $\mathcal{A}^{\mathrm{conc}}$ in terms of 6 constituent events, establish $\mathbb{P}[\mathcal{A}^{\mathrm{conc}}] \ge 1 - \frac{\delta}{2}$, and conclude with proofs of the supporting lemmas which rely on $\mathcal{A}^{\mathrm{conc}}$. We remark that many of the arguments in this section are similar to those from [16], with the main differences being strong optimism and the additional care paid to log-factors, necessary for $\log T$ regret. Again, recall the definition $\mathbf{L}(u) := \sqrt{2\log(10M^2 \max\{u,1\}/\delta)}$.

### F.1    Proof of Optimism

Here we establish the optimism of StrongEuler, and in particular, the bound $\mathbf{E}_{k,h}(x,a) \ge 0$.

**Proposition F.1.** *Under the good event $\mathcal{A}^{\mathrm{conc}}$,*

    *(a)* StrongEuler *is* optimistic*:* $\pi_{k,h}(x) = \arg\max_a \overline{\mathbf{Q}}_{k,h}(x,a)$*, where* $\overline{\mathbf{Q}}_{k,h}(x,a) \ge \mathbf{Q}_h^\star(x,a)$
    *for all* $h,x,a$*. In particular,* $\overline{\mathbf{V}}_{k,h}(x) \ge \mathbf{V}_h^\star(x)$ *for* $h \in [0:H]$*.*

*(b)* StrongEuler *is strongly optimistic* $\mathbf{E}_{k,h}(x,a) \ := \ \overline{\mathbf{Q}}_{k,h}(x,a) - r(x,a) - p(x,a)^{\top}\overline{\mathbf{V}}_{k,h+1}(x) \geq 0.$

*(c)* $\underline{\mathbf{V}}_{k,h} \leq \mathbf{V}_h^{\pi_k} \leq \mathbf{V}_h^{\star} \leq \overline{\mathbf{V}}_{k,h}$

*Proof.* The policy choice $\pi_{k,h}(x) = \arg\max_a \overline{\mathbf{Q}}_{k,h}(x,a)$ holds by definition of the algorithm. We now give the remainder of the argument by inducting backwards on $h$. For $h = H+1$, $\overline{\mathbf{V}}_{k,H+1} = \underline{\mathbf{V}}_{k,H+1} = \mathbf{V}_{k,H+1}^{\star} = \mathbf{V}_{h+1}^{\pi_k} = 0$. Now, suppose as an inductive hypothesis that $\overline{\mathbf{V}}_{k,h+1} \geq \mathbf{V}_{h+1}^{\star} \geq \mathbf{V}_{h+1}^{\pi_k} \geq \underline{\mathbf{V}}_{k,h+1}$, and $\mathbf{E}_{k,h+1}(x,a) \geq 0$ for all $x,a$.

First, we shall show that $\mathbf{E}_{k,h}(x,a) \geq 0$ for all $x,a$. This will establish the induction for point *b*. It also establishes *(a)*, since then $\overline{\mathbf{Q}}_{k,h}(x,a) \geq r(x,a) + p(x,a)^{\top}\overline{\mathbf{V}}_{k,h+1}(x) \geq r(x,a) + p(x,a)^{\top}\mathbf{V}_{h+1}^{\star} = \mathbf{Q}_h^{\star}(x,a)$, proving optimism. To this end, note that

$$\mathbf{E}_{k,h}(x,a) := \overline{\mathbf{Q}}_{k,h}(x,a) - r(x,a) - p(x,a)^{\top}\overline{\mathbf{V}}_{k,h+1}(x)$$
$$:= \min\{H - h + 1, \widehat{r}(x,a) + \widehat{p}_{k,h}(x,a)^{\top}\overline{\mathbf{V}}_{k,h+1} + \mathbf{b}_{k,h}^{\mathrm{prob}}(x,a) + \mathbf{b}_k^{\mathrm{rw}}(x,a) + \mathbf{b}_{k,h}^{\mathrm{str}}(x,a)\}$$
$$- r(x,a) - p(x,a)^{\top}\overline{\mathbf{V}}_{k,h+1}(x).$$

Since $r(x,a) + p(x,a)^{\top}\overline{\mathbf{V}}_{k,h+1}(x) \leq H - h + 1$, it suffices to show that

$$\widehat{r}(x,a) + \widehat{p}_{k,h}(x,a)^{\top}\overline{\mathbf{V}}_{k,h+1} + \mathbf{b}_{k,h}^{\mathrm{prob}}(x,a) + \mathbf{b}_k^{\mathrm{rw}}(x,a) + \mathbf{b}_{k,h}^{\mathrm{str}}(x,a) - r(x,a) - p(x,a)^{\top}\overline{\mathbf{V}}_{k,h+1}(x) \geq 0.$$

Grouping the terms, it suffices to show that $\widehat{r}(x,a) - r(x,a) + \mathbf{b}_k^{\mathrm{rw}}(x,a) \geq 0$, and that

$$0 \leq (\widehat{p}_{k,h}(x,a)^{\top} - p(x,a))^{\top}\overline{\mathbf{V}}_{k,h+1}(x) + \mathbf{b}_{k,h}^{\mathrm{prob}}(x,a) + \mathbf{b}_{k,h}^{\mathrm{str}}(x,a)$$
$$= \left\{ (\widehat{p}_{k,h}(x,a)^{\top} - p(x,a))^{\top}\mathbf{V}_{h+1}^{\star}(x) + \mathbf{b}_{k,h}^{\mathrm{prob}}(x,a) \right\}$$
$$+ \left\{ \widehat{p}_{k,h}(x,a)^{\top} - p(x,a))^{\top}(\overline{\mathbf{V}}_{k,h+1}(x) - \mathbf{V}_{h+1}^{\star}(x)) + \mathbf{b}_{k,h}^{\mathrm{str}}(x,a) \right\}.$$

We lower bound $\widehat{r}(x,a) - r(x,a) + \mathbf{b}_k^{\mathrm{rw}}(x,a)$ and $(\widehat{p}_{k,h}(x,a)^{\top} - p(x,a))^{\top}\mathbf{V}_{h+1}^{\star}(x) + \mathbf{b}_{k,h}^{\mathrm{prob}}(x,a)$ by zero with the following lemma:

**Lemma F.2.** *On the good concentration event $\mathcal{A}^{\mathrm{conc}}$, it holds that*

$$|\widehat{r}(x,a) - r(x,a)| \leq \mathbf{b}_k^{\mathrm{rw}}(x,a),$$
$$|(\widehat{p}(x,a) - p(x,a))^{\top}\mathbf{V}_{h+1}^{\star}| \leq \mathbf{b}_{k,h}^{\mathrm{prob}}(x,a) \quad \textit{if } \underline{\mathbf{V}}_{k,h+1} \leq \mathbf{V}_{h+1}^{\star} \leq \overline{\mathbf{V}}_{k,h+1}$$

We conclude the proof of (b) with the following lemma, which lets us bound

$$(\widehat{p}_{k,h}(x,a)^{\top} - p(x,a))^{\top}(\overline{\mathbf{V}}_{k,h+1}(x) - \mathbf{V}_{h+1}^{\star}(x)) + \mathbf{b}_{k,h}^{\mathrm{str}}(x,a) \geq 0$$

Precisely we apply the following lemma with $V_2 = \overline{\mathbf{V}}_{k,h+1}$ and $V_1 = \mathbf{V}_{h+1}^{\star}$:

**Lemma F.3.** *Suppose that $\mathcal{A}^{\mathrm{prob}} \supset \mathcal{A}^{\mathrm{conc}}$ holds, and suppose that $V_1, V_2 : \mathcal{S} \to \mathbb{R}$ satisfies $\underline{\mathbf{V}}_{k,h+1} \leq V_1 \leq V_2 \leq \overline{\mathbf{V}}_{k,h+1}$. Then,*

$$\left| (\widehat{p}(x,a) - p(x,a))^{\top}(V_1 - V_2) \right| \leq \mathbf{b}_{k,h}^{\mathrm{str}}(x,a)$$

This finally establishes (b). We conclude by establishing (c). Here, we note that by definition $\mathbf{V}_h^{\pi_k} \leq \mathbf{V}_h^{\star}$, and $\mathbf{V}_h^{\star} \leq \overline{\mathbf{V}}_{k,h}$ as show above. Hence, it suffices to show $\underline{\mathbf{V}}_{k,h} \leq \mathbf{V}_h^{\pi_k}$. We begin with the inequality

$$\mathbf{V}_h^{\pi_k}(x) = p(x,a^{\star})^{\top}\mathbf{V}_{h+1}^{\pi_k} + r(x,a^{\star})$$
$$= \widehat{p}(x,a^{\star})^{\top}\mathbf{V}_{h+1}^{\pi_k} + \widehat{r}(x,a^{\star}) + (r(x,a^{\star}) - \widehat{r}(x,a^{\star})) + (p(x,a^{\star})^{\top} - \widehat{p}(x,a^{\star})^{\top})\mathbf{V}_{h+1}^{\pi_k}$$
$$= \widehat{p}(x,a^{\star})^{\top}\mathbf{V}_{h+1}^{\pi_k} + \widehat{r}(x,a^{\star}) + (r(x,a^{\star}) - \widehat{r}(x,a^{\star})) + (p(x,a^{\star})^{\top} - \widehat{p}(x,a^{\star})^{\top})\mathbf{V}_{h+1}^{\star}$$
$$+ (p(x,a^{\star}) - \widehat{p}(x,a^{\star}))^{\top}(\mathbf{V}_{h+1}^{\pi_k} - \mathbf{V}_{h+1}^{\star})$$
$$\geq \widehat{p}(x,a^{\star})^{\top}\mathbf{V}_{h+1}^{\pi_k} + \widehat{r}(x,a^{\star}) - \mathbf{b}_{k,h}^{\mathrm{rw}}(x,a) - \mathbf{b}_{k,h}^{\mathrm{prob}}(x,a) - \mathbf{b}_{k,h}^{\mathrm{str}}(x,a),$$

where the last inequality uses the bounds $(r(x, a^\star) - \widehat{r}(x, a^\star)) \geq -\mathbf{b}_{k,h}^{\mathrm{rw}}(x, a)$ and $(p(x, a^\star)^\top - \widehat{p}(x, a^\star)^\top)\mathbf{V}_{h+1}^\star \geq -\mathbf{b}_{k,h}^{\mathrm{prob}}(x, a)$ on $\mathcal{A}^{\mathrm{conc}}$ due to Lemma F.2, and bounds $(p(x, a^\star) - \widehat{p}(x, a^\star))^\top(\mathbf{V}_{h+1}^{\pi_k} - \mathbf{V}_{h+1}^\star) \geq -\mathbf{b}_{k,h}^{\mathrm{str}}(x, a)$ by applying Lemma F.3 with $V_1 = \mathbf{V}_{h+1}^{\pi_k}$ and $V_2 = \mathbf{V}_{h+1}^\star$, which satisfy $\underline{\mathbf{V}}_{k,h+1} \leq V_1 \leq V_2 \leq \overline{\mathbf{V}}_{k,h+1}$ by our inductive hypothesis (namely, $\overline{\mathbf{V}}_{k,h+1} \geq \mathbf{V}_{h+1}^\star \geq \mathbf{V}_{h+1}^{\pi_k} \geq \underline{\mathbf{V}}_{k,h+1}$). Since $\mathbf{V}_h^{\pi_k}(x) \geq 0$ as well, and since $\mathbf{V}_{h+1}^{\pi_k} \geq \underline{\mathbf{V}}_{k,h+1}$ by our inductive hypothesis, we therefore have

$$\mathbf{V}_h^{\pi_k}(x) \geq 0 \vee \widehat{p}(x, a^\star)^\top \underline{\mathbf{V}}_{k,h+1} + \widehat{r}(x, a^\star) - \mathbf{b}_{k,h}^{\mathrm{rw}}(x, a) - \mathbf{b}_{k,h}^{\mathrm{prob}}(x, a) - \mathbf{b}_{k,h}^{\mathrm{str}}(x, a) = \underline{\mathbf{V}}_{k,h}(x),$$

This completes the induction. $\qquad\square$

## F.2   Proof of Surplus Bound Upper Bound

Throughout, we assume the round $k$ is fixed, and suppress the dependence of $\widehat{p}$, $\widehat{\mathrm{Var}}$, and $\widehat{r}$ on $k$. We use the shorthand $p = p(x, a)$ and $\widehat{p} = \widehat{p}(x, a)$, where the pair $(x, a)$ are clear from context.

$$
\begin{aligned}
\mathbf{E}_{k,h}(x, a) &= \overline{\mathbf{V}}_{k,h}(x, a) - r(x, a) - p(x, a)^\top \overline{\mathbf{V}}_{k,h+1} \\
&\leq \mathbf{b}_k^{\mathrm{rw}}(x, a) + \widehat{r}(x, a) + \widehat{p}(x, a)^\top \overline{\mathbf{V}}_{k,h+1} + \mathbf{b}_{k,h}^{\mathrm{prob}}(x, a) - r(x, a) - p(x, a)^\top \overline{\mathbf{V}}_{k,h+1} \\
&= (\widehat{r}(x, a) - r(x, a) + (\widehat{p}(x, a) - p(x, a))^\top \mathbf{V}_{h+1}^\star \\
&\quad + \mathbf{b}_k^{\mathrm{rw}}(x, a) + \mathbf{b}_{k,h}^{\mathrm{prob}}(x, a) + (\widehat{p} - p)^\top (\overline{\mathbf{V}}_{k,t+1} - \mathbf{V}_{h+1}^\star) \\
&\leq 2\mathbf{b}_k^{\mathrm{rw}}(x, a) + 2\mathbf{b}_{k,h}^{\mathrm{prob}}(x, a) + \mathbf{b}_{k,h}^{\mathrm{str}}(x, a).
\end{aligned}
$$

where the last line is by Lemmas F.2 and F.3. Next, we state a standard lemma that lets us swap out the empirical variance for the true variance in upper bounding $\mathbf{b}_k^{\mathrm{rw}}(x, a)$:

**Lemma F.4.** *Under the event* $\mathcal{A}^{\mathrm{conc}}$, $\mathbf{b}_k^{\mathrm{rw}}(x, a) \lesssim \sqrt{\frac{\mathrm{Var}[R(x,a)]\mathbf{L}(n_k(x,a))}{n_k(x,a)}} + \frac{\mathbf{L}(n_k(x,a))}{n_k(x,a)}.$

Next, we recall from the definition of $\mathbf{b}^{\mathrm{prob}}$,

$$\mathbf{b}_{k,h}^{\mathrm{prob}}(x, a) \lesssim \sqrt{\frac{\mathrm{Var}_{p(x,a)}[\mathbf{V}_{h+1}^\star]\mathbf{L}(n_k(x,a))}{n_k(x,a)}} + \frac{H\mathbf{L}(n_k(x,a))}{n_k(x,a)} + \sqrt{\frac{\mathbf{L}(n_k(x,a))\|\underline{\mathbf{V}}_{k,h+1} - \overline{\mathbf{V}}_{k,h+1}\|_{2,\widehat{p}}^2}{n_k(x,a)}}.$$

where we replaced $n_k(x, a) - 1$ by $n_k(x, a)$ in the deminator of one of the terms by taking advantage of the '$H\wedge$'. Furthermore, we can bound

$$
\begin{aligned}
\sqrt{\frac{\mathrm{Var}_{p(x,a)}[\mathbf{V}_{h+1}^\star]\mathbf{L}(n_k(x,a))}{n_k(x,a)}} &\leq \sqrt{\frac{\min\{\mathrm{Var}_{p(x,a)}[\mathbf{V}_{h+1}^\star], \mathrm{Var}_{p(x,a)}[\mathbf{V}_{h+1}^{\pi_k}]\}\mathbf{L}(n_k(x,a))}{n_k(x,a)}} \\
&\quad + \left|\sqrt{|\mathrm{Var}_{p(x,a)}[\mathbf{V}_{h+1}^\star]|} - \sqrt{\mathrm{Var}_{p(x,a)}[\mathbf{V}_{h+1}^{\pi_k}]}\right|\sqrt{\frac{\mathbf{L}(n_k(x,a))}{n_k(x,a)}}.
\end{aligned}
$$

We can control the difference $|\sqrt{|\mathrm{Var}_{p(x,a)}[\mathbf{V}_{h+1}^\star]} - \sqrt{\mathrm{Var}_{p(x,a)}[\mathbf{V}_{h+1}^{\pi_k}]}|$ using the following lemma:

**Lemma F.5.** *Let* $X, Y$ *be two real valued random variables, and let* $\|\cdot\|_{p,2} := \sqrt{\mathbb{E}[(\cdot)^2]}$. *Then* $|\sqrt{\mathrm{Var}[X]} - \sqrt{\mathrm{Var}[Y]}| \leq \sqrt{\mathrm{Var}[X - Y]} \leq \|X - Y\|_{2,p}$.

*Proof.* The inequality $\mathrm{Var}[X - Y] \leq \mathbb{E}[(X - Y)^2] = \|X - Y\|_{2,p}^2$ follows since $\mathrm{Var}[Z] \leq \mathbb{E}[Z^2]$ for any random variable $Z$. For the first inequality, we can assume WLOG that $X, Y$ are mean zero, in which case $\sqrt{\mathrm{Var}[X]} = \|X\|_{2,p}$, and similarly for $Y$ and $X - Y$. The result now follows from the fact that the norm $\|\cdot\|_{p,2}$ satisfies the triangle inequality. $\qquad\square$

We shall also need the following simple fact:

**Fact F.6.** *If* $V_1(x) \leq V_2(x) \leq V_3(x) \leq V_4(x)$ *for all* $x \in \mathcal{S}$, *then* $\|V_2 - V_3\|_{2,p} \leq \|V_1 - V_4\|_{2,p}$.

Since $\underline{\mathbf{V}}_{k,h+1} \le \mathbf{V}_{h+1}^{\pi_k} \le \mathbf{V}_{h+1}^{\star} \le \overline{\mathbf{V}}_{k,h+1}$ by Proposition F.1, Lemma F.5 and Fact F.6 above yield

$$|\sqrt{\mathrm{Var}_{p(x,a)}[\mathbf{V}_{h+1}^{\star}]} - \sqrt{\mathrm{Var}_{p(x,a)}[\mathbf{V}_{h+1}^{\pi_k}]}| \le \|\mathbf{V}_{h+1}^{\star} - \mathbf{V}_{h+1}^{\pi_k}\|_{2,p} \le \|\overline{\mathbf{V}}_{k,h+1} - \underline{\mathbf{V}}_{k,h+1}\|_{2,p}.$$

Together the with the elementary inequality, $\sqrt{a+b} \le \sqrt{a} + \sqrt{b} \lesssim \sqrt{a+b}$, this in turn yields

$$\mathbf{b}_{k,h}^{\mathrm{prob}}(x,a) + \mathbf{b}_{k,h}^{\mathrm{rw}}(x,a)$$

$$\lesssim \mathbf{b}_{k,h}^{\mathrm{rw}}(x,a) + \sqrt{\frac{\min\{\mathrm{Var}_{p(x,a)}[\mathbf{V}_{h+1}^{\star}], \mathrm{Var}_{p(x,a)}[\mathbf{V}_{h+1}^{\pi_k}]\}\mathbf{L}(n_k(x,a))}{n_k(x,a)}} + \frac{H\mathbf{L}(n_k(x,a))}{n_k(x,a)}$$

$$+ \sqrt{\frac{\mathbf{L}(n_k(x,a))\|\underline{\mathbf{V}}_{k,h+1} - \overline{\mathbf{V}}_{k,h+1}\|_{2,\widehat{p}+p}^2}{n_k(x,a)}}$$

$$\lesssim \sqrt{\frac{\mathrm{Var}[R(x,a)] + \min\{\mathrm{Var}_{p(x,a)}[\mathbf{V}_{h+1}^{\star}], \mathrm{Var}_{p(x,a)}[\mathbf{V}_{h+1}^{\pi_k}]\}\mathbf{L}(n_k(x,a))}{n_k(x,a)}} + \frac{H\mathbf{L}(n_k(x,a))}{n_k(x,a)}$$

$$+ \sqrt{\frac{\mathbf{L}(n_k(x,a))\|\underline{\mathbf{V}}_{k,h+1} - \overline{\mathbf{V}}_{k,h+1}\|_{2,\widehat{p}+p}^2}{n_k(x,a)}},$$

$$\lesssim \sqrt{\frac{\mathtt{Var}_{h,x,a}^{(k)}\mathbf{L}(n_k(x,a))}{n_k(x,a)}} + \frac{H\mathbf{L}(n_k(x,a))}{n_k(x,a)} + \sqrt{\frac{\mathbf{L}(n_k(x,a))\|\underline{\mathbf{V}}_{k,h+1} - \overline{\mathbf{V}}_{k,h+1}\|_{2,\widehat{p}+p}^2}{n_k(x,a)}},$$

where we use the shorthand $\|V\|_{2,\widehat{p}+p} = \sqrt{\|V\|_{2,p}^2 + \|V\|_{2,\widehat{p}}^2}$, and where in the last, we recall that $\mathtt{Var}_{h,x,a}^{(k)} = \min\{\mathtt{Var}_{h,x,a}^{\pi_k}, \mathtt{Var}_{h,x,a}^{\star}\} = \mathrm{Var}[R(x,a)] + \min\{\mathrm{Var}_{p(x,a)}[\mathbf{V}_{h+1}^{\star}], \mathrm{Var}_{p(x,a)}[\mathbf{V}_{h+1}^{\pi_k}]\}$.

Next, substituing in $\mathbf{b}_{k,h}^{\mathrm{str}}(x,a) := \|\overline{\mathbf{V}}_{k,h+1} - \underline{\mathbf{V}}_{k,h+1}\|_{2,\widehat{p}(x,a)}\sqrt{\frac{S\mathbf{L}(n_k(x,a))}{n_k(x,a)}} + \frac{8}{3}\frac{SH\mathbf{L}(n_k(x,a))}{n_k(x,a)}$, we obtain

$$\mathbf{b}_k^{\mathrm{rw}}(x,a) + \mathbf{b}_{k,h}^{\mathrm{prob}}(x,a) + \mathbf{b}_{k,h}^{\mathrm{str}}(x,a)$$

$$\lesssim \sqrt{\frac{\mathtt{Var}_{h,x,a}^{(k)}\mathbf{L}(n_k(x,a))}{n_k(x,a)}} + \frac{H\mathbf{L}(n_k(x,a))}{n_k(x,a)}$$

$$+ \sqrt{\frac{\mathbf{L}(n_k(x,a))\|\underline{\mathbf{V}}_{k,h+1} - \overline{\mathbf{V}}_{k,h+1}\|_{2,\widehat{p}+p}^2}{n_k(x,a)}} + \sqrt{\frac{S\|\overline{\mathbf{V}}_{k,h+1} - \underline{\mathbf{V}}_{k,h+1}\|_{2,\widehat{p}}^2\mathbf{L}(n_k(x,a))}{n_k(x,a)}} + \frac{SH\mathbf{L}(n_k(x,a))}{n_k(x,a)}$$

$$\lesssim \sqrt{\frac{\mathtt{Var}_{h,x,a}^{(k)}\mathbf{L}(n_k(x,a))}{n_k(x,a)}} + \frac{SH\mathbf{L}(n_k(x,a))}{n_k(x,a)} + \sqrt{\frac{S\|\overline{\mathbf{V}}_{k,h+1} - \underline{\mathbf{V}}_{k,h+1}\|_{2,p+\widehat{p}}^2\mathbf{L}(n_k(x,a))}{n_k(x,a)}}$$

$$\tag{19}$$

$$\overset{(i)}{\le} \sqrt{\frac{\mathtt{Var}_{h,x,a}^{(k)}\mathbf{L}(n_k(x,a))}{n_k(x,a)}} + \frac{SH\mathbf{L}(n_k(x,a))}{n_k(x,a)} + \frac{S\mathbf{L}(n_k(x,a))}{n_k(x,a)} + \|\overline{\mathbf{V}}_{k,h+1} - \underline{\mathbf{V}}_{k,h+1}\|_{2,\widehat{p}+p}^2$$

$$\overset{(ii)}{=} \sqrt{\frac{\mathtt{Var}_{h,x,a}^{(k)}\mathbf{L}(n_k(x,a))}{n_k(x,a)}} + \frac{SH\mathbf{L}(n_k(x,a))}{n_k(x,a)} + 2\|\overline{\mathbf{V}}_{k,h+1} - \underline{\mathbf{V}}_{k,h+1}\|_{2,p}^2 + (\widehat{p}-p)(\overline{\mathbf{V}}_{k,h+1} - \underline{\mathbf{V}}_{k,h+1})^2$$

$$\overset{(iii)}{\le} \sqrt{\frac{\mathtt{Var}_{h,x,a}^{(k)}\mathbf{L}(n_k(x,a))}{n_k(x,a)}} + \frac{SH\mathbf{L}(n_k(x,a))}{n_k(x,a)} + 2\|\overline{\mathbf{V}}_{k,h+1} - \underline{\mathbf{V}}_{k,h+1}\|_{2,p}^2 + H(p-\widehat{p})^\top(\overline{\mathbf{V}}_{k,h+1} - \underline{\mathbf{V}}_{k,h+1}),$$

where $(i)$ uses the inequality $a/b \le a^2 + \frac{1}{b^2}$, and $(ii)$ uses the facts that $\|V\|_{2,\widehat{p}+p}^2 = \|V\|_{2,p}^2 + \|V\|_{2,\widehat{p}}^2$ and $\|V\|_{2,\widehat{p}}^2 = \langle \widehat{p}, V^2 \rangle = \langle \widehat{p}, V^2 \rangle + \langle \widehat{p} - p, V^2 \rangle = \|V\|_{2,\widehat{p}}^2 + \langle \widehat{p} - p, V^2 \rangle$. Lastly, inequality $(iii)$ uses $0 \le \underline{\mathbf{V}}_{k,h+1} \le \overline{\mathbf{V}}_{k,h+1} \le H$.

We continue bounding $H(p-\widehat{p})^\top(\overline{\mathbf{V}}_{k,h+1} - \underline{\mathbf{V}}_{k,h+1})$ in much the same way that we bounded the term in Lemma F.3 in terms of $\mathbf{b}^{\mathrm{str}}$, with the exception that we seek a term which depends on the true transition probability $p(x,a)$, and not the empirical $\widehat{p}(x,a)$:

**Lemma F.7.** *Under $\mathcal{A}^{\mathrm{conc}}$,*

$$\left|(\widehat{p}(x,a) - p(x,a))^\top (\overline{\mathbf{V}}_{k,h+1} - \underline{\mathbf{V}}_{k,h+1})\right| \lesssim \|\overline{\mathbf{V}}_{k,h+1} - \underline{\mathbf{V}}_{k,h+1}\|_{2,p(x,a)} \sqrt{\frac{S\mathbf{L}(n_k(x,a))}{n_k(x,a)}} + \frac{SH\mathbf{L}(n_k(x,a))}{n_k(x,a)}.$$

The proof of the above lemma is ommitted for the sake of brevity, and follows from a simplified version of the proof of Lemma F.3 where we need not pass through an empirical variance. Applying the bound in Lemma F.7, we have

$$H(p - \widehat{p})^\top(\overline{\mathbf{V}}_{k,h+1} - \underline{\mathbf{V}}_{k,h+1}) \lesssim H\|\overline{\mathbf{V}}_{k,h+1} - \underline{\mathbf{V}}_{k,h+1}\|_{2,p}\sqrt{\frac{S\mathbf{L}(n_k(x,a))}{n_k(x,a)}} + \frac{SH^2\mathbf{L}(n_k(x,a))}{n_k(x,a)}$$

$$\lesssim \|\overline{\mathbf{V}}_{k,h+1} - \underline{\mathbf{V}}_{k,h+1}\|_{2,p}^2 + \frac{H^2 S\mathbf{L}(n_k(x,a))}{n_k(x,a)} + \frac{SH^2\mathbf{L}(n_k(x,a))}{n_k(x,a)},$$

where the last line uses the inequality $ab \le (a^2 + b^2)/2$. Finally, combining the above with our previous bound, we arrive at

$$\mathbf{E}_{k,h}(x,a) \lesssim \mathbf{b}_k^{\mathrm{rw}}(x,a) + \mathbf{b}_{k,h}^{\mathrm{prob}}(x,a) + \mathbf{b}_{k,h}^{\mathrm{str}}(x,a)$$

$$\lesssim \sqrt{\frac{\mathrm{Var}_{h,x,a}^{(k)}\mathbf{L}(n_k(x,a))}{n_k(x,a)}} + \frac{SH^2\mathbf{L}(n_k(x,a))}{n_k(x,a)} + \|\overline{\mathbf{V}}_{k,h+1} - \underline{\mathbf{V}}_{k,h+1}\|_{2,p(x,a)}^2.$$

From first principles, it is straightforward to show that $\mathbf{E}_{k,h}(x,a) \lesssim H$, which implies that

$$\mathbf{E}_{k,h}(x,a) \lesssim H \wedge \sqrt{\frac{\mathrm{Var}_{h,x,a}^{(k)}\mathbf{L}(n_k(x,a))}{n_k(x,a)}} \tag{20}$$

$$+ \left(H \wedge \frac{SH^2\mathbf{L}(n_k(x,a))}{n_k(x,a)}\right) + \|\overline{\mathbf{V}}_{k,h+1} - \underline{\mathbf{V}}_{k,h+1}\|_{2,p(x,a)}^2. \tag{21}$$

To conclude the proof, it remains to unravel the term $\|\overline{\mathbf{V}}_{k,h+1} - \underline{\mathbf{V}}_{k,h+1}\|_{2,p(x,a)}^2$.

**Lemma F.8.** *Define the term*

$$\mathbf{Z}_k(x,a) = H^2 \wedge H^2 \left(\sqrt{\frac{S\mathbf{L}(n_k(x,a))}{n_k(x,a)}} + \frac{S\mathbf{L}(n_k(x,a))}{n_k(x,a)}\right)^2,$$

*Then, we have the bound*

$$\overline{\mathbf{V}}_{k,h}(x) - \underline{\mathbf{V}}_{k,h}(x) \lesssim \mathbb{E}^{\pi_k}\left[\sum_{t=h}^H \sqrt{\mathbf{Z}(x_t,a_t)} \mid x_h = x\right]. \tag{22}$$

As a consequence, we can compute

$$(\overline{\mathbf{V}}_{k,h}(x) - \underline{\mathbf{V}}_{k,h}(x))^2 \lesssim \mathbb{E}^{\pi_k}\left[\left(\sum_{t=h}^H \sqrt{\mathbf{Z}_k(x_t,a_t)}\right)^2 \mid x_h = x\right] \le H\mathbb{E}^{\pi_k}\left[\sum_{t=h}^H \mathbf{Z}_k(x_t,a_t) \mid x_h = x\right].$$

Hence, we have

$$\|\overline{\mathbf{V}}_{k,h+1} - \underline{\mathbf{V}}_{k,h+1}\|_{2,p}^2 = \mathbb{E}_{x' \sim p(x,a)}(\overline{\mathbf{V}}_{k,h+1}(x) - \underline{\mathbf{V}}_{k,h+1}(x))^2$$

$$\lesssim H\mathbb{E}^{\pi_k}\left[\sum_{t=h+1}^H \mathbf{Z}_k(x_t,a_t) \mid (x_h,a_h) = (x,a)\right].$$

Since $H\mathbf{Z}_k(x,a) \ge H \wedge \frac{SH^2\mathbf{L}(n_k(x,a))}{n_k(x,a)}$. Hence, we can bound via (20)

$$\mathbf{E}_{k,h}(x,a) \lesssim \underbrace{H \wedge \sqrt{\frac{\mathrm{Var}_{h,x,a}^{(k)}\mathbf{L}(n_k(x,a))}{n_k(x,a)}}}_{:=\mathbf{B}_{k,h}^{\mathrm{lead}}(x,a)} + \mathbb{E}^{\pi_k}\left[\sum_{t=h}^H \underbrace{H\mathbf{Z}_k(x_t,a_t)}_{:=\mathbf{B}_k^{\mathrm{fut}}(x_t,a_t)} \mid (x_h,a_h) = (x,a)\right]$$

where we note that the summation in the expectation now begins at $t = h$ to account for the term $H \wedge \frac{SH^2\mathbf{L}(n_k(x,a))}{n_k(x,a)}$ in (20), and recal that

$$\mathbf{B}_k^{\mathrm{fut}}(x,a) := H^3 \wedge H^3 \left( \sqrt{\frac{S\mathbf{L}(n_k(x,a))}{n_k(x,a)}} + \frac{S\mathbf{L}(n_k(x,a))}{n_k(x,a)} \right)^2 = H\mathbf{Z}_k(x,a)$$

To conclude, we recall the definitions,

$$\mathbf{L}(u) := \sqrt{2\log(10M^2\max\{u,1\}/\delta)}.$$

so that, for $u \geq 1$, $\mathbf{L}(u) \lesssim \log\frac{Mu}{\delta}$.

### F.3 Definition of $\mathcal{A}^{\mathrm{conc}}$, and proofs of supporting lemmas

Before proving the lemmas above, we formally express the good event $\mathcal{A}^{\mathrm{conc}}$ as a list of constituent concentration events, and verify that it occurs with probability at least $1 - \delta/2$:

**Proposition F.9.** *The event* $\mathcal{A}^{\mathrm{conc}} := \mathcal{A}^{\mathrm{rw}} \cap \mathcal{A}^{\mathrm{prob}} \cap \mathcal{A}^{\mathrm{val}} \cap \mathcal{A}^{\mathrm{var,val}} \cap \mathcal{A}^{\mathrm{var,rw}}$ *occurs with probability* $1 - \delta/2$, *where each of the constituent events occurs with probability at least* $1 - \delta/12$:

$$\mathcal{A}^{\mathrm{rw}} := \left\{ \forall k,x,a,h : |\widehat{r}_k(x,a) - r(x,a)| \leq \sqrt{\mathrm{Var}[R(x,a)]\frac{2\mathbf{L}(n_k(x,a))}{n_k(x,a)}} + \frac{2\mathbf{L}(n_k(x,a))}{3n_k(x,a)} \right\}$$

$$\mathcal{A}^{\mathrm{prob}} := \left\{ \forall k,x,x',a,h : |\widehat{p}(x' \mid x,a) - p(x' \mid x,a)| \leq \sqrt{p(x' \mid x,a)(1 - p(x' \mid x,a))\frac{2\mathbf{L}(n_k(x,a))}{n_k(x,a)}} + \frac{2\mathbf{L}(n_k(x,a))}{3n_k(x,a)} \right\}$$

$$\mathcal{A}^{\mathrm{val}} := \left\{ \forall k,x,a,h : |(\widehat{p}(x,a) - p(x,a))^\top \mathbf{V}_{h+1}^\star| \leq \sqrt{\mathrm{Var}_{p(x,a)}[\mathbf{V}_{h+1}^\star]\frac{2\mathbf{L}(n_k(x,a))}{n_k(x,a)}} + \frac{2H\mathbf{L}(n_k(x,a))}{3n_k(x,a)} \right\}$$

$$\mathcal{A}^{\mathrm{var,prob}} := \left\{ \forall k,h,x,a : |\widehat{p}(x' \mid x,a) - p(x' \mid x,a)| \leq \sqrt{\frac{2\mathbf{L}(n_k(x,a))}{n_k(x,a)}} \right\}.$$

$$\mathcal{A}^{\mathrm{var,val}} := \left\{ \forall k,h,x,a : \left| \|\mathbf{V}_h^\star\|_{2,\widehat{p}(x,a)} - \|\mathbf{V}_h^\star\|_{2,p(x,a)} \right| \leq H\sqrt{\frac{2\mathbf{L}n_k(x,a))}{n_k(x,a) - 1}} \right\}$$

$$\mathcal{A}^{\mathrm{var,rw}} := \left\{ \forall k,h,x,a : \left| \sqrt{\widehat{\mathrm{Var}}(R(x,a))} - \sqrt{\mathrm{Var}(R(x,a))} \right| \leq \sqrt{\frac{2\mathbf{L}(n_k(x,a))}{n_k(x,a) - 1}} \right\}$$

*Proof.* The proof of these the first four events follows from standard applications of Bernstein's and Hoeffding's inequality, and the last two from Theorem 10 in [10]. Similar proofs can be found in [16, 1, 5]. As in those works, the only subtlety is to use the appropriate concentration inequality with respect to an appropriate filtration to attain bounds that depend on $\mathbf{L}(n_k(x,a))$, rather than on $\mathbf{L}(T)$.

Let's prove $\mathcal{A}^{\mathrm{rw}}$ as an example. We it suffices to only consider rounds for which $n_k(x,a) \geq 1$, for otherwise the bound is vacuous. Fix an action $(x,a)$, and let $\tau_i \in \{1,2,\dots\} \cup \{\infty\}$ denote the round $k + 1$ immediately after the $i$-th round $k$ at which a pair $(x,a)$ is observed at least once during the rollout, and define a sub-filtration $\{\mathcal{G}_i\}$ via $\mathcal{G}_i = \mathcal{F}_{\tau_i}$. Then, for any given $i$, a martingale analogue of Bernstein's inequality yields

$$\mathbb{P}\left[ |\widehat{r}_{\tau_i}(x,a) - r(x,a)|\mathbb{I}(\tau_n < \infty) \geq \sqrt{\frac{2\mathrm{Var}[R(x,a)]\log(2/\eta)}{n_{\tau_i(x,a)}}} + \frac{2\log(2/\eta)}{3n_{\tau_i(x,a)}} \right] \leq \eta.$$

Now fix an $i \geq 1$. Since $\widehat{r}_k(x,a)$ and $n_k(x,a)$ are constant for $k \in \{\tau_i, \dots, \tau_{i+1} - 1\}$, we have

$$\forall n, \ \mathbb{P}\left[ \exists k \in \{\tau_i, \dots, \tau_{i+1} - 1\} : |\widehat{r}_k(x,a) - r(x,a)| \geq \sqrt{\frac{2\mathrm{Var}[R(x,a)]\log(2/\eta)}{n_k(x,a)}} + \frac{2\log(2/\eta)}{3n_k(x,a)} \right] \leq \eta,$$

Applying the above with $\eta \leftarrow 2\eta/i^2$ and union bounding over $n$, we have

$$\mathbb{P}\left[\exists i, k : k \in \{\tau_i, \ldots, \tau_{i+1} - 1\}, |\widehat{r}_k(x,a) - r(x,a)| \geq \sqrt{\frac{2\mathrm{Var}[R(x,a)]\log(4i^2/\eta)}{n_k(x,a)}} + \frac{2\log(4i^2/\eta)}{3n_k(x,a)}\right] \leq \eta.$$

Since $n_k(x,a)$ increments by at least one for each $\tau_i$, we have $i \leq n_k(x,a)$ for $k \in \{\tau_i, \ldots, \tau_{i+1} - 1\}$. Thus,

$$\mathbb{P}\left[\exists i, k : k \in \{\tau_i, \ldots, \tau_{i+1} - 1\}, |\widehat{r}_k(x,a) - r(x,a)| \geq \sqrt{\frac{2\mathrm{Var}[R(x,a)]\log(4n_k(x,a)^2/\eta)}{n_k(x,a)}} + \frac{2\log(4n_k(x,a)^2/\eta)}{3n_k(x,a)}\right] \leq \eta.$$

Lastly, since for any $k$, there always exist some $i$ for which $k \in \{\tau_i, \ldots, \tau_{i+1} - 1\}$, we have

$$\mathbb{P}\left[k : |\widehat{r}_k(x,a) - r(x,a)| \geq \sqrt{\frac{2\mathrm{Var}[R(x,a)]\log(4n_k(x,a)^2/\eta)}{n_k(x,a)}} + \frac{2\log(4n_k(x,a)^2/\eta)}{3n_k(x,a)}\right] \leq \eta.$$

We then conclude by union bounding over $SA$, and letting $\eta = \delta/12SA$, yielding the following log factor: $\log(48SAn_k(x,a)^2/\delta) \leq \mathbf{L}(n_k(x,a))$, where we recall $\mathbf{L}(u) = \sqrt{2\log(10M^2\max\{u,1\}/\delta)}$ for $M = SAH$. The proof for $\mathcal{A}^{\mathrm{prob}}$ is analogous, the proof for $\mathcal{A}^{\mathrm{val}}$ requires union bounding over states $x'$, incuring a log factor $\log(4S^2An_k(x,a)^2/\delta) \leq \mathbf{L}(n_k(x,a))$. □

*Proof of Lemma F.2.* We prove the bound $|(\widehat{p}(x,a) - p(x,a))^\top \mathbf{V}^\star_{h+1}| \leq \mathbf{b}^{\mathrm{prob}}_{k,h}(x,a)$; the analogous bounds for rewards is similar. Note that since $\widehat{p}(x,a)^\top \mathbf{V}^\star_{h+1} \in [0, H]$ and $p(x,a)^\top \mathbf{V}^\star_{h+1} \in [0, H]$, $|(\widehat{p}(x,a) - p(x,a))^\top \mathbf{V}^\star_{h+1}| \in [0, H]$. This takes care of the first '$H\wedge$' in $\mathbf{b}^{\mathrm{prob}}_{k,h}(x,a)$. Next, on $\mathcal{A}^{\mathrm{val}}$ and $\mathcal{A}^{\mathrm{var,val}}$,

$$\begin{aligned}
&|(\widehat{p}(x,a) - p(x,a))^\top \mathbf{V}^\star_{h+1}| \\
&\leq \sqrt{\frac{2\mathrm{Var}_{p(x,a)}[\mathbf{V}^\star_{h+1}]\mathbf{L}(n_k(x,a))}{n_k(x,a)}} + \frac{2H\mathbf{L}(n_k(x,a))}{3n_k(x,a)} \qquad\qquad (\text{on } \mathcal{A}^{\mathrm{val}}) \\
&\leq \sqrt{\frac{2\mathrm{Var}_{\widehat{p}(x,a)}[\mathbf{V}^\star_{h+1}]\mathbf{L}(n_k(x,a))}{n_k(x,a)}} + \frac{8H\mathbf{L}(n_k(x,a))}{3(n_k(x,a) - 1)} \qquad\qquad (\text{on } \mathcal{A}^{\mathrm{var,val}}) \\
&= \sqrt{\frac{2\mathrm{Var}_{\widehat{p}(x,a)}[\overline{\mathbf{V}}_{k,h+1}]\mathbf{L}(n_k(x,a))}{n_k(x,a)}} + \frac{8H\mathbf{L}(n_k(x,a))}{3(n_k(x,a) - 1)} \\
&\quad + \left(\sqrt{\mathrm{Var}_{\widehat{p}(x,a)}[\mathbf{V}^\star_{h+1}]} - \sqrt{\mathrm{Var}_{\widehat{p}(x,a)}[\widehat{p}(x,a)^\top \overline{\mathbf{V}}_{k,h+1}]}\right)\sqrt{\frac{2\mathbf{L}(n_k(x,a))}{n_k(x,a)}}.
\end{aligned}$$

Lastly, by Lemma F.5, we have the bound

$$\left|\sqrt{\mathrm{Var}_{\widehat{p}(x,a)}[\mathbf{V}^\star_{h+1}]} - \sqrt{\mathrm{Var}_{\widehat{p}(x,a)}[\overline{\mathbf{V}}_{k,h+1}]}\right| \leq \|\underline{\mathbf{V}}_{k,h+1} - \overline{\mathbf{V}}_{k,h+1}\|_{2,\widehat{p}(x,a)}.$$

□

*Proof of Lemma F.3.* Summing up the condition of event $\mathcal{A}^{\text{prob}}$ over states $x' \in \mathcal{S}$, and then applying event $\mathcal{A}^{\text{var,prob}}$ to control $|p(x'|x,a) - \widehat{p}(x'|x,a)|$:

$$
(\widehat{p}(x,a) - p(x,a))^\top (\overline{\mathbf{V}}_{k,h+1} - \mathbf{V}_{h+1}^\star) \leq \sum_{x'} \sqrt{2 \frac{\mathbf{L}(n_k(x,a)) p(x'|x,a)(1 - p(x'|x,a))}{n_k(x,a)}} |V_2(x') - V_1(x')|
$$
$$
+ \frac{2}{3} \sum_{x'} \frac{\mathbf{L}(n_k(x,a))}{n_k(x,a)} |V_2(x') - V_1(x')|,
$$
$$
\stackrel{(i)}{\leq} \sum_{x'} \sqrt{2 \frac{\mathbf{L}(n_k(x,a)) p(x'|x,a)}{n_k(x,a)}} |V_2(x') - V_1(x')|
$$
$$
+ \frac{2}{3} \sum_{x'} \frac{\mathbf{L}(n_k(x,a))}{n_k(x,a)} |V_2(x') - V_1(x')|,
$$
$$
\stackrel{(ii)}{\leq} \sum_{x'} \sqrt{2 \frac{\mathbf{L}(n_k(x,a)) \widehat{p}(x'|x,a)}{n_k(x,a)}} |V_2(x') - V_1(x')|
$$
$$
+ \underbrace{\frac{8}{3} \sum_{x'} \frac{\mathbf{L}(n_k(x,a))}{n_k(x,a)} |V_2(x') - V_1(x')|}_{\leq \frac{8}{3} \frac{HS\mathbf{L}(n_k(x,a))}{n_k(x,a)}},
$$

where $(i)$ uses $p(x'|x,a)(1 - p(x'|x,a)) \leq p(x'|x,a)$, $(ii)$ uses event $\mathcal{A}^{\text{var,prob}}$, and where bound in the bracket is because there $|\overline{\mathbf{V}}_{k,t+1}(x') - \mathbf{V}_{h+1}^\star(x')| \leq H$ by Proposition F.1 part (b), and there are at most $S$ terms in the summation. To bound the first term, we have

$$
\sum_{x'} \sqrt{2\mathbf{L}(n_k(x,a)) \frac{\widehat{p}(x'|x,a)}{n_k(x,a)}} |V_2(x') - V_1(x')|
$$
$$
= \sqrt{2 \frac{\mathbf{L}(n_k(x,a))}{n_k(x,a)}} \sum_{x'} \sqrt{\widehat{p}(x'|x,a)} |V_2(x') - V_1(x')|
$$
$$
\stackrel{(i)}{\leq} \sqrt{2 \frac{\mathbf{L}(n_k(x,a))}{n_k(x,a)}} \sqrt{S \|V_2 - V_1\|_{2,\widehat{p}}^2}
$$
$$
\stackrel{(ii)}{\leq} \|\overline{\mathbf{V}}_{k,h+1} - \underline{\mathbf{V}}_{k,h+1}\|_{2,\widehat{p}} \sqrt{2 \frac{S\mathbf{L}(n_k(x,a))}{n_k(x,a)}}
$$

$(i)$ bounds uses Cauchy-Schwartz, and $(ii)$ uses Proposition F.1 part (c) to bound $\|V_2 - V_1\|_{2,p} \leq \|\overline{\mathbf{V}}_{k,h+1} - \underline{\mathbf{V}}_{k,h+1}\|_{2,\widehat{p}}$ for $\underline{\mathbf{V}}_{k,h+1} \leq V_1 \leq V_2 \leq \overline{\mathbf{V}}_{k,h+1}$, in light of Fact F.6. $\qquad\square$

*Proof of Lemma F.4.* Under the event $\mathcal{A}^{\text{conc}}$ we have

$\mathbf{b}_k^{\text{rw}}(x,a)$
$$
\lesssim 1 \wedge \left( \sqrt{\frac{\widehat{\text{Var}}[R(x,a)]\mathbf{L}(n_k(x,a))}{n_k(x,a)}} + \frac{\mathbf{L}(n_k(x,a))}{n_k(x,a) - 1} \right) \qquad\qquad \text{(definition)}
$$
$$
\leq 1 \wedge \left( \sqrt{\frac{\text{Var}[R(x,a)]\mathbf{L}(n_k(x,a))}{n_k(x,a)}} + \left| \sqrt{\text{Var}[R(x,a)]} - \sqrt{\widehat{\text{Var}}[R(x,a)]} \right| \sqrt{\frac{\mathbf{L}(n_k(x,a))}{n_k(x,a)}} + \frac{\mathbf{L}(n_k(x,a))}{n_k(x,a) - 1} \right)
$$
$$
\lesssim 1 \wedge \left( \sqrt{\frac{\text{Var}[R(x,a)]\mathbf{L}(n_k(x,a))}{n_k(x,a)}} + \frac{\mathbf{L}(n_k(x,a))}{n_k(x,a) - 1} \right) \lesssim \sqrt{\frac{\text{Var}[R(x,a)]\mathbf{L}(n_k(x,a))}{n_k(x,a)}} + \frac{\mathbf{L}(n_k(x,a))}{n_k(x,a)},
$$

where in the second-to-last inequality, we used the event $\mathcal{A}^{\text{var,rw}}$ to control $\left| \sqrt{\text{Var}[R(x,a)]} - \sqrt{\widehat{\text{Var}}[R(x,a)]} \right| \lesssim \sqrt{\frac{\mathbf{L}(n_k(x,a))}{n_k(x,a) - 1}}$. $\qquad\square$

*Proof of Lemma F.8.* Let $a = \pi_{k,h}(x)$. Then, by definition of $\overline{\mathbf{V}}_{k,h}(x), \underline{\mathbf{V}}_{k,h}(x)$

$$\overline{\mathbf{V}}_{k,h}(x) - \underline{\mathbf{V}}_{k,h}(x) = (H - h + 1) \wedge (\widehat{p}_{k,h}(x,a)^\top \overline{\mathbf{V}}_{k,h+1} + \mathbf{b}_k^{\mathrm{rw}}(x,a) + \mathbf{b}_{k,h}^{\mathrm{prob}}(x,a) + \mathbf{b}_{k,h}^{\mathrm{str}}(x,a))$$

$$- 0 \vee (\widehat{p}_{k,h}(x,a)^\top \underline{\mathbf{V}}_{k,h+1} - \mathbf{b}_k^{\mathrm{rw}}(x,a) - \mathbf{b}_{k,h}^{\mathrm{prob}}(x,a))$$

$$\leq H \wedge \left\{ \widehat{p}_{k,h}(x,a)^\top (\overline{\mathbf{V}}_{k,h+1} - \underline{\mathbf{V}}_{k,h+1}) + 2\mathbf{b}_k^{\mathrm{rw}}(x,a) + 2\mathbf{b}_{k,h}^{\mathrm{prob}}(x,a) + \mathbf{b}_{k,h}^{\mathrm{str}}(x,a) \right\}$$

$$= H \wedge \left\{ p(x,a)^\top (\overline{\mathbf{V}}_{k,h+1} - \underline{\mathbf{V}}_{k,h+1}) + (\widehat{p}(x,a) - p(x,a))^\top (\overline{\mathbf{V}}_{k,h+1} - \underline{\mathbf{V}}_{k,h+1}) \right.$$

$$\left. + 2\mathbf{b}_k^{\mathrm{rw}}(x,a) + 2\mathbf{b}_{k,h}^{\mathrm{prob}}(x,a) + \mathbf{b}_{k,h}^{\mathrm{str}}(x,a) \right\}$$

$$\leq H \wedge \left\{ p(x,a)^\top (\overline{\mathbf{V}}_{k,h+1} - \underline{\mathbf{V}}_{k,h+1}) + 2\mathbf{b}_k^{\mathrm{rw}}(x,a) + 2\mathbf{b}_{k,h}^{\mathrm{prob}}(x,a) + 2\mathbf{b}_{k,h}^{\mathrm{str}}(x,a) \right\},$$
$$\text{(Lemma F.3)}$$

$$\leq p(x,a)^\top (\overline{\mathbf{V}}_{k,h+1} - \underline{\mathbf{V}}_{k,h+1}) + H \wedge \left\{ 2\mathbf{b}_k^{\mathrm{rw}}(x,a) + 2\mathbf{b}_{k,h}^{\mathrm{prob}}(x,a) + 2\mathbf{b}_{k,h}^{\mathrm{str}}(x,a) \right\},$$

where the last line uses the fact that $p(x,a)^\top (\overline{\mathbf{V}}_{k,h+1} - \underline{\mathbf{V}}_{k,h+1}) \geq 0$ on $\mathcal{A}^{\mathrm{conc}}$ (Proposition F.1, part (c)). Unfolding the above expression inductively, we then find that

$$\overline{\mathbf{V}}_{k,h}(x) - \underline{\mathbf{V}}_{k,h}(x) \leq \mathbb{E}^{\pi_k} \left[ \sum_{t=h}^{H} H \wedge \left\{ 2\mathbf{b}_k^{\mathrm{rw}}(x_t, a_t) + 2\mathbf{b}_{k,h}^{\mathrm{prob}}(x_t, a_t) + 2\mathbf{b}_{k,h}^{\mathrm{str}}(x_t, a_t) \right\} \mid x_t = x \right].$$

To conclude, it suffices to check that $H \wedge \left\{ 2\mathbf{b}_k^{\mathrm{rw}}(x,a) + 2\mathbf{b}_{k,h}^{\mathrm{prob}}(x,a) + 2\mathbf{b}_{k,h}^{\mathrm{str}}(x,a) \right\} \lesssim \sqrt{\mathbf{Z}_k(x,a)}$, for any triple $x, a, h$. To check that this bound holds, we have from (19) that

$$2\mathbf{b}_k^{\mathrm{rw}}(x_t, a_t) + 2\mathbf{b}_{k,t}^{\mathrm{prob}}(x_t, a_t) + 2\mathbf{b}_{k,t}^{\mathrm{str}}(x_t, a_t)$$

$$\lesssim \sqrt{\frac{\mathtt{Var}_{t,x,a}^\star \mathbf{L}(n_k(x_t, a_t))}{n_k(x_t, a_t)}} + \frac{SH\mathbf{L}(n_k(x_t, a_t))}{n_k(x_t, a_t)} + \sqrt{\frac{S \|\overline{\mathbf{V}}_{k,t+1} - \underline{\mathbf{V}}_{k,t+1}\|_{2,(\widehat{p}_k + p)(x_t, a_t)}^2 \mathbf{L}(n_k(x_t, a_t))}{n_k(x_t, a_t)}}$$

$$\lesssim \sqrt{\frac{H\mathbf{L}(n_k(x_t, a_t))}{n_k(x_t, a_t)}} + \frac{SH\mathbf{L}(n_k(x_t, a_t))}{n_k(x_t, a_t)} + H\sqrt{\frac{S\mathbf{L}(n_k(x_t, a_t))}{n_k(x_t, a_t)}},$$

where we recall the notation $\|V\|_{2,\widehat{p}+p} = \sqrt{\|V\|_{2,p}^2 + \|V\|_{2,\widehat{p}}^2}$, and thus the final bound holds since $\mathtt{Var}_{t,x,a}^\star \leq H$ implying that $\|\overline{\mathbf{V}}_{k,t+1} - \underline{\mathbf{V}}_{k,t+1}\|_{2,(\widehat{p}_k + p)(x_t, a_t)}^2 \leq 4H$ for $0 \leq \underline{\mathbf{V}}_{k,t+1} \leq \overline{\mathbf{V}}_{k,t+1} \leq H$. Consolidating the terms, we have $2\mathbf{b}_k^{\mathrm{rw}}(x_t, a_t) + 2\mathbf{b}_{k,t}^{\mathrm{prob}}(x_t, a_t) + 2\mathbf{b}_{k,t}^{\mathrm{str}}(x_t, a_t)$ is at most $\lesssim \left( H\sqrt{\frac{S\mathbf{L}(n_k(x,a))}{n_k(x,a)}} + \frac{SH\mathbf{L}(n_k(x,a))}{n_k(x,a)} \right)$, and thus $H \wedge 2\mathbf{b}_k^{\mathrm{rw}}(x_t, a_t) + 2\mathbf{b}_{k,t}^{\mathrm{prob}}(x_t, a_t) + 2\mathbf{b}_{k,t}^{\mathrm{str}}(x_t, a_t)$ is $\lesssim H \wedge \left( H\sqrt{\frac{S\mathbf{L}(n_k(x,a))}{n_k(x,a)}} + \frac{SH\mathbf{L}(n_k(x,a))}{n_k(x,a)} \right) := \sqrt{\mathbf{Z}_k(x,a)}$. $\qquad\square$

# Part III

# Lower Bounds

## G  Min-Gap Lower Bound for Optimistic Algorithms (Theorem 2.3)

### G.1  Formal Statement

We begin a formal version of the lower bound, Theorem 2.3.

**Theorem G.1.** *Let $c_1, c_2, c_3$ be absolute constants that may depend on the constants defined in Section G.2. Let* Alg *denote an algorithm in the class described in Section G.2 run with confidence parameter $\delta \in (0, 1/8)$. For any $S \geq 1$ and $\epsilon \leq 1/\lceil c_1 S \log(S/\delta) \rceil$, fix any MDP in the class described in Section G.3 so that $|\mathcal{S}| = 2S + 1$, $|\mathcal{A}| = 2$, $H = 2$, and exactly one state has a sub-optimality gap of $\mathrm{gap}_{\min} = \epsilon$ and all other states have a minimum sub-optimality gap of at least $1/2$. Then $\sum_{h,x,a:\mathrm{gap}_h(x,a)>0} \frac{1}{\mathrm{gap}_h(x,a)} \lesssim S + \frac{1}{\mathrm{gap}_{\min}}$ but* Alg *for all sufficiently large $K$ suffers a regret*

$$\mathrm{Regret}_K \geq \frac{c_2 S}{\mathrm{gap}_{\min}} \log(1/\delta) \gtrsim \sum_{h,x,a:\mathrm{gap}_h(x,a)>0} \frac{1}{\mathrm{gap}_h(x,a)} + \frac{S}{\mathrm{gap}_{\min}}$$

*with probability at least $1 - c_2 S \epsilon^{-2} \log(1/\delta) e^{-c_3 S} - 3\delta$.*

In particular, for any $\epsilon \in (0, c)$ for some constant $c$, if $\log(\epsilon^{-1}/\delta) \lesssim S \lesssim \epsilon^{-1}/\log(\epsilon^{-1}/\delta)$ then the above regret lower bound holds with probability $1 - O(\delta)$.

### G.2  Algorithm Class

**Optimistic Q-functions:** We consider algorithms where the optimistic Q-function is constructed as follows: given a reward bonus function $\mathbf{b}_k^{\mathrm{rw}}(x,a) \geq 0$ and an additional nonnegative stage-dependent bonus $\mathbf{b}_{k,h}(x,a)$, and empirical estimates $\widehat{r}_k(x,a)$ of the reward and $\widehat{p}_k(x,a) = (\widehat{p}(x'|x,a))$ of the transition probabilities. We set the Q-function at stage $H$ as $\overline{\mathbf{Q}}_{k,H}(x,a) = \widehat{r}_k(x,a) + \mathbf{b}_k^{\mathrm{rw}}(x,a)$, where $\widehat{r}_k(x,a)$, and for $h \in \{1, \ldots, H-1\}$,

$$\overline{\mathbf{V}}_{k,h+1}(x) := \max_a \overline{\mathbf{Q}}_{k,h+1}(x',a)$$
$$\overline{\mathbf{Q}}_{k,h}(x,a) := \widehat{r}_k(x,a) + \mathbf{b}_k^{\mathrm{rw}}(x,a) + \widehat{p}_k(x,a)^\top \overline{\mathbf{V}}_{k,h+1} + \mathbf{b}_{k,h}(x,a) \tag{23}$$

Lastly, suppose that $\mathbf{b}_k^{\mathrm{rw}}(x,a)$ depends only on rewards collected when the state $(x,a)$ is visited.

Note that this template subsumes the model-based approaches of [1, 16, 6], and if $\mathbf{b}^{\mathrm{rw}}(x,a)$ is made to be time dependent, captures the approach of [5] as well. For the specific lower bound instance we consider, each stage $x \in \mathcal{S}$ can only be visited at a single stages $h \in [2]$, so $\mathbf{b}^{\mathrm{rw}}$ may be chosen to be time dependent without loss of generality. In order to capture the "model-free" methods based on Q-learning due to [9], we can instead mandate that

$$\overline{\mathbf{Q}}_{k,h}(x,a) := \widehat{r}_k(x,a) + \mathbf{b}_k^{\mathrm{rw}}(x,a) + \left( \widehat{p(x,a)^\top \mathbf{V}_{h+1}^\star} \right) + \mathbf{b}_{k,h}(x,a),$$

where $\left( \widehat{p(x,a)^\top \mathbf{V}_{h+1}^\star} \right)$ is a generalized estimate of $p(x,a)^\top \mathbf{V}_{h+1}^\star$, and such that $\left( \widehat{p(x,a)^\top \mathbf{V}_{h+1}^\star} \right)$ is nonnegative. In Lemma 4.2 in [9], one can see that we can take

$$\left( \widehat{p(x,a)^\top \mathbf{V}_{h+1}^\star} \right) = \sum_{s=1}^{n_k(x,a)} \alpha_s \widehat{\mathbb{P}}_{k_s,h}(x,a)^\top \overline{\mathbf{V}}_{k_s,h+1}(x,a),$$

where $k_s$ is the round at which $(x,a)$ was selected for the $s$-th time, $\alpha_s$ is an appropriate weight, $\widehat{\mathbb{P}}_{k_s,h}(x,a)$ is the empirical probability estimate $\widehat{\mathbb{P}}_{k_s}(x,a)[x'] = \mathbb{I}(x' = x_{k_s,h+1})$ equal to indicator at the state $x_{k_s,h+1}$ visited after playing $a$ at $x$ at round $k_s$, and where $\overline{\mathbf{V}}_{k_s,h+1}$ is an optimistic estimate of $\mathbf{V}_{h+1}^\star$ at round $k_s$.

For simplicity, we shall work with the model based formulation (23), though the lower bound can be extended to this more general class.

**Confidence Interval Assumptions:** Our class of algorithms takes in a confidence parameter $\delta \in (0, 1/8)$. We shall also assume that there exists consants $c_{\mathrm{bon}}, \bar{c}_{\mathrm{bon}}$ such that, when the algorithm is run with parameter $\delta$, the bonuses $\mathbf{b}^{\mathrm{rw}}$ and $\mathbf{b}_k^{\mathrm{rw}}$ satisfy[7]

$$\mathbf{b}_{k,h}(x,a) \geq \frac{c_{\mathrm{bon}}}{1 \vee n_k(x,a)}, \quad c_{\mathrm{bon}}\sqrt{\frac{\mathrm{Var}[R(x,a)]\log(1/\delta)}{1 \vee n_k(x,a)}} \leq \mathbf{b}_k^{\mathrm{rw}}(x,a) \leq \bar{c}_{\mathrm{bon}}\sqrt{\frac{\log(M(1 \vee n_k(x,a))/\delta)}{1 \vee n_k(x,a)}}$$

We further assume that $\mathbf{b}^{\mathrm{rw}}(x,a)$ is $\delta$-correct, in the sense that,

$$\mathbb{P}[\forall x, a, k : \mathbf{b}_k^{\mathrm{rw}}(x,a) + \widehat{r}_k(x,a) \geq r(x,a)] \geq 1 - \delta.$$

Lastly, we shall assume that the optimistic overestimate is *consistent* in the sense that for any MDP $\mathcal{M}$ with optimal value $\mathbf{V}_0^{*,\mathcal{M}}$, for any $\epsilon, \delta > 0$ there exists a function $f_{\mathcal{M}}$ such that

$$\mathbb{P}[\forall k \geq f_{\mathcal{M}}(\epsilon, \delta), \ \overline{\mathbf{V}}_{k,0} - \mathbf{V}_0^{*,\mathcal{M}} \leq \epsilon] \geq 1 - \delta.$$

Intuitively, this condition states that with high probability, the optimistic over-estimate of the value estimate approaches the expected reward under the optimal policy. Note that this does *not* assume uniform convergence of the entire value function itself, just the expected reward with respect to the initial state distribution $p_0$ on the optimal policy.

**Remark G.1.** Note that we do not require that our algorithm's confidence intervals are "inflated", in the sense that, with high probability, $\widehat{r}_k(x,a) + \mathbf{b}_k^{\mathrm{rw}}(x,a) - r(x,a) \geq c\mathbf{b}_k^{\mathrm{rw}}(x,a)$, for a universal constant $c$. With this stronger assumption, we note that the proof of the lower bound can be simplified, and some restrictions on $S, \epsilon$ removed. In the interest of generality, we refrain from making this assumption.

### G.3 Formal Lower Bound Instance

Consider the following simple game with $H = 2$, $\mathcal{A} = \{-1, +1\}$ and $\mathcal{S} = \{-S, \ldots, -1, 0, 1, \ldots, S\} = \mathcal{S}_- \cup \{0\} \cup \mathcal{S}_+$, where $\mathcal{S}_- = -[S]$ and $\mathcal{S}_+ = [S]$ (note $|\mathcal{S}| = 2S + 1$). The game always begins at state $x_1 = 0$ with two available actions, $a \in \{-1, +1\}$. Then, $x_2|(x_1 = 0, a_1 = +1) \overset{unif}{\sim} \mathcal{S}_+$, and $x_2|(x_1 = 0, a_1 = -1) \overset{unif}{\sim} \mathcal{S}_-$. Lastly, let $\mathcal{D}$ denote *any* symmetric distribution on $[-1, 1]$ with $\Omega(1)$ variance. For $\epsilon \in (0, 1/8)$, we formally define the reward distributions

$$R(x,a) \sim \begin{cases} 0 & x = 0 \text{ or } a = -1 \\ \frac{1}{2} + \epsilon + \frac{1}{4}\mathcal{D} & (x,a) = (s, 1), s \in [S] \\ \frac{1}{2} + \frac{1}{4}\mathcal{D} & (x,a) = (-s, 1), s \in [S] \end{cases}.$$

It is straightforward to verify the following fact

**Fact G.2.** *The optimal action is always $a = 1$. Moreover,* $\mathtt{gap}_1(0, -1) = \mathtt{gap}_{\min} = \epsilon$, *whereas* $\mathtt{gap}_2(x, -1) \geq \frac{1}{2}$ *for $x \neq 0$.*

In other words, all the gaps for suboptimal arms are $\Omega(1)$, except for the gap at state $x = 0$, which means for this instance with $H = 2$ and $A = 2$ we have $\sum_{x,a,h} \frac{1}{\mathtt{gap}_h(x,a)} \approx S + \frac{1}{\epsilon}$. Nevertheless, we shall show that any algorithm in the class above suffers regret

$$\gtrsim \frac{S}{\epsilon}\log(1/\delta) = \frac{S}{\mathtt{gap}_{\min}}\log(1/\delta).$$

### G.4 The Lower Bound:

**The Lower Bound:** We first show that the optimistic Q-function relative to the optimal value at $(0, 1)$ decays at a rate of at least $\sqrt{S\log(1/\delta)/n_k(0, 1)}$. This will ultimately lead to incurring a regret of $\frac{S\log(1/\delta)}{\epsilon}$, despite the fact that all but one of the Q-function gaps are $\Omega(1)$.

**Proposition G.3.** *Let* Alg *denote an algorithm in the class described in Section G.2 run with confidence parameter* $\delta \in (0, 1/8)$. *Then there exists constants* $c_1, c_2, c_3$, *depending only on the constants described in Section G.2, such that the following holds. For any* $\epsilon \leq 1/\lceil c_1 S \log(S/\delta) \rceil$ *and for* $N = \lfloor c_2 S \log(1/\delta)/\epsilon^2 \rfloor$,

$$\mathbb{P}\left[\forall k : n_k(0, -1) \leq N, \ \overline{\mathbf{Q}}_{k,1}(0, -1) - \mathbf{V}_1^\star(0) \geq \epsilon\right] \geq 1 - Ne^{-c_3 S} - 2\delta$$

We now use Proposition G.3 to prove Theorem G.1. Note that $\mathbf{V}_1^\star(0) = \mathbf{V}_0^\star$. By assumption, with probability $1 - \delta$, $\overline{\mathbf{V}}_0 \leq \mathbf{V}_0^\star + \eta$ after $f(\eta, \delta)$ rounds. Fix an appropriate $\epsilon$ and $N$ in Proposition G.3 and let $K \geq f_\mathcal{M}(\epsilon/2, \delta) + N$. If $n_K(0, -1) > N$ times, then we have

$$\text{Regret}_K > \epsilon N \gtrsim \frac{S \log(1/\delta)}{\epsilon}$$

and the theorem is proved. Thus, suppose not so that $n_K(0, -1) \leq N$. Then by Proposition G.3 we have with high probability that

$$\overline{\mathbf{V}}_0 - \mathbf{V}_1^\star(0) = \max_{a \in \{-1,1\}} \overline{\mathbf{Q}}_{k,1}(0, a) - \mathbf{V}_1^\star(0) \geq \overline{\mathbf{Q}}_{k,1}(0, -1) - \mathbf{V}_1^\star(0) \geq \epsilon$$

However, by assumption $K \geq f_\mathcal{M}(\epsilon/2, \delta)$ which means that on an event that holds with probability at least $1 - \delta$, we have $\overline{\mathbf{V}}_0 - \mathbf{V}_1^\star(0) = \max_{a \in \{-1,1\}} \overline{\mathbf{Q}}_{k,1}(0, a) - \mathbf{V}_1^\star(0) \leq \epsilon/2$, a contradiction.

### G.4.1 Proof of Proposition G.3

Throughout, we will use upper case $C_1, C_2, \dots$ to do denote possibly changing numerical constants that depend on the the constants in the definition of Alg, as set in Section G.2. The lower cast constants $c_1, c_2$ will be coincide with those in Proposition G.3.

Since $\mathbf{Q}_1^\star(0, 1) = \frac{1}{2} + \epsilon$, it suffices to show that

$$\mathbb{P}\left[\forall k : n_k(0, -1) \leq N, \ \overline{\mathbf{Q}}_{k,1}(0, -1) - \frac{1}{2} \geq 2\epsilon\right] \geq 1 - Ne^{-C_3 S} - \delta$$

Fix an $n_0 = \lceil c_1 S / \log(S/\delta) \rceil$ for a constant $c_1$ be specified later, and let

$$\mathcal{E}^{\text{opt}} := \left\{\forall k \geq 1, x \in \mathcal{S}_-, \ \widehat{r}_k(x, 1) + \mathbf{b}_k^{\text{rw}}(x, a) \geq r(x, 1) = \frac{1}{2}\right\}.$$

By the optimism assumption, $\mathcal{E}^{\text{opt}}$ holds with probability at least $1 - \delta$. First we verify that $\overline{\mathbf{Q}}_{k,1}(0, -1) - \frac{1}{2} \geq 2\epsilon$ for $0 \leq n_k(0, -1) \leq n_0$, provided that $\epsilon$ is sufficiently small:

**Claim G.4.** *Suppose that* $\epsilon \leq \frac{c_{\text{bon}}}{2n_0}$. *Then, with probability* $1 - \delta$, $\overline{\mathbf{Q}}_{k,1}(0, -1) - \frac{1}{2} \geq 2\epsilon$ *whenever* $0 \leq n_k(0, -1) \leq n_0$:

*Proof.* We have that

$$\overline{\mathbf{Q}}_{k,1}(0, -1) = \mathbf{b}_{k,1}(x, a) + \sum_{x' \in \mathcal{S}} \widehat{p}_k(x'|0, -1)\overline{\mathbf{V}}_{k,2}(x').$$

Since $p(x|0, -1) = 0$ for $x \notin \mathcal{S}_-$, the empirical probability $\widehat{p}(x|0, -1)$ is also 0, and thus

$$\overline{\mathbf{Q}}_{k,1}(0, -1) - \frac{1}{2} = \mathbf{b}_{k,1}(x, a) + \sum_{x' \in \mathcal{S}_-} \widehat{p}_k(x'|0, -1)\left(\overline{\mathbf{V}}_{k,2}(x') - \frac{1}{2}\right) \tag{24}$$

$$\geq \mathbf{b}_{k,1}(x, a) + \min_{x' \in \mathcal{S}_-}\left(\overline{\mathbf{V}}_{k,2}(x') - \frac{1}{2}\right) \geq \mathbf{b}_{k,1}(x, a),$$

where the first equality and first inequality use $\sum_{x' \in \mathcal{S}_-} \widehat{p}(x|0, -1) = 1$, and the second uses the optimistic event $\mathcal{E}^{\text{opt}}$ to show that $\overline{\mathbf{V}}_{k,2}(x') \geq \widehat{r}_k(x', 1) + \mathbf{b}_k^{\text{rw}}(x', 1) \geq r(x', 1) = \frac{1}{2}$ for $x' \in \mathcal{S}_-$. Using the assumption that $\mathbf{b}_{k,1}(x, a) \geq \frac{c_{\text{bon}}}{1 \vee n_k(x, a)}$, we see that if $n_k(x, a) \leq n_0$ and $\epsilon \leq \frac{c_{\text{bon}}}{2n_0}$, then $\mathbf{b}_{k,1}(x, a) \geq \frac{c_{\text{bon}}}{n_0} \geq 2\epsilon$, as needed. $\square$

Now, we turn to the case where $n_k(x,a) \in \{n_0, \ldots, N\}$ for some $N = \lfloor c_2 S \log(1/\delta)/\epsilon^2 \rfloor$. It light of (24), it suffices to show that for $n_k \leq N$,

$$\sum_{x' \in \mathcal{S}_-} \widehat{p}_k(x'|0,-1)(\overline{\mathbf{V}}_{k,2}(x') - \frac{1}{2}) \geq 2\epsilon. \tag{25}$$

By the definition of our algorithm class, the optimistic Q-function at stage $h = 2$ and pair $(x,a)$ depend only at rewards collected at $(x,a)$, and the construction of our MDP, pairs $(x,a)$ for $x \in \mathcal{S}_-$ are only accessible by playing $(0,-1)$. Hence, to analyze $\overline{\mathbf{Q}}_{k,1}(0,-1)$, for $n_0 \leq n_k(0,-1) \leq N$, it suffices to prove our described lower bound on $\overline{\mathbf{Q}}_{k,1}(0,-1)$ in the simplified game, where at each round $k = 1, 2, \ldots$, the algorithm *always* selects $(0,-1)$, and show that for this algorithm

$$\Delta_0(k) := \sum_{x \in \mathcal{S}_-} \widehat{p}_k(x|0,-1)(\overline{\mathbf{V}}_{k,2}(x) - \frac{1}{2}) \geq 2\epsilon, \forall k \in \{n_0, \ldots, N\}.$$

Turning our attention to this simplified game, for $x \in \mathcal{S}_-$ let $n_k(x)$ denote the number of times $x$ has been visited up to round $k$, and recall $n_k(x,a)$ is the number of times action $a$ is played at stage $s$. Further, set

$$\Delta(x,k) := \overline{\mathbf{V}}_{k,2}(x) - \frac{1}{2}$$

We now make a couple of observations

(a) The vector $(n_k(x))_{x \in \mathcal{S}_-}$ is a uniform multinomial on the states in $\mathcal{S}_-$.

(b) Conditioned on $(n_k(x))_{x \in \mathcal{S}_-}$, we can see that the values of $\overline{\mathbf{V}}_{k,2}(x)$ are independent, because for each $x \in \mathcal{S}_-$, the game decouples into $n_k(x)$ rounds of a two arm bandit game on actions $a \in \{-1, 1\}$.

Using these observations, we prove the following claim:

**Claim G.5.** *There exists constants $C_1, C_2$ such that for any $x \in \mathcal{S}_-$, if $\delta \leq 1/8$ and $n_k(x) \geq C_1 \log(M/\delta)$, then conditioned on the history $(n_j(x'))_{x' \in \mathcal{S}_-, j \geq 1}$, the following event holds with probability at least $1/4$:*

$$\mathcal{E}_k^{\Delta}(x) := \left\{ \Delta(k,x) := \overline{\mathbf{V}}_{k,2}(x) - \frac{1}{2} \geq C_2 \sqrt{\log(1/\delta)/n_k(x)} \right\},$$

*and the events $\{\mathcal{E}_j^{\Delta}(x) : x \in \mathcal{S}_-\}$ are mutually independent (again, given $(n_j(x'))_{x' \in \mathcal{S}_-, j \geq 1}$).*

Therefore, on the optimistic event $\mathcal{E}^{\text{opt}}$, where $\{\Delta(k,x) \geq 0\}$, we can lower bound (again, in the simplified game where we always select action $(0,-1)$),

$$\Delta_0(k) \geq \sum_{x \in \mathcal{S}_-} \widehat{p}(x'|0,-1)\Delta(x,k)$$

$$\geq \sum_{x \in \mathcal{S}_-} \widehat{p}(x'|0,-1)\mathbb{I}(\mathcal{E}_k^{\Delta}(x))C_2\sqrt{\frac{\log(1/\delta)}{n_k(x)}}$$

$$\overset{(i)}{=} \sum_{x \in \mathcal{S}_-} \frac{n_k(x)}{k}\mathbb{I}(\mathcal{E}_k^{\Delta}(x))C_2\sqrt{\frac{\log(1/\delta)}{n_k(x)}}$$

$$= \frac{C_2\sqrt{\log(1/\delta)}}{k} \sum_{x \in \mathcal{S}_-} \mathbb{I}(\mathcal{E}_k^{\Delta}(x))\sqrt{n_k(x)}$$

where $(i)$ uses the fact that for $x \in \mathcal{S}_-$ is only accessible through $(0,-1)$, and that $(0,-1)$ is always selected in the simplified game. Next, observe that in the simplified game, $\overline{n}_k(x) = k/S$, so that if $n_0/S \geq C_3 \log(1/\delta)$ for some constant $C_3$, it holds by an argument similar to Lemma B.7 that with

probability $1 - \delta$, the event $\mathcal{E}_1 := \{\forall x \in \mathcal{S}_-, \forall k \geq n_0, n_k(x) \geq \overline{n}_k(x)/4 = k/4S\}$ holds, yielding

$$\Delta_0(k) \geq \frac{C_2 \sqrt{\log(1/\delta)}}{2k} \sqrt{k/S} \cdot \sum_{x \in \mathcal{S}_-} \mathbb{I}(\mathcal{E}_k^\Delta)$$

$$= \frac{C_2}{2} \sqrt{S \log(1/\delta)/k} \cdot \left( \frac{1}{S} \sum_{x \in \mathcal{S}_-} \mathbb{I}(\mathcal{E}_k^\Delta) \right).$$

Finally, if in addition $n_0/4S \geq C_1 \log(1/\delta)$, where $C_1$ is the constant from claim G.5, then on $\mathcal{E}_1$, it holds that for $k \geq n_0$, $n_k(x) \geq C_1 \log(1/\delta)$. We then set the constant $c_1$ so that $n_0/S \geq C_3 \log(1/\delta)$ and $n_0/4S \geq C_1 \log(1/\delta)$ hold.

Lastly, since (a) $\mathcal{E}_1$ is measurable with respect to the counts $(n_j(x'))_{x' \in \mathcal{S}_-, j \geq 1}$, (b) since $\mathcal{E}_k^\Delta(x)$ are independent given these counts, and (c) $\mathbb{E}[\mathbb{I}(\mathcal{E}_k^\Delta)] \geq 1/4$, a Chernoff bound shows that for $k \geq n_0$, the event $\mathcal{E}_2(k) := \{ \left( \frac{1}{S} \sum_{x \in \mathcal{S}_-} \mathbb{I}(\mathcal{E}_k^\Delta) \right) \geq 1/8 \}$ holds with probability at least $e^{-C_5 S}$ conditioned on $\mathcal{E}_1$. Hence, on $\mathcal{E}^{\mathrm{opt}} \cap \mathcal{E}_1 \cap \bigcup_{k=n_0}^N \mathcal{E}_2(k)$, we have

$$\Delta_0(k) \geq \frac{C_2}{16} \sqrt{S \log(1/\delta)/k} \geq \frac{C_2}{16} \sqrt{S \log(1/\delta)/N}, \quad \forall k \in \{n_0, \dots, N\}.$$

Hence, if $N \leq c_2 \frac{S \log(1/\delta)}{\epsilon^2}$ for some constant $c_2$, we see that $\Delta_0(k) \geq 2\epsilon$ for all $k \in \{n_0, \dots, N\}$. Lastly, we see that

$$\mathbb{P}[(\mathcal{E}^{\mathrm{opt}} \cap \mathcal{E}_1 \cap \bigcup_{k=n_0}^N \mathcal{E}_2(k))^c] \leq \mathbb{P}[(\mathcal{E}^{\mathrm{opt}})^c] + \mathbb{P}[\mathcal{E}_1^c] + \mathbb{P}[(\bigcup_{k=n_0}^N \mathcal{E}_2(k))^c \wedge \mathcal{E}_1]$$

$$\leq \mathbb{P}[(\mathcal{E}^{\mathrm{opt}})^c] + \mathbb{P}[\mathcal{E}_1^c] + N \max_{k \geq n_0} \mathbb{P}[\mathcal{E}_2(k)^c \mid \mathcal{E}_1] \leq 2\delta + N e^{-C_4 S}.$$

Translating to the non-simplified game, we have therefore established that

$$\mathbb{P}[\forall k : n_0 \leq n_k(0, -1) \leq N, \overline{\mathbf{Q}}_{k,1}(-1, 1) - \frac{1}{2} \geq 2\epsilon] \geq 1 - 2\delta + N e^{-C_4 S}.$$

Combining with the additional probability of error $\delta$ for the case $n_k(0, -1) \leq n_0$ concludes the proof.

### G.5 Proof of Claim G.5

We observe that conditioned on the vector $(n_j(x'))_{x' \in \mathcal{S}_-, j \geq 1}$, the games at states $x$ and round $k$ are equivalent to $S$ independent two-arm bandit games with $n_k(x)$ rounds. Note moreover that $\Delta(x, k) = \overline{\mathbf{V}}_{k,2}(x) - \frac{1}{2} \geq \mathbf{b}_k^{\mathrm{rw}}(x, 1) + \widehat{r}_k(x, 1) - \frac{1}{2}$. Hence, restricting to a single state $x$ (and dropping the dependence on $x$ for simplicity), it suffices to show that for $k$ rounds of an appropriate two-arm bandit game with $a \in \{-1, 1\}$ with empirical rewards $\widehat{r}_k(a)$ and bonuses $\mathbf{b}_k^{\mathrm{rw}}(a)$, $R(-1) = 0$ and $R(1) \sim \frac{1}{2} + \frac{1}{4}\mathcal{D}$, that

$$\forall k \geq \mathsf{C}_1 \log(S/\delta), \quad \mathbb{P}[\mathbf{b}_k^{\mathrm{rw}}(1) + \widehat{r}_k(1) - \frac{1}{2} \geq \sqrt{\log(1/\delta)/k}] \geq \frac{1}{4}$$

where we have dropped the dependence on $x$ for simplicity. Throughout, we will also use the notation $\mathsf{C}_1, \mathsf{C}_2, \mathsf{C}_3$ to denote constants specific to the proof of Claim G.5, and reserve $C_1, C_2$ for the constants in the claim statement.

If $\delta \leq 1/8$, then a standard argument shows that for some constant $\mathsf{C}_1$ (depending on $\overline{c}_{\mathrm{bon}}$), $n_k(-1) \leq \mathsf{C}_1 \log(S/\delta)$. Indeed, define the event $\mathcal{E}_0 := \{\forall k \geq 1 : \mathbf{b}_k^{\mathrm{rw}}(1) + \widehat{r}_k(1) \geq r(1) = \frac{1}{2}\}$; by assumption on our confidence intervals, complement of this event occurs with probability at most $\delta \leq 1/8$. Note also that on $\mathcal{E}_0$, since $R(-1) = 0$ with probability 1, it holds that for any $j \leq k$ with $n_j(-1) \geq \mathsf{C}_1 \log(S/\delta)$

$$\widehat{r}_j(-1) + \mathbf{b}_j^{\mathrm{rw}}(x, a) = \mathbf{b}_j^{\mathrm{rw}}(x, a) \overset{(i)}{\leq} \overline{c}_{\mathrm{bon}} \sqrt{\frac{\log(S n_j(-1)/\delta)}{n_j(-1)}}$$

$$\overset{(ii)}{\leq} \frac{1}{2} = r(1) \leq \widehat{r}_j(1) + \mathbf{b}_j^{\mathrm{rw}}(1),$$

where in $(i)$ we used the definition of the confidence interval with $M \lesssim S$, and in $(ii)$ we used $n_j(-1) \geq \mathsf{C}_1 \log(S/\delta)$ for an appropriately tuned constant $\mathsf{C}_1$. Since $a_j := \arg\max_a \widehat{r}_j(a) + \mathbf{b}_j^{\text{rw}}(a)$, we have $a_j = 1$. This implies that $n_k(-1) \leq \max_{j \geq 1} n_j(-1) \leq \mathsf{C}_1 \log(M/\delta)$.

Next, set $k_0 = \mathsf{C}_1 \log(M/\delta)$. We wish to show that for $k \geq k_0$,

$$\widehat{r}_k(1) + \mathbf{b}_k^{\text{rw}}(1) \gtrsim \frac{\log(1/\delta)}{k}$$

There are two technical challenges: first, the confidence interval $\mathbf{b}_k^{\text{rw}}(1)$ might be nearly tight, so that we cannot show that with high probability, $\widehat{r}_k(1) + \mathbf{b}_k^{\text{rw}}(1) \gtrsim \mathbf{b}_k^{\text{rw}}(1)$. Second, because the algorithm adaptively chooses to sample actions $a \in \{-1, 1\}$, $\widehat{r}_k(1)$ *does not* have the distribution of $n_k(1)$ i.i.d. samples from $R(1)$.

We can get around this as follows. We can imagine all rewards sampled from action 1 as being drawn at the start of the game, and constituting a sequence $R^{(1)}(1), R^{(2)}(1), \ldots$ and so on. Then, $\widehat{r}_k(1)$ is the average of the samples $1, \ldots, n_k(1)$, where $n_k(1) \leq k$. Therefore

$$n_k(1)(\widehat{r}_k(1) - \frac{1}{2}) = \sum_{i=1}^{n_k(1)} (R^{(i)}(1) - \frac{1}{2}) = \sum_{i=1}^{k} (R^{(i)}(1) - \frac{1}{2}) - \sum_{i=n_k(1)+1}^{k} (R^{(i)}(1) - \frac{1}{2}).$$

$$= \sum_{i=1}^{k} (R^{(i)}(1) - \frac{1}{2}) - \sum_{i=k-n_k(-1)+1}^{k} (R^{(i)}(1) - \frac{1}{2}),$$

where the last line uses $n_k(1) + n_k(-1) = k$.

Now consider the event $\mathcal{E}_1(\delta) := \{n_k(-1) \leq k_0\}$, where we recall $k_0 = \mathsf{C}_1 \log(M/\delta)$ was our $1 - \delta$-probability upper bound on $n_k(-1)$. On $\mathcal{E}_1(\delta)$, $n_k(-1) = j$ for some $j \in \{0, 1, \ldots, k_0\}$, and we can lower bound the above expression by

$$\geq \sum_{i=1}^{k} (R^{(i)}(1) - \frac{1}{2}) - \max_{j=0,\ldots,k_0} \sum_{i=k-j+1}^{k} (R^{(i)}(1) - \frac{1}{2}).$$

Observe now that we have lower bounded $n_k(1)(\widehat{r}_k(1) - \frac{1}{2})$ in terms of quantities depending only on the i.i.d. reward sequence $(R^{(i)}(1))$, and *not* on the quantities $n_k(-1), n_k(1)$.

Moreover, a standard maximal inequality implies that the following event $\mathcal{E}_2(\delta)$ holds for an appropriate constant $\mathsf{C}_2$ with probability $1 - \delta$:

$$\mathcal{E}_2(\delta) := \left\{ \max_{j=0,\ldots,k_0} \sum_{i=k-j+1}^{k} (R^{(i)}(1) - \frac{1}{2}) \leq \mathsf{C}_2 \sqrt{k_0 \log(1/\delta)} \right\} \tag{26}$$

Lastly, since $R^{(i)}$ is symmetric, we have that the following event $\mathcal{E}_3$ holds with probability $1/2$:

$$\mathcal{E}_3 := \left\{ \sum_{i=1}^{k} (R^{(i)}(1) - \frac{1}{2}) \geq 0 \right\}.$$

Hence, on $\mathcal{E}_1(\delta) \cap \mathcal{E}_2(\delta) \cap \mathcal{E}_3$,

$$n_k(1)(\widehat{r}_k(1) - \frac{1}{2}) \geq \underbrace{\sum_{i=1}^{k} (R^{(i)}(1) - \frac{1}{2})}_{\geq 0} - \max_{j=0,\ldots,k_0} \sum_{i=k-j+1}^{k} (R^{(i)}(1) - \frac{1}{2})$$

$$\geq -\mathsf{C}_2 \sqrt{\log(1/\delta) k_0}.$$

If we further assume that $k \geq 2\mathsf{C}_1 \log(M/\delta)$, then $n_k(-1) \leq k_0 \leq k/2$, so that $\mathcal{E}_1(\delta)$ implies $n_k(1) \geq k/2$. Dividing both sides of the above by $k$ and bringing $1/k$ into the square root yields (again on $\mathcal{E}_1(\delta) \cap \mathcal{E}_2(\delta) \cap \mathcal{E}_3$)

$$(\widehat{r}_k(1) - \frac{1}{2}) \geq -\frac{\mathsf{C}_2}{2} \sqrt{\frac{\log(1/\eta)}{k} \cdot \frac{k_0}{k}}. \tag{27}$$

Moreover, by the lower bound assumption on $\mathbf{b}^{\mathrm{rw}}$ and the fact that $R(1)$ has $\Omega(1)$ variance, there exists some constant $\mathsf{C}_3$ such that

$$\mathbf{b}^{\mathrm{rw}}(n_k(1)) \geq c_{\mathrm{bon}}\sqrt{\frac{\mathrm{Var}[R(x,a)]\log(1/\delta)}{n_k(1)}} \geq \mathsf{C}_3\sqrt{\frac{\log(1/\delta)}{k}},$$

where again we use $n_k(1) \leq k$. Combining with (27), we have on $\mathcal{E}_1(\delta) \cap \mathcal{E}_2(\delta) \cap \mathcal{E}_3$ that

$$(\widehat{r}_k(1) - \frac{1}{2}) + \mathbf{b}^{\mathrm{rw}}(n_k(1)) \geq \mathsf{C}_3\sqrt{\frac{\log(1/\delta)}{k}} - \frac{\mathsf{C}_2}{2}\sqrt{\frac{\log(1/\delta)}{k} \cdot \frac{k_0}{k}}.$$

Hence, if $k_0/k \leq (\mathsf{C}_3/\mathsf{C}_2)^2$, or equivalently if $k \geq \mathsf{C}_1(\mathsf{C}_3/\mathsf{C}_2)^{-2}\log(M/\delta)$, then

$$(\widehat{r}_k(1) - \frac{1}{2}) + \mathbf{b}^{\mathrm{rw}}(n_k(1)) \geq \frac{\mathsf{C}_3}{2}\sqrt{\frac{\log(1/\delta)}{k}} \geq \frac{\mathsf{C}_3}{2}\sqrt{\frac{\log(1/\delta)}{2n_k(1)}}$$

on the event $\mathcal{E}_1(\delta) \cap \mathcal{E}_2(\delta) \cap \mathcal{E}_3$. Lastly, for $\delta \leq 1/8$, we note $\mathbb{P}[\mathcal{E}_1(\delta) \cap \mathcal{E}_2(\delta) \cap \mathcal{E}_3] \geq \frac{1}{2} - 2\delta \geq 1/4$. Recalling our earlier condition $k \geq 2\mathsf{C}_1\log(M/\delta)$, the claim now holds with by setting the constant $C_1$ in the claim statement to be $\mathsf{C}_1 \max\{2, (\mathsf{C}_3/\mathsf{C}_2)^{-2}\}$, and $C_2$ to be $\frac{\mathsf{C}_3}{2\sqrt{2}}$.

# H  Information Theoretic Lower Bound (Proposition 2.2)

In this section we construct give a proof of the information theoretic lower bound Proposition 2.2, as well as a non-asymptotic bound that holds even for non-uniformly good algorithms.

## H.1  Construction of the hard instance

Our construction mirrors the lower bounds due to [4], but with specific and non-uniform gaps. We define $\mathcal{M}$ as an MDP on state space $\mathcal{S} = [S + 2]$, with actions $\mathcal{A} = [A]$, and horizon $[H]$. We will first state the construction for $H \geq 2$, and then remark on the modification for $H = 1$ at the end of the section. For $a \in [A], x \in [S]$, we set

$$p(x' = S + 1 | x, a) = \frac{3}{4} - \frac{2}{H - 1}\Delta_{x,a}, \quad p(x' = S + 2 | x, a) = 1 - p(x' = S + 1 | x, a).$$

Furthermore, we set the initial state to have the distribution $x_1 \overset{unif}{\sim} [S]$, and set

$$p(x' = S + 1 | x = S + 1, a) = 1, \quad p(x' = S + 2 | x = S + 2, a) = 1 \forall a \in [A].$$

Finally, the rewards are set deterministically as

$$R(x, a) := \begin{cases} 0 & x \in [S] \\ 0 & x \in \{S + 1, S + 2\}, a > 1 \\ 1 & (x, a) = (S + 1, 1) \\ \frac{1}{2} & (x, a) = (S + 2, 1) \end{cases}$$

We may then verify that $\mathbf{V}_h^{\star}(S + 1) = (H - h + 1)$ and $\mathbf{V}_h^{\star}(S + 1) = (h - H + 1)/2$, which implies that that for $x \in [S]$,

$$
\begin{aligned}
\mathtt{gap}_h(x, a) &= \left( \max_{a'} \sum_{x'} p(x' | x, a') \mathbf{V}_{h+1}^{\star}(x') \right) \\
&= \max_{a'}(p(S + 1 | x, a') - p(S + 1 | x, a))(H - h) + (p(S + 2 | x, a') - p(S + 2 | x, a))\frac{(H - h)}{2} \\
&= \max_{a'}(p(S + 1 | x, a') - p(S + 1 | x, a))(H - h) - (p(S + 1 | x, a') - p(S + 1 | x, a))\frac{(H - h)}{2} \\
&= \max_{a'}(p(S + 1 | x, a') - p(S + 1 | x, a))\frac{H - h}{2} \\
&= \frac{2\Delta_{x,a}}{(H - 1)} \cdot \frac{H - h}{2},
\end{aligned}
$$

and in particular that $\mathtt{gap}_1(x, a) = \Delta_{x,a}$. For $H = 1$, the construction is modified so that $\mathcal{S} = [S]$, and

$$R(x, a) \sim \text{Bernoulli}(\frac{3}{4} - \Delta_{x,a}), \text{ and } x_1 \overset{unif}{\sim} [S].$$

Then, we see that $\mathtt{gap}_1(x, a) = \Delta_{x,a}$. In what follows, we will adress the $H \geq 2$ case; the case $H = 1$ will follow from similar, but simpler arguments.

## H.2 Regret Lower Bound Decomposition

We can now lower bound the expected regret as

$$\mathbb{E}^{\mathcal{M}}[\text{Regret}_K] := \mathbb{E}^{\mathcal{M}}[\sum_{k=1}^{K} \mathbf{V}_0^{\star} - \mathbf{V}_0^{\pi_k}]$$

$$:= \mathbb{E}[\sum_{k=1}^{K} \sum_{x} p(x_1 = x)\{\mathbf{V}_1^{\star}(x) - \mathbf{V}_1^{\pi_k}(x)\}]$$

$$\overset{(i)}{\geq} \mathbb{E}^{\mathcal{M}} \left[\sum_{k=1}^{K} \sum_{x} p(x_1 = x)\{\mathbf{V}_1^{\star}(x) - \mathbf{Q}_1^{\star}(x, \pi_{k,1}(x))\}\right]$$

$$= \mathbb{E}^{\mathcal{M}} \left[\sum_{k=1}^{K} \sum_{x} p(x_1 = x)\mathtt{gap}_1(x, \pi_{k,1}(x))\right]$$

$$= \mathbb{E}^{\mathcal{M}} \left[\sum_{k=1}^{K} \sum_{x,a} p(x_1 = x)\mathbb{I}(\pi_{k,1}(x) = a)\mathtt{gap}_1(x, a)\right]$$

$$= \sum_{x,a} \mathbb{E}^{\mathcal{M}}[\overline{n}_K(x,a)]\mathtt{gap}_1(x,a), \tag{28}$$

where inequality $(i)$ follows since $\mathbf{V}_1^{\pi_k}(x) = \mathbf{Q}_1^{\pi_k}(x, \pi_{k,1}(x)) \leq \mathbf{Q}_1^{\star}(x, \pi_{k,1}(x))$. We now show that for all sufficiently large $K \geq K_0(\mathcal{M})$, any uniformly correct algorithm must have

$$\forall (x,a) : x \in [S], \ \mathtt{gap}_1(x,a) > 0, \forall K \geq K_0(\mathcal{M})$$

$$\mathbb{E}^{\mathcal{M}}[\overline{n}_K((x,a)] \gtrsim (\frac{2}{H-1}\Delta_{x,a})^{-2} \log K \gtrsim \frac{H^2}{\Delta_{x,a}^{-2}} \log K = \frac{H^2}{\mathtt{gap}_1(x,a)^2} \log K, \tag{29}$$

which concludes the proof since

$$\mathbb{E}^{\mathcal{M}}[\text{Regret}_K] \gtrsim \sum_{x,a:\mathtt{gap}_1(x,a)>0} \frac{H^2}{\mathtt{gap}_1(x,a)^2} \log K, \cdot \mathtt{gap}_1(x,a) = \sum_{x,a:\mathtt{gap}_1(x,a)>0} \frac{H^2}{\mathtt{gap}_1(x,a)} \log K.$$

We further note that this argument can also show that, for all $K$ sufficiently large and all $h \in [H-1]$

$$\mathbb{E}^{\mathcal{M}}[\text{Regret}_K] \gtrsim \sum_{x,a:\mathtt{gap}_h(x,a)>0} \frac{(H-h)^2}{\mathtt{gap}_h(x,a)} \log K. \tag{30}$$

as well.

## H.3 Proof of Equation (29)

Throughout, we fix a state $x \in [S]$, and an action $a : \mathtt{gap}_1(x,a) > 0$. We shall further introduce the shorhand

$$\mathbf{\Delta}_{x,a} := \frac{2\Delta_{x,a}}{H-1} \in (0, 1/2), \tag{31}$$

where the bound on $\mathbf{\Delta}_{x,a}$ follows from $\Delta_{x,a} \in (0, H/8)$.

To lower bound Equation (29), we follow steps analogues to standard information theoretic lower bounds. Our exposition will follow [7]. First, we state a lemma which is the MDP analogue of Equation (6) in [7]. Its proof is analogous, and omitted for the sake of brevity:

**Lemma H.1.** *Let $\mathcal{M} = (\mathcal{S}, \mathcal{A}, H, r, p^{\mathcal{M}}, p_0, R^{\mathcal{M}})$ and $\mathcal{M}' = (\mathcal{S}, \mathcal{A}, H, r, p^{\mathcal{M}'}, p_0, R^{\mathcal{M}'})$ denote two episodic MDPs with the same state space $\mathcal{S}$, action space $\mathcal{A}$ and horizon $h$, and initial state distribution $p_0$. For any $(x,a) \in \mathcal{S} \times \mathcal{A}$, let $\nu^{\mathcal{M}}(x,a)$ denote the law of the joint distribution of $(X', R)$ where $X' \sim p^{\mathcal{M}}(\cdot|x,a)$ and $R \sim R^{\mathcal{M}}(x,a)$; define the law $\nu^{\mathcal{M}}(x,a)$ analogously.*

*Finally, fix a horizon $K \geq 1$, and let $\mathcal{F}_K$ denote the filtration generated by all rollouts up to episode $K$. Then, for any $\mathcal{F}_K$-measurable random variable $Z \in [0, 1]$,*

$$\mathrm{kl}(\mathbb{E}^{\mathcal{M}}[Z], \mathbb{E}^{\mathcal{M}'}[Z]) \leq \sum_{x,a} \mathbb{E}^{\mathcal{M}}[\overline{n}_K(x, a)] \mathrm{KL}(\nu^{\mathcal{M}}(x, a), \nu^{\mathcal{M}'}(x, a)),$$

*where $\mathrm{kl}(x, y) = x \log \frac{x}{y} + (1 - x) \log \frac{1-x}{1-y}$ denotes the binary KL-divergence, and $\mathrm{KL}(\cdot, \cdot)$ denotes the KL-divergence between two probability laws.*

We apply the above lemma as follows. For our fixed pair $(x, a)$, define an alternate $\mathcal{M}'$ to be the MDP which coincides with $\mathcal{M}$ except that

$$p(x'|x, a) = \frac{3}{4} + \eta, \ \eta = \min\{7/8, \frac{3}{4} + \mathbf{\Delta}_{x,a}\}$$

By construction, $\mathcal{M}$ and $\mathcal{M}'$ differ only at their law at $(x, a)$. Thus,

$$\mathrm{kl}(\mathbb{E}^{\mathcal{M}}[Z], \mathbb{E}^{\mathcal{M}'}[Z]) \leq \mathbb{E}^{\mathcal{M}}[\overline{n}_K(x, a)] \mathrm{KL}(\nu^{\mathcal{M}}(x, a), \nu^{\mathcal{M}'}(x, a)).$$

We the following lower bound controls the KL divergence between the laws $\nu^{\mathcal{M}}(x, a), \nu^{\mathcal{M}'}(x, a)$:

**Claim H.2.** *There exists a universal constant $c$ such that*

$$\mathrm{KL}(\nu^{\mathcal{M}}(x, a), \nu^{\mathcal{M}'}(x, a)) \lesssim c\mathbf{\Delta}_{x,a}^2.$$

*Proof.* At $(x, a)$, $R(x, a) = 0$ with probability under both $\mathcal{M}, \mathcal{M}'$. Moreover, recall that under $\mathcal{M}$, $(x, a)$ transition to state $S+1$ with probability $\frac{3}{4} - \mathbf{\Delta}_{x,a}$, and to $S+2$ with probability $1-(\frac{3}{4} - \mathbf{\Delta}_{x,a},)$. On the other hand, $\mathcal{M}'$ transtion to $S+1$ with probability $\frac{3}{4} + \eta$, and $S+2$ with probability $1-(\frac{3}{4} + \eta)$. Consequently both laws are equivalent to Bernoulli distributions with parameters $\frac{3}{4} - \mathbf{\Delta}_{x,a}$ and $\frac{3}{4} + \eta$, respectively. Since $\mathrm{kl}(x, y)$ is precisely $\mathrm{KL}(\mathrm{Bernoulli}(x), \mathrm{Bernoulli}(y))$ for $x, y \in (0, 1)$,

$$\mathrm{KL}(\nu^{\mathcal{M}}(x, a), \nu^{\mathcal{M}'}(x, a)) = \mathrm{kl}\left(\frac{3}{4} - \mathbf{\Delta}_{x,a}, \frac{3}{4} + \eta\right).$$

Lastly, set $x = \frac{3}{4} - \mathbf{\Delta}_{x,a}$ and $y = \frac{3}{4} + \min\{\frac{7}{8}, \mathbf{\Delta}_{x,a}\}$. We $y - x \leq 2\mathbf{\Delta}_{x,a}$, and by assumption on $\mathbf{\Delta}_{x,a} \leq 1/2$, Thus, $1/4 \leq x \leq y \leq 7/8$. Hence, a standard Taylor expansion (e.g. **(author?)** [13, Lemma E.1]) shows that there exists a universal constant $c$ such that $\mathrm{kl}(x, y) \leq \frac{c}{(x-y)^2} \leq \frac{4c}{\mathbf{\Delta}_{x,a}^2}$, as needed. $\square$

As a consequence, we see that for any $\mathcal{F}_K$-measurable $Z \in [0, 1]$, we find

$$\mathbb{E}^{\mathcal{M}}[\overline{n}_K(x, a)] \gtrsim \mathbf{\Delta}_{x,a}^{-2}\mathrm{kl}(\mathbb{E}^{\mathcal{M}}[Z], \mathbb{E}^{\mathcal{M}'}[Z]) \gtrsim \frac{H^2}{\Delta_{x,a}^2}\mathrm{kl}(\mathbb{E}^{\mathcal{M}}[Z], \mathbb{E}^{\mathcal{M}'}[Z]),$$

where the last inequality uses that $\mathbf{\Delta}_{x,a} \lesssim \Delta_{x,a}/H$.

To conclude, it suffices to exhibit a random variable $Z_K$ such that, for $K$ sufficiently large,

$$\mathrm{kl}(\mathbb{E}^{\mathcal{M}}[Z], \mathbb{E}^{\mathcal{M}'}[Z]) \gtrsim (1 - \alpha) \log K.$$

To this end, consider $Z_K = \frac{S\overline{n}_K(x,a)}{K}$. Note that since $x$ is only visited with probability at most $1/S$ at stage $h = 1$, and with probability $0$ for stages $h \geq 2$, we have

$$\overline{n}_K(x, a) = \sum_{k=1}^{K} \mathbb{E}^{\mathcal{M}}[\mathbb{P}(x_1 = x)\mathbb{I}(\pi_{k,1}(x_1) = 1)] = \frac{1}{S}\sum_{k=1}^{K} \mathbb{E}^{\mathcal{M}}[\mathbb{I}(\pi_{k,1}(x_1) = 1)] \leq K/S,$$

which implies that, $Z_K \in [0, 1]$ with probability one. Moreover, note that by an argument similar to that of (28), that under the MDP $\mathcal{M}'$,

$$\mathbb{E}^{\mathcal{M}'}[\mathrm{Regret}_K] \geq \eta\mathbb{E}^{\mathcal{M}'}[\sum_{a' \neq a} \overline{n}_K(x, a')] = \eta(\frac{K}{S} - \mathbb{E}^{\mathcal{M}'}\overline{n}_K(x, a')]) = \frac{\eta K}{S}(1 - \mathbb{E}^{\mathcal{M}'}[Z_K]).$$

Hence, if Alg is $\alpha$-uniformly good, then there existsa constant $C_{\mathcal{M}'}$ such that

$$1 - \mathbb{E}^{\mathcal{M}'}[Z_K] \leq \frac{C_{\mathcal{M}'}\eta}{S} K^{\alpha-1}.$$

By the same token, there exists a constant $C_{\mathcal{M}}$ such that

$$C_{\mathcal{M}}K^{\alpha} \geq \mathbb{E}^{\mathcal{M}}[\mathrm{Regret}_K] \geq \mathrm{gap}_1(x,a)\mathbb{E}^{\mathcal{M}}[\bar{n}_K(x,a)] = \frac{K\mathrm{gap}_1(x,a)}{S}\mathbb{E}^{\mathcal{M}}[Z_K]\mathrm{gap}_1(x,a)/S.$$

which implies that $\mathbb{E}^{\mathcal{M}}[Z_K] \leq \frac{SC_{\mathcal{M}}K^{\alpha-1}}{\mathrm{gap}_1(x,a)}$. Furthermore, by Inequality (11) in [7], it holds that

$$\mathrm{kl}(x,y) \geq (1-x)\log\frac{1}{1-y} - \log 2$$

which implies that for $K$ sufficiently large,

$$\mathrm{kl}(\mathbb{E}^{\mathcal{M}}[Z_K], \mathbb{E}^{\mathcal{M}'}[Z_K]) \geq (1 - \frac{SC_{\mathcal{M}}K^{\alpha-1}}{\mathrm{gap}_1(x,a)})\left\{(1-\alpha)\log K - \log\frac{C_{\mathcal{M}'}\eta}{2S}\right\} \gtrsim (1-\alpha)\log K.$$