[Reviews · NeurIPS 2019]

Reviewer 1



This work gives a gap-dependent regret analysis for tabular MDPs. Since most recent works concentrate on the minimax regret bounds of MDPs, it is important but hard to analysis the problem-dependent performance of algorithms. I think this work gives one step for this analysis. The final result involving the sub-optimal gaps is also natural. The paper is clearly written and easy to read. Further, the definition of $gap(x, a)$ seems to be strange for me. For finite-horizon MDPs, it is possible that $x$ has different optimal action at different $h$. Is it possible that there exists a worst case where the $gap(x,a)=0$ for all $x,a$? If so, what will the regret like? The work is organized in a clear way, although there are many notations. ----- The feedback responds to my questions.

Reviewer 2



This paper provides with a theoretical analysis of optimistic algorithms in the setting of episodic MDPs. The authors provide with a problem-dependent and gap-dependent log(T) regret bound which is a finite-time bound, that unlike previous bounds does not take into account other problem properties such as diameter or mixing times. The paper includes several theoretical results, and introduces a novel 'clipping' trick used to prove the bounds. I found the paper interesting, however, it is constructed in a way which is hard to follow and not very readable. First, many concepts are referred to before they are defined (for instance, the notion of optimistic algorithms). Furthermore, the paper is packed with notations and it is very difficult to keep track of all of them, it might be useful to remind the notations when they are used in some critical places in the paper. Lastly, I found that intuition is very lacking in the paper. Many technical details are provided but not much intuition regarding the analysis, why is it possible to use the clipping trick and why with the specific clipping threshold that is given, etc. I believe the contributions of this work are important and relevant for the community, and the novelty in analysis can be used in future research in the field, however, I feel that an improvement of the presentation, by adding some intuition, defining the concepts and notations before they are referred to, and clear explanations of the steps to get the bound, will significantly improve the readability of this paper. General comments: line 97 - what does 'average reward of \pi_*' mean? Average over what? Is it an average over states? Do you mean the average return or average over immediate reward? line 168 - You are noting that the regret is bounded by KH = T. It is not clear why is this true without assuming any assumptions on the reward distributions.. if rewards are not bounded, this is not necessarily true. line 200 - you define gap_h - for what h is this defined? all h? Technical comments: line 39 - 'worse-cast' should be 'worst-case' line 100 = 'stage' should be 'state' line 110 - 'upper bounds' should be 'lower bounds' line 228 - 'futher' should be 'further' line 311 - 'that that' - remove one 'that' --Added after author feedback-- I have read the authors' feedback. Given that they will reorganize the paper to improve readability and clarity, and lighten the notation overhead, I vote for an accept.

Reviewer 3



The paper investigates an interesting problem in RL in the episodic setup, which is of significance in practical domains. The proposed algorithm, StrongEuler, simultaneously achieves the worst-case optimal regret bound of \sqrt{HSAT} and the problem-dependent regret bound established in the paper, which depends on the gap quantities gap(x,a) for various (x,a) pairs. This is a nice contribution to the filed, and is of practical significance for RL applications. From the technical perspective, this paper provides a solid contribution: In order to derive the logarithmic regret bound with the right dependence on the gap parameters, the authors leveraged novel proof techniques, which could be independent interest for the analysis of other algorithms as well. I have read most of the proofs, and think they are correct (though due to time constraints, I couldn’t check all the details). About weaknesses. Despite the great technical material covered in the paper, its choice of organization makes the paper hard to follow; see detailed comments below. In addition, some related and recent literature on regret minimization in RL seem to be missing. Besides, I have some technical comments, which I detail below. 1. Organization: As said earlier, despite solid theoretical content, one may find the paper hard to follow due to the choice of organization. In particular: (i) one may expect that the algorithm StrongEuler to be fully specified in the main text rather than the supplementary; (ii) The proofs in the supplementary are not well-organized, and are hard-to-follow for most readers; and (ii) there is no conclusion section. 2. About Definition 3.1. The optimism should hold “with high probability”, right? Without this, I am not sure if one can guarantee that \bar Q_{k,h} \ge Q*_{k,h} for all x, a, k, and h. Could you explain? 3. Line 218, the statement “all existing optimistic algorithm”: the aforementioned collection of algorithms is not defined precisely. Since you are making a claim about the regret of such algorithm (which seems to be a strong one), you must specify this set clearly. Otherwise, the statement *should* be relaxed. Another vague statement of this type appears in line 212: “… is unavoidable for the sorts of the optimistic algorithms that we typically see in the literature”: again, I may ask to make the statement more precise and specific; otherwise, the claim needs to be accordingly relaxed. Minor comments: a. Theorem 2.4: \epsilon is not introduce. Does it have the same range as in Theorem 2.3? b. Line 31: The logarithmic regret bound in Jaksch et al. is an exception, as it is non-asymptotic (though it depends on the “wrong gap” gap_*). c. Line 48: the term “almost-gap-dependent” does not appear to be precise enough. d. Line 96: what is \tilde S? Shouldn’t it be S? e. Line 107: P(s’|s,a) is not introduced here yet, and later it seems you use p(s’|s,a). f. Line 107: that can be made optimal --> … **uniquely** optimal (Note the uniqueness here is crucial for the proof in [15] based on change-of-measure argument to work). e. Line 897: I couldn’t verify the second inequality \sqrt{a} + \sqrt{b} \precsim \sqrt{a+b}. Could you explain? Typos: Line 20: number states --> number of states Line 39: worse-case --> worst-case Line 93: discounted regret --> undiscounted regret (Isn’t it correct?) Line 95: [8] give … --> [8] gives Line 120: et seq. --> etc. (??) Line 129: a_{a’} --> a_{h’} Line 166-167: sharped --> sharpened Line 169: in right hand side --> in the right-hand side Line 196: we say than … --> we say that … Line 230: In the regret bound, remove extra “}”. Line 269: \sum_{t=1}^K … \omega_{k,h} --> …\omega_{t,h} Line 268: at the end of the sentence, “)” is missing. Line 275 (and elsewhere): Cauchy-Schwartz --> Cauchy-Schwarz Line 438: the second term does not dependent --> the rest of the sentence is missing. Line 502, 590, and elsewhere: author? --> use correct referencing! Line 525: similar to … --> similarly to … Line 634: Then suppose that … --> is it correct to begin the lemma with “then”? Line 656: proven Section … --> proven in Section … Line 668: We can with a crude comparison … --> The sentence does not make sense. Line 705 – Eq. 19: gap_h --> gap_h(x,a) AND “)” is missing Line 703 – iii : “)” is missing. Line 814: similar to --> similarly to Algorithm 1: Input is not provided. Algorithm 2, Line 4: rsum --> rsum_k AND the second rsum should perhaps be rsumsq Line 1085: Lemma 6 in [6] in shows --> remove the second “in” -- Updates (after author feedback) -- I have read the other reviews and have gone through authors' response. The response satisfactorily clarified my raised comments. In particular, the authors provided a precise plan to revise the organization of the paper. I therefore increase my score to 8.

[Author Response · NeurIPS 2019]

We thank the reviewers for their extremely helpful feedback. We also thank all reviewers for pointing out the typos, which we shall all correct in the final revision.

**Organization, Clarity, Notation**: We appreciate your feedback regarding the organization. Due to space constraints, we provided a brief proof sketch, and chose not to include further details of the StrongEuler algorithm. However, we recognize that this can obfuscate the intuition about both the algorithm and its analysis. We therefore propose the following reorganization: **Optimistic Algorithms:** After describing the general MDP setting but before the statement of our main results, we will introduce the notion of optimistic algorithms. We will explain how StrongEuler and related algorithms in the literature achieve optimism by combining empirical estimates of transition probabilities/rewards with confidence bound bonuses. The specific form of the bonuses for StrongEuler will remain in the appendix to save space, but this should give more intuition for how StrongEuler operates. This will also address help to clarify for which class of algorithms Theorem 2.3 applies (re: Reviewer 3's comments). **Proof Sektch:** In order to improve intuition and cut down notation overhead, we will instead state a simplified but informal version of Prop 3.2 that specifies the dominant term, albeit with a coarser dependence on the variances. This will eliminate clutter and allow us to simplify the rest of proof. In addition, this will free up additional room to further explain the intuition behind the clipping in Prop 3.1. In particular, we will provide intuition for the proof, and clarify why $\mathrm{gap}_{\min}/H$ appears in the analysis. This will in turn clarify the dependence of $\mathrm{gap}_{\min}$ in the final bounds. **Organization of Supplement:** The proofs were structured so that the proofs of the supporting lemmas were deferred until Appendix F. We now recognize that it may be easier to incorporate the proofs of these lemmas at the end of the section, and so we will refactor Appendix F so that the proofs of the lemmas come closer to their statements. We shall also include an "organization" section for the appendix to explain how the different sections fit together, and split the supplement into groups (i.e. Part I. proof of upper bounds Part II. Proof of clipping and other technical results, Part III proof of lower bounds). **Notational Overhead:** Regarding the notational overhead, this seems to be a difficulty encountered by many papers in the MDP regret space, and we intended our notation to be consistent with prior art. While we do include a table of notation, we appreciate the reviewers suggestions that we remind the reader of notation. We will also include remarks that explain notational rational (e.g. overline means "optimistic estimate"). We attempted to lighten notation by providing bounds at three levels of granularity: (in the body, in App A, and in App C), and because of this, we should be able to simplify notation in the proof sketch in the body of the paper by incorporating an informal statement of Prop 3.2 as described above.

**Reviewer 1:** Regarding your concern about the horizon dependence of the gaps, we define a horizon dependent gap in Definition 1 (i.e., for times $h \in [H]$). For simplicity, we state the results in the main text in terms of the minimum over the horizon dependent gaps, though more granular bounds are given in Appendix C. We confirm your suspicion that $\mathrm{gap}_{\min}$, as defined, may be zero if $\mathrm{gap}_h(x,a) = 0$ for all $x,a$. However, our results will go through with a more refined definition of $\mathrm{gap}_{\min} := \min_{x,a,h}\{\mathrm{gap}_h(x,a) : \mathrm{gap}_h(x,a) > 0\}$, and we will be sure to restate our results to use this definition instead. This way, $\mathrm{gap}_{\min}$ is strictly positive unless all actions are equally good for all states and all stages (in which case the regret is trivially zero). We will also include a remark to clarify this point.

**Reviewer 2** Thank you for your helpful feedback. I hope your major concerns are addressed in the organization suggestions listed above. Regarding the general comments: **1.** Line 97 regarding "average" reward. We apologize for the imprecision. **2.** Line 168: We state in Sec 1.2 (the problem setting) that the rewards is a variable is bounded in [0,1]. **3.** Line 200: We will include the quantifier for $h \in [H]$. Thanks for pointing out the omission

**Reviewer 3:** Thank you for the detail feedback **1.** We hope we addressed your concerns about organization in the discussion above. **2.** Regarding point 2, yes you are correct an algorithm is optimistic with high probability. It would be more accurate to state in the definition that "an algorithm satisfies (strong) optimism if...", and then say that we informally refer to an optimistic algorithm as one that satisfies optimism with high probability. **3.** In appendix H, the formal statement of the lower bound does explicitly give the class of algorithms which suffer Thm 2.3, and we will do a better job to signpost this. To our knowledge, every low regret algorithm in the literature for this setting falls under this class, and we can add this clarification as well. Moreover, as mentioned in the discussion about organization above, explaining how optimistic algorithms are constructed from confidence bonuses will allow us to provide, in the main text, an intuitive explanation of the formal class of algorithms which suffer the lower bound. **Additional comments: 4.** We can explain the intuition for the $\epsilon$ parameter in the bound, and add a quantifier ($\forall \epsilon$) in the definition of Zsub. Since the bound takes a minimum over $\epsilon$, it is not a parameter that requires specification. **5.** Thanks for the clarification of Jaksch. We will clarify to state that their gap $\mathrm{gap}_*$ depends on measures of ergodicity that show up implicitly in the other asymptotic analyses. **6.** To clarify what we mean by "almost" gap-independent, we will add a term $\mathrm{poly}(\log(\mathrm{gap}_{\min}))$ to capture an at most poly logarithmic dependence on the min gap. **7.** $\widetilde{S}$ and capital $P$ are typos - thanks for the catch! **8.** Thank you, we will clarify that uniqueness is essential for the proof of [15] **9.** For the inequality on 897 we will clarify that a, b are positive. Then $\sqrt{a} + \sqrt{b} \le \sqrt{a+b} + \sqrt{a+b} = 2\sqrt{a+b}$, whence the inequality follows.

[Meta-Review · NeurIPS 2019]

The paper contributes useful structural results in regret minimization in the Markov Decision Process setting of RL, specifically for the class of tabular (i.e., unstructured) finite-horizon episodic MDPs. The paper is likely to stimulate the finite-sample analysis of online learning in MDPs via its new theoretical techniques.